# Nucleation modeling of the Antarctic stratospheric CN layer and derivation of sulfuric acid profiles

Steffen Münch[1,*] and Joachim Curtius[1]

[1]Institute for Atmospheric and Environmental Sciences, Goethe University Frankfurt am Main, Frankfurt am Main, 60323, Germany
[*]now at: Institute for Atmospheric and Climate Science, ETH Zürich, Zürich, 8092, Switzerland

*Correspondence to*: Steffen Münch (steffen.muench@env.ethz.ch)

**Abstract.** Recent analysis of long-term balloon-born measurements of Antarctic stratospheric condensation nuclei (CN) between July and October showed the formation of a volatile CN layer at 21-27 km altitude in a background of existing particles. We use the nucleation model SAWNUC to simulate these CN in subsiding air parcels and study their nucleation and coagulation characteristics. Our simulations show that the development of the CN layer can be explained with neutral sulfuric acid-water nucleation while outside the CN layer the measured CN concentrations are well reproduced just considering coagulation and the subsidence of the air parcels. While ion-induced nucleation is expected as the dominating formation process at higher temperatures, it does not play a significant role during the CN layer formation as the charged clusters recombine too fast. Further, we derive sulfuric acid concentrations for the CN layer formation. Our concentrations are about one order of magnitude higher than previously presented concentrations as our simulations consider that nucleated clusters have to grow to CN size and can coagulate with preexisting particles. Finally, we calculate threshold sulfuric acid profiles that show which concentration of sulfuric acid is necessary for nucleation and growth to observable size. The threshold profiles should represent upper limits of the actual sulfuric acid outside the CN layer. According to our profiles, sulfuric acid concentrations seem to be below mid-latitude average during Antarctic winter but above mid-latitude average for the CN layer formation.

## 1 Introduction

Atmospheric aerosol particles are of interest due to their various influences on radiation, clouds, chemistry, and air quality. Condensation nuclei counters measure the number concentration of aerosol particles by growing them to optically detectable sizes by condensation (e.g. McMurry, 2000). Therefore, condensation nuclei (CN) are defined as all aerosol particles that are large enough to be measured by a CN-counter, which typically can measure particles with diameters larger than ~10 nm. New particle formation to supply the particles for the stratosphere mostly occurs in the tropical tropopause layer (Brock et al., 1995; Thomason and Peter, 2006; Weigel et al., 2011). The particles are then distributed in the lower stratosphere and constitute the stratospheric aerosol layer (Junge et al., 1961). The particles are sulfuric acid-water droplets (Arnold et al., 1998; Deshler, 2008; Junge et al., 1961) and if they are completely volatile, they are assumed to be formed by ion-induced or neutral

homogeneous nucleation of sulfuric acid and water (binary nucleation). Binary nucleation occurs at low temperatures and high sulfuric acid concentrations. In the stratosphere the lowest temperatures are found in the polar vortex of the winter hemisphere. In the Antarctic polar vortex a background CN concentration of ~10 cm$^{-3}$ is found (Campbell and Deshler, 2014). Volatility measurements indicate that more than half of them have a nonvolatile core which could be meteoric material (Campbell and

Deshler, 2014; Curtius et al., 2005). Sulfuric acid is expected to condense on these CN (Borrmann et al., 2010; Murphy et al., 1998, 2013) and the formation rate of gaseous sulfuric acid should be very small during polar night. Therefore, nucleation is not expected to occur in the polar vortex.

Contrary to this expectation, Rosen and Hofmann (1983) first observed an increase of volatile CN at 25-30 km altitude during winter at Laramie, Wyoming (41°N). They assumed the CN to be freshly nucleated sulfuric acid-water particles with the polar

stratosphere as the source region. Above McMurdo Station, Antarctica (78°S), Hofmann and Rosen (1985) also observed an increased CN concentration between 20 and 25 km after sunrise (CN layer). To check if the occurrence of this CN layer was an annual polar phenomenon, further measurements were performed that also observed the formation of a CN layer after sunrise (e.g. Hofmann (1990) at Kiruna, Sweden (68°N)). Therefore, sulfuric acid production by sunlight after the end of the polar night was suggested as the nucleation source.

Based on these observations, modeling studies began to investigate the formation of the CN layer. Hamill et al. (1990) calculated nucleation rates indicating that binary nucleation could occur in the polar winter stratosphere if sulfuric acid concentrations were high enough. Zhao et al. (1995) developed a one-dimensional (altitude) aerosol model that showed that the transformation of OCS to $SO_2$ and further oxidation of $SO_2$ to sulfuric acid are too slow to reproduce the observed CN increase. They could only reproduce the formation of the CN layer when they added downward transport of $SO_2$ from the

mesosphere inside the polar vortex. Mills et al. (1999) and Mills et al. (2005) presented modeling with a two dimensional (altitude and latitude) aerosol model that was able to reproduce the formation of the CN layer when including production of mesospheric $SO_2$ by photolysis of sulfuric acid and $SO_3$ (see also Vaida et al., 2003).

In summary, and contrary to the initial expectation, nucleation seems to occur in the polar stratosphere. During polar winter, more $SO_2$ is transported downward inside the polar vortex without being oxidized by photochemical reactions. After sunrise,

this $SO_2$ is oxidized to sulfuric acid which initiates nucleation and forms the volatile CN layer despite the presence of nonvolatile background CN.

More recently, Campbell and Deshler (2014) presented an overview of all balloon-borne CN measurements between 15 and 35 km above McMurdo Station, Antarctica (78°S), that were performed between 1986 and 2010, 2-3 times a year during

winter. They present monthly averaged CN concentration and temperature profiles which capture the unperturbed CN, with concentrations around 10-20 cm$^{-3}$ in June/July as well as the development of the CN layer at 21-27 km, with concentrations increasing to 100 cm$^{-3}$ from August until October during sunrise and warming. Campbell and Deshler (2014) also presented volatility measurements of the CN showing that in general more than half of the CN have a nonvolatile core except in the CN layers where they observe significant and rapid formation of new particles that are completely volatile. Additionally, Campbell

et al. (2014) used a 3-dimensional chemistry climate model (English et al., 2011; Hurrell et al., 2013) to reveal the global extent of the CN layer.

Campbell and Deshler (2014) describe a method where they derive an Antarctic sulfuric acid profile from the measured CN by inverting the neutral binary nucleation equation. They used the difference between the CN before sunrise and two weeks after sunrise averaged over all years to derive a nucleation rate for all altitudes from which they derived the corresponding sulfuric acid. This profile is useful e.g. for evaluating global models (Campbell et al., 2014) as no Antarctic sulfuric acid measurements exist. However, Campbell and Deshler (2014) and Campbell et al. (2014) also note that their derived profile might be an underestimation as their method does not consider the particles smaller than their experimental CN detection threshold particle size, losses to preexisting particles, and ion-induced nucleation.

We find this approach of deriving a sulfuric acid profile from the measured CN intriguing. Here we use the nucleation model SAWNUC that simulates small particles, ion-induced nucleation, coagulation, and losses to preexisting particles. We model the Antarctic CN layer based on the observations of Campbell and Deshler (2014), and derive Antarctic stratospheric sulfuric acid profiles.

## 2 Methods

### 2.1 The SAWNUC model

The SAWNUC model (Sulfuric Acid Water NUCeation model, Lovejoy et al., 2004) simulates binary sulfuric acid water neutral and ion-induced nucleation. SAWNUC uses thermodynamic stabilities that are based on experimental values and quantum chemical calculations (Lovejoy and Curtius, 2001; Froyd and Lovejoy, 2003a+b; Hanson and Lovejoy, 2006) and it explicitly simulates step-by-step addition of sulfuric acid molecules in linear size bins for cluster sizes below 2 nm. Above 2 nm particle concentrations are collected in geometric size bins. Here, we simulate 30 geometric size bins with a scale factor of 1.7 which range up to about 400 nm for neutral and negatively charged clusters. For each size bin, SAWNUC can simulate condensation and evaporation of sulfuric acid, coagulation with neutral clusters, recombination of negative clusters with positive ions, and losses to preexisting particles. SAWNUC has been previously described and used (among others) by Lovejoy et al. (2004), Ehrhart and Curtius (2013), Ehrhart et al. (2016), and its parameterized version PARNUC (Kazil and Lovejoy, 2007) is used in Kirkby et al. (2011).

For this study, we extended the SAWNUC model. Coagulation rates between neutral clusters are now calculated including van der Waals forces according to Chan and Mozurkewich (2001) and SAWNUC can use the updated sulfuric acid dimer stabilities reported by Kürten et al. (2015). We redesigned the model code to allow ambient condition changes during a simulation and added the ability to perform multiple simulations within one program run. For this study, the basic processes simulated by SAWNUC are condensation and evaporation of sulfuric acid and coagulation for every size bin. Preexisting particles are fully simulated as particles and not just as surface area to which particles can be lost. Condensation and evaporation of sulfuric acid

are the dominating processes for the formation of new particles while coagulation and condensation of sulfuric acid, if present, determine growth and reduction of existing particles.

## 2.2 Ambient parameters

To perform a simulation of the Antarctic stratosphere with SAWNUC, we need to know temperature, pressure, ion pair production rate, relative humidity, and sulfuric acid concentration. Particle concentrations and sizes are the model output at every time step. As we invert the model, we also need the particle concentrations to derive the sulfuric acid concentrations. Temperatures above Antarctica were taken from Campbell and Deshler (2014). Temperatures that were below 190 K (maximum 5 K below), which is SAWNUC's lower temperature range, are held fixed at 190 K. This introduces some uncertainty which is estimated in our sensitivity test of a 5 K temperature increase (Sect. 3.3). Altitudes were converted to pressures according to the global modeling of Campbell et al. (2014).

The ionization rate of the Antarctic stratosphere in August-September 2010 was 3e5 ion pairs per gram of air and second (Ilya Usoskin, personal communication, 2013; according to Usoskin et al., 2011) which converts to e.g. ~10 ion pairs $cm^{-3}$ $s^{-1}$ at 200 K and 20 hPa.

The water vapor profile for July was chosen to be a linear increase from 3.0 to 6.0 ppm from 18 km to 25 km and above a constant value of 6.0 ppm up to 32 km based on MLS and hygrometer measurements in Fig. 7a in Campbell and Deshler (2014). The mixing ratio is kept constant during the subsidence of the simulated air parcel (see below).

CN concentrations were taken from Campbell and Deshler (2014). The measured CN were compared with the simulated CN by summing over all simulated particles with diameters above 20 nm, as Campbell and Deshler (2014) reported a detection limit of their CN counters of 6-20 nm diameter. As we do not know the exact size of the measured CN, we assume the initial preexisting CN to be large CN with a diameter of 100 nm (see below), but we also perform a sensitivity study assuming a diameter of 50 nm in Sect. 3.3. We simulate them as pure sulfuric acid-water particles but as temperatures are too low for significant evaporation, they could also include a nonvolatile core.

## 3 Results

### 3.1 CN simulations and sulfuric acid profiles

We start our simulations with a simplified *reference case* where we assume for all altitudes (18 - 32 km) and for every month (July - October) a constant monodisperse background CN concentration of 10 $cm^{-3}$ with a size of 100 nm diameter. For this reference case, we do not simulate the highest altitudes in September and October, as there, high temperatures lead to CN evaporation and complicate the interpretation. For all other altitudes and months, we simulate one month with constant ambient conditions chosen according to Sect. 2.2. We use the temperatures reported by Campbell and Deshler (2014) which are reproduced in Fig. 1a. We set the 10 CN $cm^{-3}$ as initial particles at the beginning of the month and simulate the month without gaseous sulfuric acid being present. The CN concentration then reduces somewhat over time as the particles coagulate. Then,

we simulate the month again and derive the gaseous sulfuric acid concentration that leads to a 10 % higher monthly mean CN concentration. This we term the "*nucleation threshold profile*" for sulfuric acid as it defines the minimum gaseous sulfuric acid that leads to nucleation and growth to observable CN size of about 1 additional CN per cm$^3$ and month.

The nucleation threshold sulfuric acid profiles are shown in Fig. 1b. Their shapes are similar to the temperature profiles because temperature, sulfuric acid, and losses to preexisting particles mainly determine the nucleation rate. As we have the same preexisting particle concentrations and target almost the same nucleation rate everywhere, the temperature determines the derived sulfuric acid concentration and our nucleation threshold profiles increase with increasing temperature.

We continue by studying how the measured CN of Campbell and Deshler (2014) coagulate outside the CN layer. Therefore, we drop the assumption of 10 CN cm$^{-3}$ from the reference case and simulate the CN inside air parcels that subside in the polar vortex. The CN profiles presented by Campbell and Deshler (2014) are our basis for the air parcel subsidence trajectories. We assume that the CN maximum of each month resides in a single subsiding air parcel, and place the other air parcel trajectories around this CN maximum trajectory (Fig. 2a). For an air parcel simulation, we set the ambient conditions at the beginning of each month according to Sect. 2.2 and keep them constant for the whole month. We simulate and compare on a monthly basis as the measured input and target values (temperature and CN) are also monthly averages. As we want to study only the coagulation of the CN, we assume no gaseous sulfuric acid being present during the simulation.

For the first month of each parcel simulation, we use the measurements of Campbell and Deshler (2014) to determine the initial CN concentration. Therefore, we set an initial concentration at the beginning of the first month and simulate this month. Coagulation is the only process taking place which reduces the CN by some amount. We choose the initial CN concentration so that the mean CN concentration in the first month matches the measurements.

After the first month, we let the model run free, still assuming no sulfuric acid being present. Therefore, the only two effects on CN concentration are a decrease due to coagulation and a change of the air volume when the ambient conditions change between the months. The latter mostly results in a CN concentration increase due to pressure increase.

Figure 2b shows the simulated CN without sulfuric acid being present and therefore no nucleation. The uncertainty ranges of the measured CN from Campbell and Deshler (2014) are shown for comparison (-10% to +35%). As the first value of every simulated air parcel is chosen based on the observations, the CN in July and at the top in August and September are identical with the measurements. In August, the measured CN can be fully reproduced within the uncertainty range by coagulation and air volume compression of the July CN. No nucleation is necessary. In September and October, the modeled CN concentrations at 20-27 km are too small when no gaseous sulfuric acid is present and the CN layer can not be simulated.

As the next step we study how much sulfuric acid is necessary to form the CN layer and thereby reproduce the observations. We use the same method as described before and start by assuming no presence of sulfuric acid. Only if we do not simulate enough CN without gaseous sulfuric acid, we derive the sulfuric acid concentration that is necessary to produce enough CN, so that the simulated and measured CN match.

Figure 3a shows the sulfuric acid profiles that are necessary to form the CN layer and reproduce the observations (termed "*CN layer profiles*"). Figure 3b shows the simulated CN when we use the CN layer sulfuric acid profiles. Now, the measured CN profiles can be reproduced for all months in almost all areas. As already indicated by Fig. 2b, we only need sulfuric acid in September at 21-26 km to form the CN layer. But also in October at 20-24 km, the presence of sulfuric acid is required as otherwise the CN would decrease too fast due to coagulation.

For a complete interpretation, we combine our nucleation threshold profiles and CN layer profiles in Figure 4. Additionally, we derive the sulfuric acid concentrations that lead to a CN increase in our simulation of the observed CN (Fig. 3b) and include them in Figure 4. We use the same method as for our nucleation threshold profiles (deriving the amount of sulfuric acid that leads to a 10 % CN increase), but now with the simulated CN as background. Note that outside of the CN layer, these profiles represent only *upper limits* for the gaseous sulfuric acid in the atmosphere as neither the observations nor our simulations indicate nucleation in these areas.

In July, August, and September, the profiles show the sulfuric acid that is necessary for nucleation and growth to CN size. The concentrations are higher than our nucleation threshold profiles because we have a higher concentration of preexisting CN compared to the 10 cm$^{-3}$ in the reference case. Therefore, more small clusters are lost by coagulation with large CN. Here nucleation is in competition with losses to preexisting particles (Ehrhart and Curtius, 2013). In October however, the history of the nucleation event in September reduces the sulfuric acid that is necessary for a CN increase. The reason are small particles that still exist from the nucleation event in September and just have to grow above the counting threshold. Therefore, a CN increase requires less sulfuric acid than the nucleation of new particles in our reference case.

The CN layer profile in September has higher concentrations than the upper limit case because more than 10 % additional CN have to form. However, as the nucleation rate is very sensitive to changes in sulfuric acid, the derived sulfuric acid concentrations are not much higher. In October, however, the sulfuric acid in the CN layer lies between our upper limit for no CN increase and our nucleation threshold profile. This shows that no new particles have to nucleate as it suffices that existing small particles grow across the counting threshold to CN size.

Note, however, that the sulfuric acid in the CN layer in October is quite uncertain as the sensitivity studies below show. Additionally, it is only needed when we try to reproduce the measured CN as a monthly mean. Campbell and Deshler (2014) note that most measurements were performed between late August and early October. If the measured CN are representative for the beginning of October, the derived sulfuric acid should be lower or might not be necessary at all.

In September above the CN layer at ~28 km, too many CN are simulated even without sulfuric acid being present (Fig. 2b). This is the result of an air volume compression in the subsiding air parcels from 31 km in August to 28 km in September which increases the CN concentration by ~60 %. Here, coagulation is not efficient enough to reduce the monthly mean CN to the observed value. The high October CN at ~26.5 km are then a result of the high September values. If the August CN at ~31 km would be about a third lower, the simulated CN in September and October would be in the measurement range. Similarly, the

simulated CN in September at ~20 km are a little higher than the observations. Note, however, that we assume no sulfuric acid. If there were some sulfuric acid present, it would condense on the existing CN, increase their size and coagulation efficiency, which would result in lower CN concentrations.

In the following studies we show and discuss only the nucleation threshold and the CN layer profiles to avoid overloaded figures, but the conclusions for the upper limit profiles are analogous to the other profiles.

## 3.2 Ion-induced nucleation

To study the role of ion-induced nucleation, we derived all sulfuric acid profiles again, but this time we do not simulate ions. The comparison is shown in Fig. 5. In areas with low sulfuric acid concentrations removing the ions has nearly no effect on the derived profiles, however, in areas with higher sulfuric acid concentrations the derived profiles increase by almost an order of magnitude. At low sulfuric acid concentrations, the small clusters are not growing fast enough by condensation. Negatively charged clusters recombine too early with positively charged ions and therefore are too small to overcome the nucleation barrier of neutral nucleation. At higher sulfuric acid concentrations, ion-induced nucleation occurs as expected. The charged clusters grow larger than the critical size before they recombine and increase the nucleation rate. Thus to create the same amount of CN without ions, more sulfuric acid is required than if ions are present.

For the nucleation threshold profiles, ion-induced nucleation starts to occur at sulfuric acid concentrations of $\sim 4 \cdot 10^5$ cm$^{-3}$. In the CN layer profiles, however, this limit is higher as more preexisting CN are present that reduce the nucleation efficiency. Therefore in September, the CN layer forms at sulfuric acid concentrations where ion-induced nucleation does not occur in a significant amount. In October, the CN layer is located mostly in the area where ions do change the nucleation rate. However, as no new particles have to nucleate there, but only particles that formed already in September but remained smaller than the CN-counter threshold have to grow to explain the observations, the necessary sulfuric acid hardly changes.

## 3.3 Sensitivity studies

To study the uncertainty introduced by the CN measurement cutoff size we derive the profiles with a lower cutoff diameter of 6 nm, which is the CN counter's lower end according to Campbell and Deshler (2014). The lower cutoff leads to lower sulfuric acid concentrations as the nucleated CN do not have to grow as large by sulfuric acid condensation to be counted (Fig. 6a). This effect decreases with increasing sulfuric acid as at higher concentrations, the clusters grow quickly once they are nucleated. In October, however, there is more sulfuric acid needed in the CN layer as less small clusters exist that can grow across the cutoff size so that sometimes nucleation of new CN is needed.

Lowering the size of the initial preexisting particles from 100 nm to 50 nm diameter reduces their coagulation efficiency and they present a smaller loss during nucleation. Therefore, the sulfuric acid concentrations are lower (Fig. 6b). For the same reason there is no sulfuric acid needed in October in the CN layer. However, the simulated CN concentrations are a little higher (not shown here) so that the observed CN can be better reproduced with an initial particle diameter of 100 nm.

We study model uncertainties according to Lovejoy et al. (2004) by adding 0.5 kcal to all changes in Gibbs free energy of negatively charged clusters. This only increases the profiles in regions where ion-induced nucleation dominates (see Sect. 3.2 and Fig. 5). A reduction of all coagulation and condensation rates by 20% increases all profiles only a little but leads to a poorer CN simulation in comparison with the observations. The updated neutral sulfuric acid dimer thermodynamic stabilities presented by Kürten et al. (2015), which have a higher relative humidity dependence of the equilibrium constant, lead to higher dimer evaporation rates. Therefore they increase our profiles at low relative humidities (high temperatures), but only if neutral binary nucleation dominates there. A combination of these influences is shown in Fig. 6c. The increase of the September CN layer profile at 24-26 km is mainly due to the updated dimer thermodynamic stabilities. The October CN layer profile mostly decreases as coagulation is less efficient which requires less growth of additional small particles. At the lowest altitude no nucleation is needed in September but therefore nucleation of additional CN is necessary in October.

As the derived sulfuric acid profiles are mainly determined by temperature we also test the effect of a 5 K temperature increase (Fig. 6d). We removed the responses at the highest September and October values as there the temperature was too high so that evaporating particles complicate the situation. In general, higher temperatures significantly increase the sulfuric acid profiles. Fortunately, the temperature measurement uncertainty is only 0.5 K (Campbell and Deshler, 2014). However, this temperature sensitivity has to be considered when interpreting the July and August profiles at low altitudes as there the temperature was below SAWNUC's lower temperature range of 190 K (maximum 5 K below, see Fig. 1a).

Additional sensitivity studies (not shown here) showed that the exact amount of ions or water molecules (e.g. 5 ppm everywhere) has only a small influence on the derived profiles because the ion concentrations are high enough so that they are not a limiting factor, and the few parts per million stratospheric water vapor uncertainty is too small. The formation of 35 % more CN in the layer (CN measurement uncertainty) needs only little additional sulfuric acid.

### 3.4 Comparison with mid-latitude sulfuric acid and the derived profile of Campbell and Deshler (2014)

In Figure 7 we compare our derived September CN layer sulfuric acid profile with the profile derived by Campbell and Deshler (2014). Campbell and Deshler (2014) derived sulfuric acid concentrations for 15 to 33 km (dark red, dashed). Our derived sulfuric acid (black, dashed) is only shown between 21 and 26 km as we need no nucleation above and below the CN layer to reproduce the observations. Our concentrations are about one order of magnitude higher. This is because our CN have to form in a background of preexisting particles and they have to grow to observable size. As our sensitivity tests show, both requires more sulfuric acid. In our nucleation threshold profile for a cutoff of 6 nm which has a background of 10 CN cm$^{-3}$ (black, dotted) these two effects are less pronounced. Therefore, this profile compares better with the derived profile of Campbell and Deshler (2014).

Krieger and Arnold (1994) presented inferred sulfuric acid concentrations for the Arctic stratosphere in January 1992. Their concentrations are mostly below or only a little above our July nucleation threshold profile which should represent an upper limit for temperatures colder than in the Arctic. Therefore, our derived Antarctic July nucleation threshold profile is compatible with the inferred Arctic January concentrations of Krieger and Arnold (1994) as no nucleation should occur there.

We can not compare our derived sulfuric acid profiles with Antarctic in situ or remote sensing measurements as such data does not exist to our knowledge. However, northern mid-latitude balloon-borne measurements mainly from September and October have been published (Arnold et al., 1981; Reiner and Arnold, 1997; Schlager and Arnold, 1987; Viggiano and Arnold, 1981) and summarized by Mills et al. (2005). Due to the different tropopause heights (43°N vs. 78°S) our derived profiles might

need to be shifted upwards for comparison. In July and August our profiles lie within the mid-latitude values. As our profiles represent upper limits, Antarctic sulfuric acid should be comparable to or even below the lower mid-latitude concentrations. In September our derived CN layer profile is comparable to or a little above the higher mid-latitude concentrations. This comparison supports the formation explanation of the CN layer with low sulfuric acid during Antarctic winter followed by an area of increased sulfuric acid after sunrise.

We did not derive sulfuric acid profiles above Wyoming according to Campbell and Deshler (2014) Fig. 1a+b, as these CN are assumed to have nucleated in the polar region. However, as temperature is mainly determining the nucleation rate, we can use our nucleation threshold sulfuric acid profiles at temperatures above Wyoming for comparison. In Autumn, the temperature above Wyoming lies between -60°C at 17 km and -40°C at 34 km (Campbell and Deshler, 2014, Fig. 1b). The same temperature range is found over Antarctica in September between 27 km and 33 km (Campbell and Deshler, 2014, Fig. 1d). If we compare

our September nucleation threshold profile between 27 km and 33 km with the mid-latitude values, we see that at all mid-latitude altitudes the concentrations end just below our nucleation threshold values. This suggests that usually no nucleation occurs in the mid-latitudes. Sometimes, however, the sulfuric acid might be close to the limit for ion-induced nucleation becoming efficient.

## 4 Summary and Conclusions

Longtime averages of balloon-born stratospheric CN measurements above McMurdo Station, Antarctica, between July and October reveal the formation of a layer of volatile CN at 21-27 km altitude in a background of preexisting particles (Campbell and Deshler, 2014). We use the nucleation box model SAWNUC to simulate these CN in subsiding air parcels and study the nucleation processes.

We can reproduce the observed CN of Campbell and Deshler (2014) by simulating subsiding air parcels with volume

compression, coagulation, and nucleation. We only need the presence of gaseous sulfuric acid in September for the formation of the CN layer and in October for maintaining it. We show that Antarctic CN concentrations can be explained by coagulation if air volume compression due to air parcel subsidence is considered. This complements Campbell and Deshler (2014) who showed that the CN decrease above Laramie, Wyoming, can be explained by coagulation. Campbell and Deshler (2014) reported that more than half of the CN outside the CN layer contain a nonvolatile core but almost all CN inside the CN layer

are volatile and therefore could be produced by binary nucleation. Our simulations are compatible with these findings as we only need binary nucleation to produce the CN layer. All other CN in our simulation could potentially also contain a nonvolatile core.

In agreement with Campbell and Deshler (2014) we find that the development of the CN layer can be explained by neutral sulfuric acid-water nucleation. The sulfuric acid concentrations are too low and the negative clusters recombine too fast for ion-induced nucleation to occur at significant levels. However, nucleation would be dominated by ion-induced nucleation if it occurred at slightly higher temperatures.

We derived sulfuric acid concentrations for September during the CN layer formation. Our concentrations are about an order of magnitude higher than the concentrations derived by Campbell and Deshler (2014). Our sensitivity tests show that this is because our simulations consider that nucleated clusters have to grow to the CN-counter's threshold size for detection and can coagulate with preexisting particles. Therefore, we can confirm Campbell and Deshler (2014) who suggested that their profiles might be an underestimation due to these effects.

Finally, we derived gaseous sulfuric acid profiles that show which concentration would be necessary for nucleation and growth to CN size to occur. Outside of the CN layer, these values should represent upper limits of the actual sulfuric acid as neither the observations nor our simulations indicate nucleation in these areas. According to these upper limits, sulfuric acid concentrations seem to be below mid-latitude average during Antarctic winter but above mid-latitude average during the CN layer formation. Mid-latitude sulfuric acid concentrations seem to be too low for nucleation to occur. This is also in agreement

with e.g. Campbell and Deshler (2014) who suggest that the mid-latitude CN layer originally formed in the polar region. However, if the stratospheric sulfuric acid would increase above our upper limits, e.g., because of volcanic eruptions or geoengineering, nucleation could occur. In the mid-latitudes and in some relatively warm areas above Antarctica this nucleation would be dominated by ion-induced nucleation and therefore would require less sulfuric acid than predicted by neutral binary nucleation theory. Note, however, that our upper limits would increase if there were more preexisting particles.

In conclusion, our study supports the explanation of the CN layer as presented by Campbell and Deshler (2014). We can reproduce the CN that decrease over time by coagulation in a low sulfuric acid environment during Antarctic winter. These CN could contain a nonvolatile core. In September between 21 and 26 km we can reproduce the observed CN layer only if we assume a higher sulfuric acid concentration that produces volatile CN mainly by neutral binary nucleation.

**Acknowledgements**

We thank Edward R. Lovejoy, Karl D. Froyd, Jan Kazil, and Sebastian Ehrhart for providing the SAWNUC code and Andreas Engel for useful discussion. We also thank the two anonymous reviewers for their help to improve the manuscript.

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

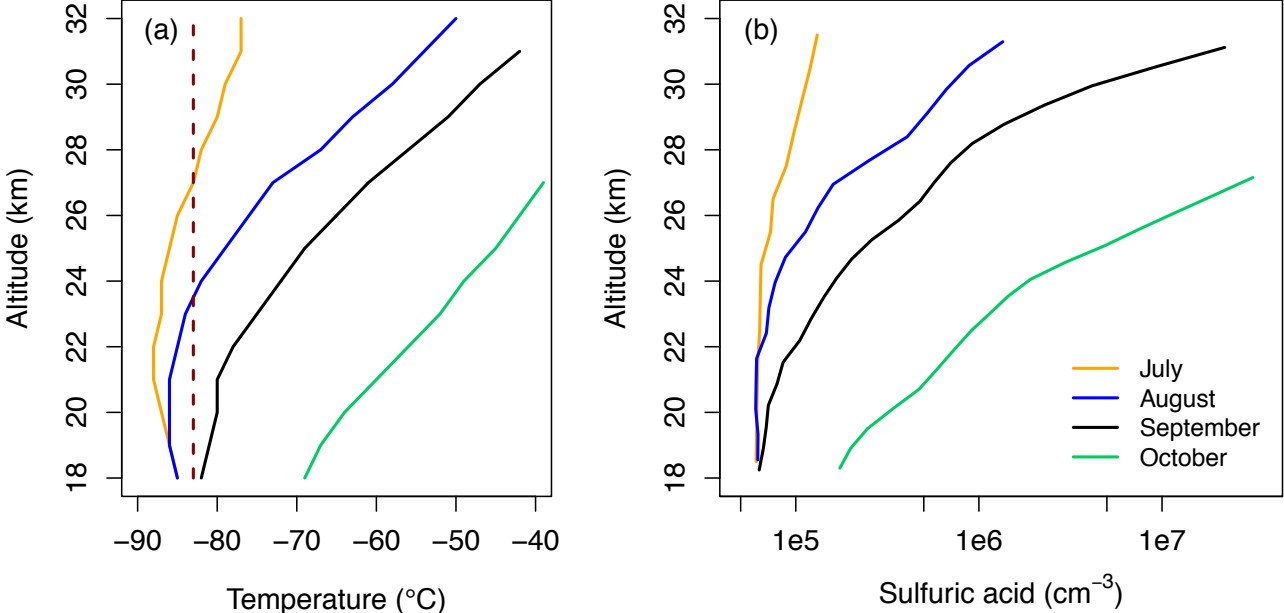

**Figure 1: Temperatures (a) during Antarctic winter above McMurdo, Antarctica (78°S), as presented in Campbell and Deshler (2014). The dashed line shows the lower temperature limit for which the SAWNUC model is valid and at which lower temperatures were kept fixed. In (b), corresponding sulfuric acid profiles are shown that lead to a 10 % CN increase by nucleation and growth to observable size during one month. For these *nucleation threshold profiles*, we assume a monodisperse CN background of 10 cm$^{-3}$ with 100 nm diameter at all altitudes (18-32 km) for every month (July-October).**

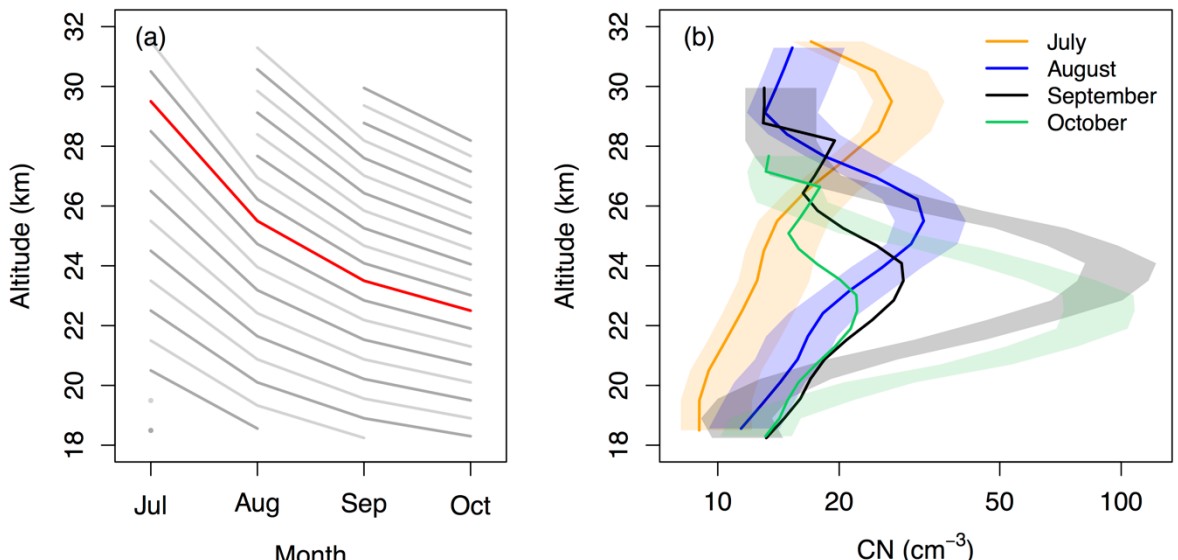

**Figure 2:** Air parcel subsidence trajectories (a) and simulated CN (monthly mean) without gaseous sulfuric acid being present (b). The uncertainty ranges of the measured CN presented in Campbell and Deshler (2014) are shown as shaded areas in (b) for comparison. The trajectories of the simulated air parcels were placed around the subsidence of the measured CN maximum (red). In the simulation, the ambient conditions are kept constant during each month. For the first month of each trajectory, the CN concentrations are chosen based on Campbell and Deshler (2014). In the following months, the simulated CN concentrations are the result of only coagulation and air volume compression, as there is no gaseous sulfuric acid present.

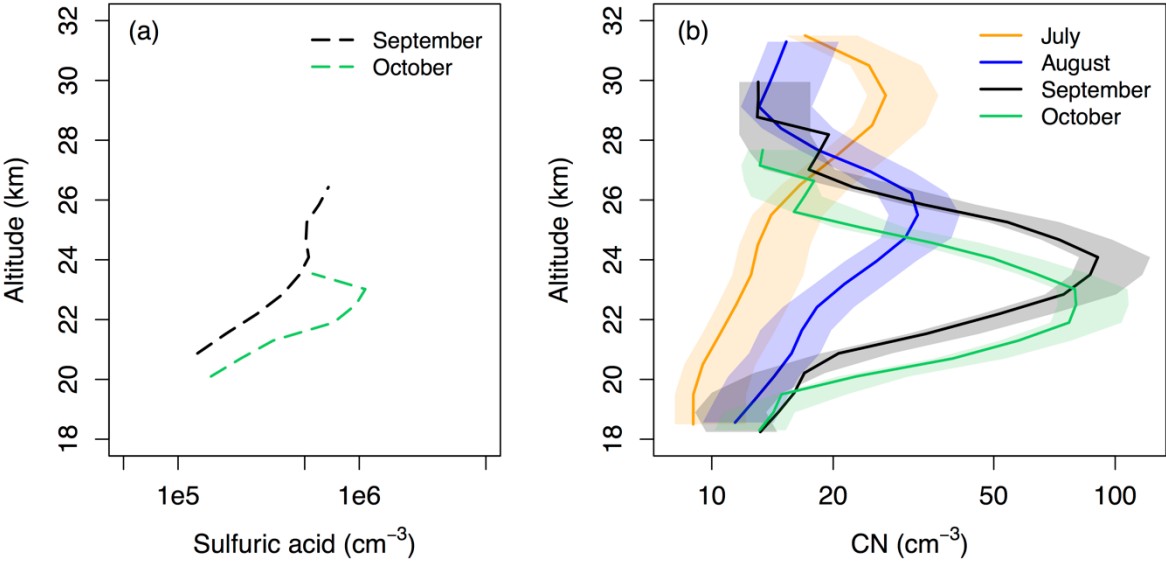

**Figure 3:** CN layer gaseous sulfuric acid profiles (a) and the simulated CN using these profiles (b). We derive the sulfuric acid if the simulated CN concentration in Fig. 2b is too low without gaseous sulfuric acid and therefore nucleation and condensational growth being present. The uncertainty ranges of the measured CN from Campbell and Deshler (2014) are shown as shaded areas for comparison.

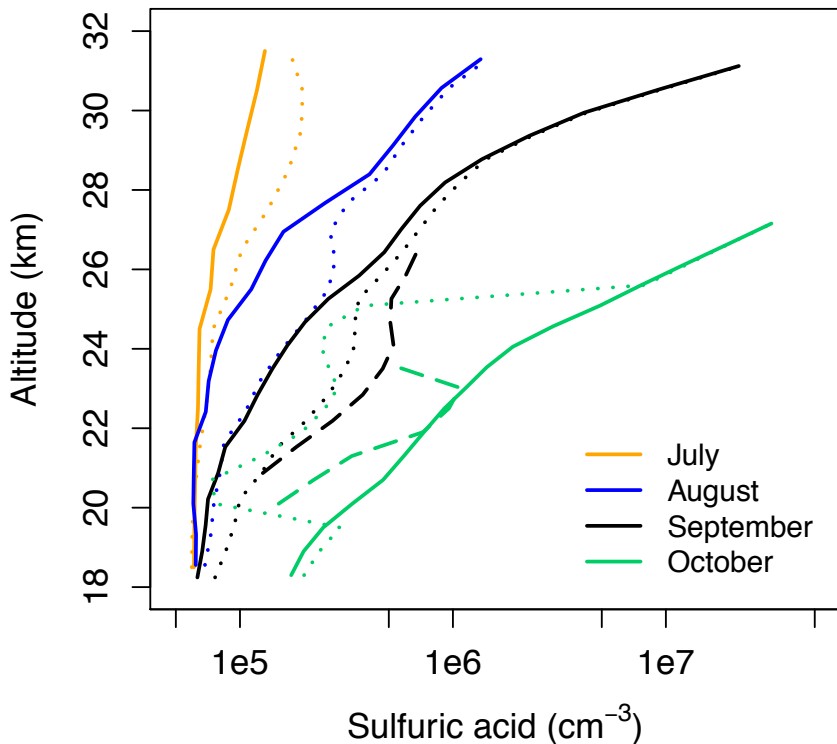

**Figure 4: Combination of the nucleation threshold sulfuric acid profiles from Fig. 1b (solid) and the CN layer sulfuric acid profiles from Fig. 3a (dashed). Additionally, we show sulfuric acid profiles that cause a CN increase in our CN simulation of Fig. 3b (dotted) which should represent upper limits of the Antarctic winter stratospheric sulfuric acid outside the CN layer.**

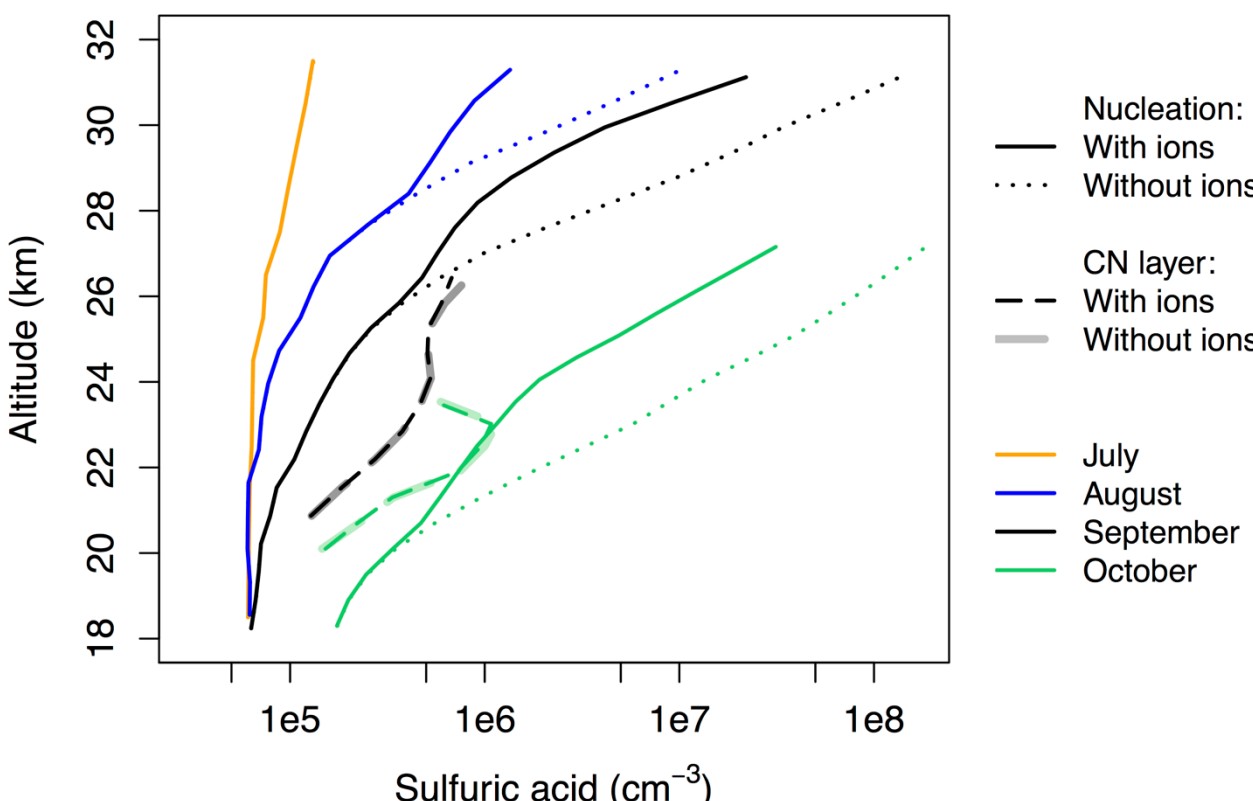

**Figure 5: Comparison of the nucleation threshold sulfuric acid profiles derived including ion-induced nucleation (solid lines) and without simulating ions (dotted lines). At low sulfuric acid concentrations the derived profiles do not change. The CN layer profiles also hardly change (thick dashed lines; grey and light green are without ions and black and green are with ions, but they are almost identical).**

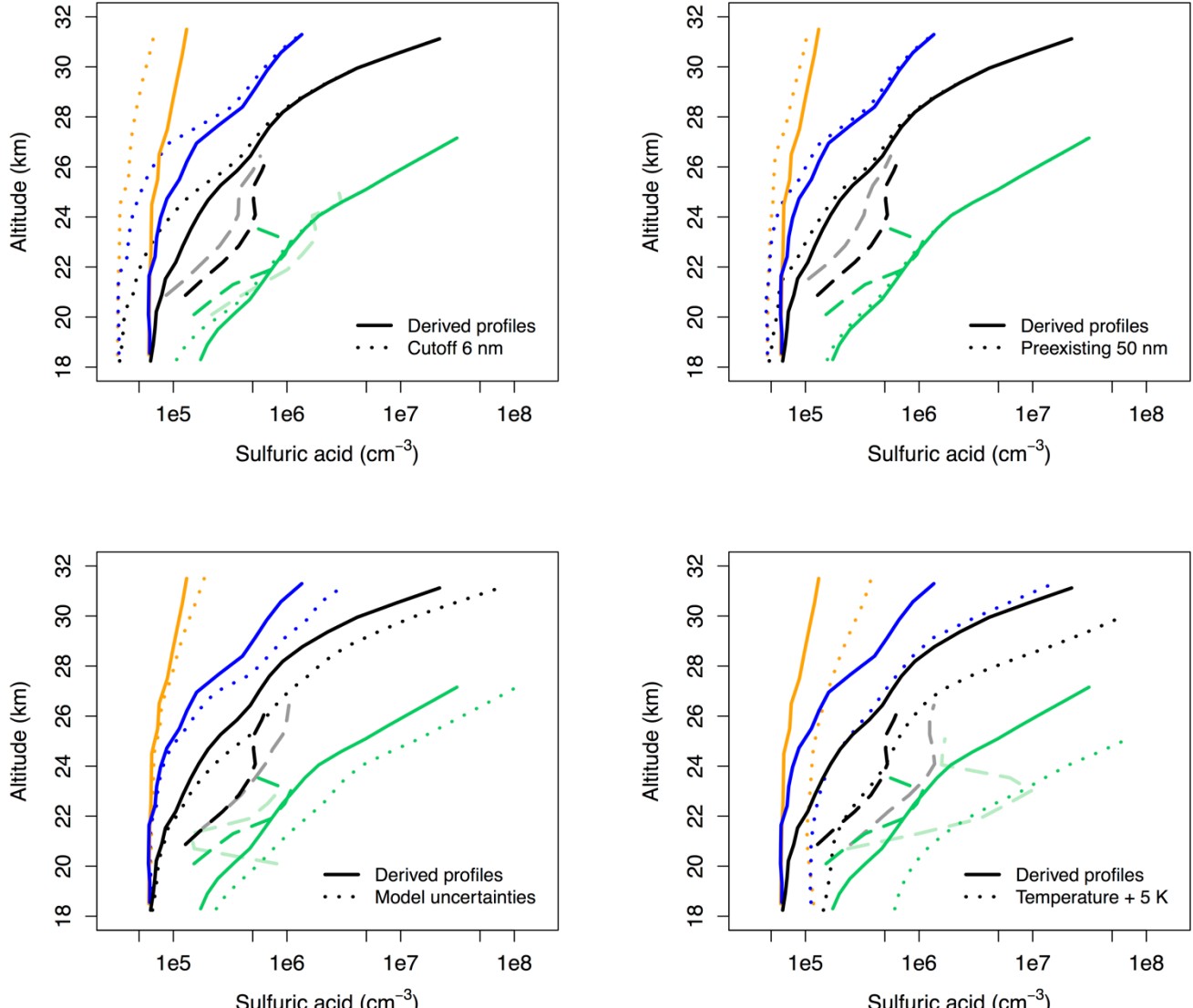

**Figure 6: Sensitivity studies varying (a) CN counter cutoff size, (b) preexisting particle size, (c) model thermochemical and dynamic parameters, and (d) temperature, to estimate the uncertainties of the derived sulfuric acid profiles. As in Fig. 5, the solid and dark dashed lines show the nucleation threshold and CN layer formation profiles as presented in Fig. 4. The dotted and light dashed lines show the changed profiles according to the sensitivity tests.**

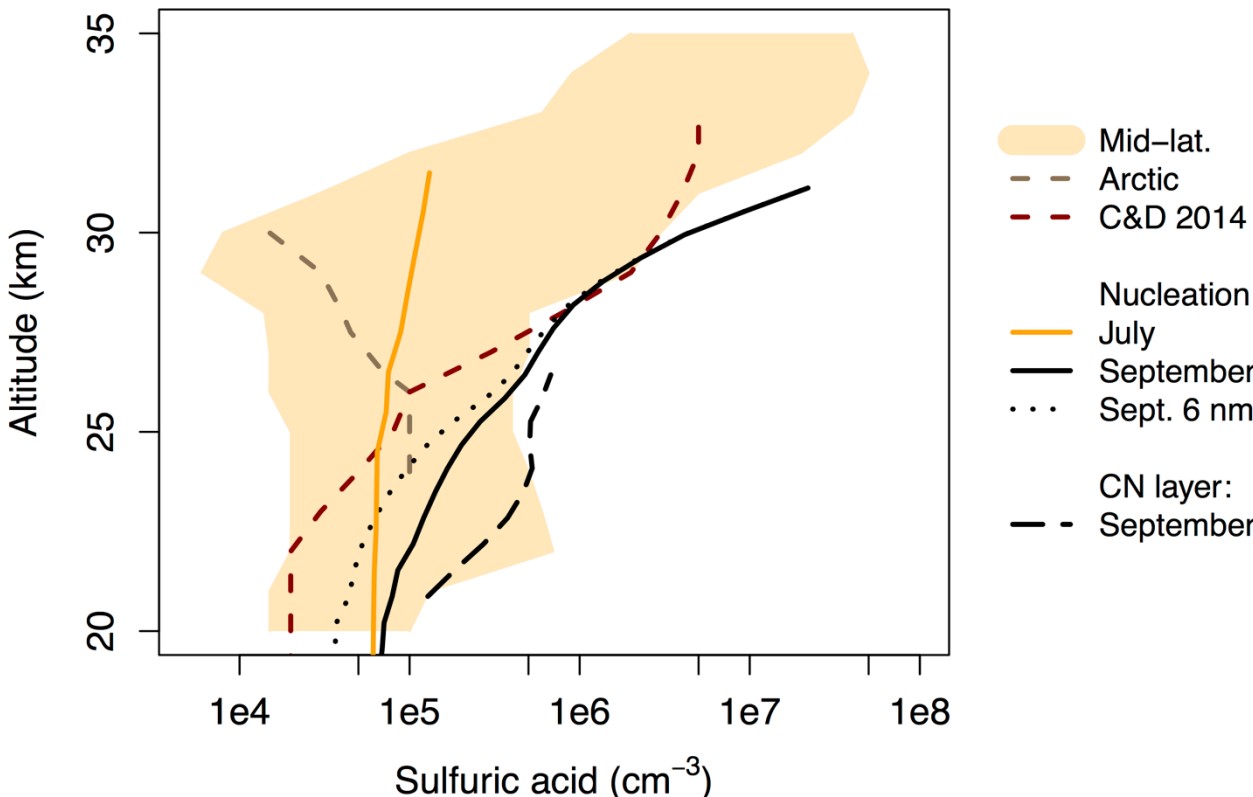

**Figure 7:** Comparison of our derived Antarctic sulfuric acid profiles (nucleation threshold: solid, CN layer: long dashed) with the derived profile from Campbell and Deshler (2014) (dark red, short dashed), inferred Arctic sulfuric acid from measurements in January presented by Krieger and Arnold (1994) (brown, short dashed), and mid latitude measurements and modeling of Arnold et al. (1981), Reiner and Arnold (1997), Schlager and Arnold (1987), Viggiano and Arnold (1981), and Mills et al. (2005) (shaded area). The September nucleation threshold profile for nucleation and growth to a lower cutoff of 6 nm from Fig. 6a is also included (black dotted).