# Peer review of "Nucleation modeling of the Antarctic stratospheric CN layer and derivation of sulfuric acid profiles"

_Atmospheric Chemistry and Physics, 2016_

## Referee Comment (RC1) · Anonymous Referee #1 · 11 Aug 2016

Review For ACP-2016-583: Derivation of Antarctic stratospheric sulfuric acid profiles and nucleation modeling of the polar stratospheric CN layer

Summary and Recommendations:

The paper ACP-2016-583 derives sulfuric acid profiles over McMurdo Station, Antarctica, and investigates nucleation in regards to the observed stratospheric CN layer that has a global extent. It is closely tied to the CN layer work presented in Campbell and Deshler (2014) and Campbell et al. (2014), where they presented a method of inverting the binary homogeneous nucleation rate equation to get "derived" sulfuric acid profiles that are based on measured CN profiles.

[Figure]

The authors of this paper were intrigued by the inversion method of Campbell et al. (2014), and thus extended the nucleation inversion idea by using the more robust SAWNUC model to include more processes and refine the sulfuric acid profiles over Antarctica (for which no measurements exist). Overall, the results in Section 4 compare favorably with Campbell et al. (2014), despite some differences in magnitude and altitude. The paper provides useful information to the scientific community regarding sulfuric acid profiles and a concentration range representative of the winter-spring polar vortex over Antarctica.

Overall, the paper was generally well written and laid out, the methods are sound, and the results are scientifically significant. The paper does, however, include some technical flaws, relatively weak arguments and statements, and suffers from considerable presentation and writing fatigue, especially towards the later sections of the paper. Sentence structure and organization can also be improved. Finally, some of the analysis and presentation of the results needs work. While my formal recommendation is to accept this paper with minor revisions, it is noted that a fair number of changes (with a few technical corrections) should be made before publication. Please see specific comments included here and in the annotated PDF that contains additional writing details for help in these areas.

General Comments:

1. Does the paper address relevant scientific questions within the scope of ACP?

Yes, the paper addresses the scientific question and challenge of deriving sulfuric acid concentration profiles over Antarctica, where no measurements exist. This topic does fit into the scope of material appropriate for ACP.

2. Does the paper present novel concepts, ideas, tools, or data?

The paper does not present an entirely novel concept, as it was first suggested and discussed in Campbell et al. (2014). The paper does, however, extend and refine the

work of Campbell et al. (2014) using a more robust nucleation model, while providing adequate citations and discussions of the Campbell et al. (2014) work.

3. Are substantial conclusions reached?

Yes, the conclusions are substantial in that they provide refined sulfuric acid profiles from a box model over a region where no measurements exist. I envision these profiles can be used in further modeling studies of the Antarctic stratosphere, while also helping to place any potential future model results or observations into context. Furthermore, the results are also used to confirm the validity of the model used to predict the global extent of the CN layer in Campbell et al. (2014).

4. Are the scientific methods and assumptions valid and clearly outlined?

The scientific methods appear valid, however, the choice of some assumptions and approximations used in the modeling need to be further clarified or presented more clearly in regards to the available measurements.

5. Are the results sufficient to support the interpretations and conclusions?

Yes, I agree that the results generally support the interpretations and conclusions stated in this paper.

6. Is the description of experiments and calculations sufficiently complete and precise to allow their reproduction by fellow scientists (traceability of results)?

While the basic description of experiments and calculations are sufficiently laid out, it would be difficult to reproduce them unless more detailed descriptions/equations of the SAWNUC model are provided, possibly in a supplementary section. Although, I understand the complexity of this box model, and that providing all details is not necessarily possible.

7. Do the authors give proper credit to related work and clearly indicate their own new/original contribution?

Yes, there is sufficient credit and citations given, and the authors clearly indicate their new/original contribution.

8. Does the title clearly reflect the contents of the paper?

Yes.

9. Does the abstract provide a concise and complete summary?

Yes.

10. Is the overall presentation well structured and clear?

While the overall presentation is good, there are some areas that can be better structured and written. These include grammatical errors, a cumbersome writing style, confusing section titles and organization, and a general writing fatigue in the later sections of the paper. These issues need to be fixed to improve the overall quality of the paper. I make suggestions in the annotated PDF of the paper.

11. Is the language fluent and precise?

Yes, the language is fluent, but could be more precise in many areas. I make suggestions in the annotated PDF of the paper.

12. Are mathematical formulae, symbols, abbreviations, and units correctly defined and used?

There are no mathematical formulae present, but it may be nice to include some of the fundamental equations used in SAWNUC (e.g., the major nucleation equation that is inverted) possibly in a supplemental section. Showing and explaining the driving terms in the SAWNUC nucleation equation would be helpful.

13. Should any parts of the paper (text, formulae, figures, tables) be clarified, reduced, combined, or eliminated?

There are some areas of clarification needed regarding text and figure presentation

(see specific comments), but I see nothing that need to be completely eliminated. In fact, the addition of some more figures and explanation may further improve the paper.

14. Are the number and quality of references appropriate?

There are some places in the paper where additional references citing relevant work are needed to give credit, and to give further basis for approximations/assumptions made.

15. Is the amount and quality of supplementary material appropriate?

Not necessarily. See comment 12 above.

Specific Comments (please see annotated PDF for additional writing suggestions and corrections):

Page 1, Line 25: This is technically not true, because Campbell and Deshler (2014) note that for balloon-borne measurements, this two-cylinder growth chamber (to grow CN measurements) is mounted vertically on top of an optical particle counter sensitive to particles> 150 nm radius [Rosen, 1964]. This CN instrument has been used to measure CN above Laramie since the early 1980s and above McMurdo since 1986. Thus their CN counters measure particles that could be larger than what you state as a 300 nm upper size threshold for a "typical CN counter". Please revise. Rosen, J. M. (1964), Vertical distribution of dust to 30 kilometers, J. Geophys. Res., 69, 4673–4676, doi:10.1029/JZ069i021p04673.

Page 1, Lines 25-26: It is important to note that Campbell and Deshler (2014) results also qualify Murphy et al. [1998, 2013], where it is suggested that in polar winter, sulfuric acid primarily condenses on nonvolatile meteoritic material. Campbell and Deshler results are in general agreement with this, except in the CN layers with significant, and rapid, formation of new particles without a nonvolatile core. Its possible that the upper boundaries of the CN layer may be impacted by meteoritic material. This may be even more important to introduce here, if some type of sensitivity run is completed by adding

an enhancement in theoretical meteoritic pre-existing particles are added to your study, and impacts on sulfuric acid profiles and CN layers are discussed (see comments in Sections 2 and 3). Campbell, P., and T. Deshler (2014), Condensation nuclei measurements in the midlatitude (1982–2012) and Antarctic (1986–2010) stratosphere between 20 and 35 km, J. Geophys. Res. Atmos., 119, doi:10.1002/2013JD019710. Murphy, D. M., D. S. Thomson, and M. J. Mahoney (1998), In situ measurements of organics, meteoritic material, mercury, and other elements in aerosols at 5 to 19 kilometers, Science, 282, 1664–1669, doi:10.1126/science.282.5394.1664. Murphy, D. M., K. D. Froyd, J. P. Schwarz, and J. C. Wilson (2013), Observations of the chemical composition of stratospheric aerosol particles, Q. J. R. Meteorol. Soc., doi:10.1002/qj.2213.

Page 2, Lines 11-12: Please be consistent with your "sulfuric acid" or "H2SO4" convention through the paper. It is used rather interchangeably.

Page 3, Lines 9 – 11: This is rather different then in Campbell et al. (2014), where their CARMA model divides sulfate aerosols amongst 30 size bins in CARMA, with sulfate mass increasing by a factor of 2.4 between adjacent bins. The dry radii of these 30 aerosol size bins range from 0.343nm to 2.17 $\mu$m.

Page 4, Line 9: Is it technically true to call the SAWNUC output CN concentrations, as you have previously defined CN from a measurement perspective? The output from SAWNUC contains concentrations of particles that are not observable in CN counters at the supersaturations employed in Campbell and Deshler, 2014 ($\sim < 3 - 10$ nm). Maybe just state "Particle concentrations and sizes...."

Page 4, Lines 12-13: What is the chosen 5 nm detection limit based on here? Is this an arbitrary cutoff selection for the model, or do you have appropriate reference for this? Campbell and Deshler (2014) report that their CN measurements contain all particles r > 3 - 10 nm in size, dependent on pressure altitude.

Page 4 Lines 16 – 17: Provide justification for this H2O profile. Show references, such as "...based on July MLS and hygrometer measurements in Figure 7a in Campbell and

Deshler (2014)", or if you used a combination of different measurements to get your assumed profile.

Page 5, Lines 9 – 11: What altitude(s) is (are) represented in the model "maximum trajectories" in Figure 1? This information is very important to describe and show. If these "maximum trajectory" altitudes are representative of the matched maximum CN concentrations in Campbell and Deshler (2014) Figure 1, then better explanation is needed in this section.

Page 5, Lines 17 -21: Why a decrease in10% of CN from the May 1 - June 15 period to the beginning of June/July period? Can this possibly be based on limited CN measurements in June/July from Campbell and Deshler (2014)? Is this an arbitrary value?

Page 6, Lines 7 – 8: What are these "assumed functions"? Are they derived from Campbell et al. (2014), for example in Figure 3 they show time series of CN concentration and maximum CN altitude with time for McMurdo, derived from many years of measurements at different julian days.

Page 6, Lines 32-33: Does this statement really justify that neglecting temperatures < 190 K introduces little uncertainty. Suggest rewording, if not additional discussion as to the uncertainty it really introduces to the nucleation rate. See Figure 5d.

Page 7, Lines 2-3: What part of the design does not allow for these three trajectories in Sept? Rather a mystery here for the reader.

Page 7, Line 28: What is the total stratospheric sulfur value based upon?

Page 7, Lines 29 – 30: Issue regarding your definition of a "typical CN Counter" and the statements made here. Typical CN counters in Campbell and Deshler (2014) and Campbell et al. (2014) measure all particles with r > 3 - 10 nm (dependent on pressure/altitude), and thus could technically count particles above 300 nm. Thus, I do not understand why the derivation mechanism fails here.

Page 8, Lines 3 – 5: Also Murphy et al. (2013) suggested that in polar winter, sulfuric acid primarily condenses on nonvolatile meteoritic material. The results of Campbell and Deshler (2014) are in general agreement with this at higher altitudes, except in the CN layers with significant, and rapid, formation of new particles without a nonvolatile core. Murphy, D. M., K. D. Froyd, J. P. Schwarz, and J. C. Wilson (2013), Observations of the chemical composition of stratospheric aerosol particles, Q. J. R. Meteorol. Soc., doi:10.1002/qj.2213.

Page 8, Lines 8 – 21: This is an intriguing comparison to measured mid-latitude profiles indeed. However, couldn't the SAWNUC inversion method also be applied to derive a mid-latitude sulfuric acid profile using the mid-latitude seasonal CN/temperature profiles shown in Figure 1a and 1b of Campbell and Deshler (2014)? I would like to see an additional derived mid-latitude sulfuric acid profile since it is more a direct comparison to the mid-latitude measurements, and also does not incur the same altitude/adiabatic expansion shifts when comparing polar to mid-latitude regions. The summary of Figure 1 in Mills et al. (2005) show sulfuric acid measurements mainly during September and October, so you could use the representative fall season CN/temp profiles in Campbell and Deshler Figure 1a and 1b to derive a "fall mid-latitude sulfuric acid profile". This can provide supplemental information on the performance of the SAWNUC inversion method used.

Page 8, Lines 31 – 33: You should add that these results also further support Campbell and Deshler (2014) findings that the majority of the observable polar CN layer is due to neutral nucleation.

Page 9, Line 8: What (if any) sensitivity runs were performed due to holding H2O profile constant? Were different H2O profiles tested? It is likely a small uncertainty in the nucleation rate, but may be worth investigating and mentioning.

Page 9, Lines 11 – 14: Since the profiles derived are $\sim$> 20 km, then why is the cutoff > 5 nm chosen, which is more representative of lower atmosphere conditions. Shouldn't

the profiles be derived using a cutoff closer to > 20 nm (as reported in Campbell and Deshler, 2014)?

Page 9, Lines 15 – 16: If the main growth mechanism to a higher observable cutoff size was due to coagulation, then does the sulfuric acid concentrations have to increase in the nucleation areas? In other words, what effect does coagulation rates have on this sensitivity? Also, the profiles are nearly the same above about 28 km between June/July and September, regardless of cutoff size.

Page 9, Lines 21-22: Ultimately, it would be best to avoid using a constant CN measurement cutoff size, since it is clearly dependent on altitude. Can this be varied for the multiple trajectories used in SAWNUC to derive the sulfuric acid profiles, possibly arriving at a "best guess" sulfuric acid profiles over McMurdo?

Page 10, Line 10: Does both coagulation and condensation rates both have little influence?

Page 10, Line 16: What is this inter-annual CN increase based on? Campbell and Deshler (2014) indicate a range in CN concentrations dependent on altitude in Figure 3, and for the CN layer maximum level in Figure 4 for McMurdo.

Page 10, Line 28: This whole section seems rather out of place in the paper, and would be better placed as the last summary part for Section 3, and named "3.4 Summary and discussion of model uncertainties"

Page 11, Line 1: I don't see a compensation between the monthly values and cutoff size, as they are in the same direction for both June/July to September and in October. Additional clarification is needed.

Page 11, Lines 5 - 6: If accepting my previous comment about moving this whole section to the last part in Section 3, this statement could then be made more robust and stated such as the following: "The June/July profiles have additional uncertainty due to rather unknown sulfuric acid production rates from processes that do not require

sunlight over Antarctica in the winter. This uncertainty is compounded by sparse CN measurements made during June/July (and prior) over Antarctica."

Page 11, Lines 7-8: This should not be a separate paragraph if that was what was intended. Also, I do not know if this can be quantitatively determined from the results presented. Please make sure to state that this is a qualitative approximation. Finally, from the discussion of the sulfuric acid production rates for June/July, it seems that this period may have even greater uncertainty compared to the October period, especially considering the much less CN measurements during June/July compared to October (see Figure 3a in Campbell and Deshler, 2014)

Page 11, Lines 17 – 18: Shouldn't this be "therefore this production term cannot represent any sulfuric acid that evaporates from preexisting particles"

Page 12, Lines 2 – 3: Please clarify what you mean by "one additional step away from the measurements"

Page 12, Lines 3 – 4: This is a confusing argument. You note that the derived sulfuric acid profiles in Figure 6 could be an overestimate (by a max of one order of magnitude) in this model, but then wouldn't the inclusion of these processes further increase the sulfuric acid production rates, and thus lead to a possibly even greater overestimate (although understandably unknown). In essence, it may indeed be difficult to compare to the limited measurements of Krieger and Arnold over the Arctic, and may in fact be underestimated in this model for the Antarctic June/July period. Please revise this statement.

Page 12, Lines 8 -12: Why not generate a plot to show these temperature, sulfuric acid, and CN comparisons. It would be much clearer for the reader to associate the altitude discrepancies between this work and Campbell et al. (2014) during early July and late October, and beneficial to the discussions that follow in this section.

Page 12, Lines 16 – 17: While this is indeed possible, is there any evidence of OH

production of sulfuric acid over the Antarctic region? I would be very careful in drawing such suggestions that cannot be backed by measurements. Also, you make suggestions regarding July underestimate by global model compared to SAWNUC, but what about the October overestimate?

Page 12, Lines 18 – 23: This argument seems to suffer from some writing fatigue and is rather weak. No major assessments are made as to why this altitude shift of nucleation controlling variables occurs in the global model. Please improve this paragraph and make better supporting statements/arguments. Are you (and the satellite obs. by Hopfner et al., 2013) suggesting too high SO2 at hight altitudes in the global model compared to observations and SAWNUC? And what can be inferred as to the cause of this using your SAWNUC model (e.g., enhanced SO2 subsidence from mesosphere)?

Page 12, Lines 24 – 28: Section 3 and Figures 1 and 3 in Campbell et al. (2014) already support the ability of the global model to reproduce the observed temperature, water vapor, and CN profiles Antarctica and mid-latitudes, with the exception of an altitude shift in CN over Antarctica, and possible underprediction in magnitude of sulfuric acid. Again, can you provide potential reasoning into why this altitude shift occurs for all nucleation controlling variables against SAWNUC? Have you checked to make sure that there are no discrepancies between altitude definitions in the global model and how they are defined in SAWNUC to ensure an apples-to-apples comparison? Remember that the global model is an Eulerian framework, while the trajectories predicted in SAWNUC are Lagrangian.

Page 13, Lines 6 – 7: This is misleading as stated, because it implies that the global model in Campbell et al. (2014) was run with their derived sulfuric acid profiles, while in fact, it was run with the WACCM sulfuric acid profiles, and the derived profiles were used for further insight and to assess possible uncertainties in the WACCM profiles. Please revise.

Page 13, Lines 16 – 18: This argument could be improved from my previous comment regarding derived SAWNUC sulfuric acid profiles over mid-latitudes compared to measurements over mid-latitudes.

Page 14, Lines 2-3: While this section adequately summarizes the findings presented here, more discussion on what SAWNUC can answer as to why the global model predicts sulfuric acid at too high an altitude, will make the discussion and results more robust in this paper. What are some of the potential processes leading to this altitude shift?

Please also note the supplement to this comment:
http://www.atmos-chem-phys-discuss.net/acp-2016-583/acp-2016-583-RC1-supplement.pdf

**Supplement:**

[Figure]

**Derivation of Antarctic stratospheric sulfuric acid profiles and nucleation modeling of the polar stratospheric CN layer**

Steffen Münch[1,*], Joachim Curtius[1]

[1]Institute for Atmospheric and Environmental Sciences, Goethe University Frankfurt am Main, Frankfurt am Main, 60323, Germany
[*]now at: Institute for Atmospheric and Climate Science, ETH Zürich, Zürich, 8092, Switzerland

*Correspondence to*: Steffen Münch (steffen.muench@env.ethz.ch)

**Abstract.** Recent analysis of long-term balloon-borne measurements of Antarctic stratospheric condensation nuclei (CN) and temperature combined with global model calculations showed the wide extent of a mid stratospheric layer of new particles. Here the nucleation model SAWNUC is used to derive Antarctic stratospheric gaseous sulfuric acid profiles and to investigate the nucleation process of this CN layer. The sulfuric acid profiles were derived for an altitude range of 18-32 km between July and October by simulating air parcel trajectories that descend inside the polar vortex and calculating the sulfuric acid amount that reproduces the observations. The derived sulfuric acid concentrations (volume mixing ratios) are of the order of magnitude of $10^4$ cm$^{-3}$ ($10^{-14}$) in July. In the following months the concentrations increase to about $10^7$ cm$^{-3}$ ($10^{-11}$) in October. They depend strongly on the temperature because a given temperature leaves only a small sulfuric acid range to reproduce the observed magnitude of CN. Ion-induced nucleation occurs. However, while it dominates nucleation at higher temperatures it has no significant influence on the nucleation rates at lower temperatures. Preexisting particles significantly reduce nucleation at sulfuric acid mixing ratios below 1 ppt. First estimates of sulfuric acid production rates range from 0.5 to about 500 molecules cm$^{-3}$ s$^{-1}$. A production mechanism for gaseous sulfuric acid during the Antarctic winter seems to be necessary to fully explain the observations. The derived sulfuric acid profiles compare well with mid-latitude and Arctic sulfuric acid concentrations.

**1 Introduction**

When investigating the condensation nuclei layer in the stratosphere, condensation nuclei (CN) are defined as all aerosol particles that are large enough to be measured by a CN-counter but too small to be measured by an optical particle counter, typically covering a range of particle diameters between ~10 and 300 nm. The particles in the CN layer are assumed to be formed by ion-induced or neutral homogeneous nucleation of sulfuric acid and water (binary nucleation).

Rosen and Hofmann (1983) first observed an increase of volatile CN at 25-30 km altitudes during winter at Laramie, Wyoming (41°N). They assumed the CN to be freshly nucleated sulfuric acid water particles with the polar stratosphere as the source region. Above McMurdo Station, Antarctica (78°S), Hofmann and Rosen (1985) also observed an increased CN concentration between 20 and 25 km after sunrise (CN layer). To check if the occurrence of this CN layer was an annual

polar phenomenon further measurements were performed which also observed the formation of a CN layer after sunrise (e.g. Hofmann (1990) at Kiruna, Sweden (68°N)). Therefore, sulfuric acid production by sunlight after the end of the polar night was suggested as the nucleation source.

With these observations modeling started to investigate the formation of the CN layer. Nucleation rate calculations by

5   Hamill et al. (1990) indicated that binary nucleation could occur in the polar winter stratosphere if sulfuric acid concentrations were high enough. Zhao et al. (1995) developed a one-dimensional (altitude) aerosol model that showed that the transformation of OCS to $SO_2$ and further oxidation of $SO_2$ to $H_2SO_4$ are too slow to reproduce the observed CN increase. However, they could reproduce the formation of the CN layer when they added downward transport of $SO_2$ from the mesosphere inside the polar vortex. Mills et al. (1999) and Mills et al. (2005) presented modeling with a two dimensional

10  (altitude and latitude) aerosol model that was able to reproduce the formation of the CN layer when including production of mesospheric $SO_2$ by photolysis of sulfuric acid and $SO_3$ (see also Vaida et al., 2003).

In summary, $SO_2$ is produced in the mesosphere by $H_2SO_4$ photolysis. During polar winter, more $SO_2$ is transported downward inside the polar vortex without being oxidized by photochemical reactions. After sunrise, this $SO_2$ is oxidized to sulfuric acid which initiates nucleation in the cold polar stratosphere and forms the CN layer.

More recently, Campbell and Deshler (2014) presented averaged balloon-borne CN measurements between 15 and 35 km above McMurdo Station, Antarctica (78°S) that were performed continuously for 24 years. They capture the already existing CN with concentrations around 20 cm$^{-3}$ in June/July as well as the layer's development at 20-25 km to concentrations of up to 100 cm$^{-3}$ from August until October during sunrise and warming by presenting monthly averaged CN concentration and

20  temperature profiles.

Campbell et al. (2014) used a 3-dimensional chemistry climate model (Hurrell et al., 2013, English et al., 2011) and revealed the global extent of the CN layer. They simulated the year 2010 and compared the results to measurements above McMurdo Station (Campbell et al., 2014). The model reproduces the CN layer but at higher altitudes (around 30 km in the model vs. around 23 km in the observations). As an explanation they suggested (among others) biases in the critical nucleation

25  variables: temperature, sulfuric acid and water concentration. However, there are no Antarctic stratospheric sulfuric acid measurements. Therefore, they inverted the nucleation equation to calculate the sulfuric acid concentration that corresponds to the CN increase over three weeks between two measurements. Their derived profile indicates that in the global model the sulfuric acid also has a shift towards higher altitudes which could explain the altitude shift of the simulated CN layer.

30  The approach of deriving a sulfuric acid profile from the measured CN is intriguing. However, they did invert a nucleation formulation that describes the nucleation with only one equation. No ion-induced nucleation, coagulation, or losses to preexisting particles were considered during the inversion. Therefore in this study, we use the nucleation model SAWNUC, which simulates all these processes, to derive the Antarctic stratospheric sulfuric acid profiles and to investigate the

processes that influence the nucleation of the CN layer. We also  estimates of the corresponding sulfuric acid production rates.

The model and the derivation process are described in Sect. 2. The derived profiles, the role of ion-induced nucleation, the uncertainties and estimated sulfuric acid production rates are presented and discussed in Sect. 3.  derived profiles

5 are compared to the global modeling of Campbell et al. (2014) in Sect. 4.

**2 Methods**

**2.1 The SAWNUC model**

The SAWNUC model (Sulfuric Acid Water NUCeation model, Lovejoy et al., 2004) simulates binary sulfuric acid water neutral and ion-induced nucleation. It explicitly simulates the step-by-step addition of sulfuric acid molecules in linear size

10 bins for cluster sizes below 2 nm. Above 2 nm the particle concentrations are collected in geometric size bins. Here, 30 size bins with a geometric scale factor of 1.7 were used which range up to about 400 nm. Neutral and negatively charged clusters were simulated. For each size bin, SAWNUC simulates condensation and evaporation of sulfuric acid, coagulation with neutral clusters, recombination of negative clusters with positive ions, and losses to preexisting particles or chamber walls. Condensation, coagulation, and preexisting loss rates are calculated based on the hard sphere collision theory of Fuchs

15 (1964). For the charged clusters, the Coulomb forces are calculated based on the intercluster potentials (Yu and Turco, 1998). For the neutral clusters, thermodynamic stabilities were calculated with the On-line Aerosol Inorganics Model (Carslaw et al., 1995). The neutral thermodynamics are adjusted to reproduce experimental nucleation rates of Ball et al. (1999). The thermodynamic stabilities of the negative clusters are calculated with the Thomson equation. However for small clusters, the values reported by Froyd and Lovejoy (2003a) and Lovejoy and Curtius (2001) are directly implemented into

20 the model, which are based on experimental values and quantum chemical calculations (for more details see Lovejoy et al., 2004). For the neutral dimer and trimer, the thermodynamic stabilities presented by Hanson and Lovejoy (2006) are also implemented into the model. The SAWNUC model (Lovejoy et al., 2004) has been previously used (among others) in Ehrhart and Curtius (2013) and its parameterized version PARNUC (Kazil and Lovejoy, 2007) in Kirkby et al. (2011).

For this study the SAWNUC model was extended. Coagulation rates between neutral clusters are now calculated including

25 van der Waals forces according to Chan and Mozurkewich (2001). The updated sulfuric acid dimer stabilities reported by Kürten et al. (2015) can be used for the simulation. The model code was redesigned to allow ambient condition changes during a simulation. Also the ability to perform multiple simulations within one program run was added and is used for the model "inversion" in this study.

**2.2 Deriving the profiles**

30 This section describes the derivation of the Antarctic stratospheric sulfuric acid profiles. The profiles are based on the measured CN concentrations and temperatures above McMurdo Station, Antarctica (78°S) as presented by Campbell and

[Figure]

Deshler (2014, Fig. 1). Corresponding sulfuric acid concentrations and mixing ratios were derived for the time interval of June/July to October at altitudes from 18 km to 32 km. The SAWNUC model was "inverted" by performing multiple simulations with the same ambient conditions but varying sulfuric acid concentrations, and searching for the sulfuric acid concentration that reproduces the observed CN. Thereby, all effects like coagulation, ions and preexisting particles are included in the inversion process.

**2.2.1 Ambient parameters**

To perform a simulation with SAWNUC, values for the following ambient parameters are needed: pressure, temperature, relative humidity, sulfuric acid concentration, ion pair production rate, as well as surface area and diameter of preexisting particles. CN concentrations and sizes for every time step are the model output. As the model was inverted, CN concentrations were also needed to search for the sulfuric acid concentrations.

Temperatures and CN concentrations were taken from Fig. 1 (Campbell and Deshler, 2014). The CN concentrations were compared with the measurements by summing over all CN concentrations with sizes above a detection limit (5 nm diameter was used in this study). Altitudes were converted to pressures according to the global modeling of Campbell et al. (2014).

The ionization rate of the Antarctic stratosphere in August-September 2010 was 3e5 ion pairs per gram of air and second (Ilya Usoskin, personal communication, 2013; according to Usoskin et al., 2011).

The water vapor profile was chosen to be a linear increase from 3.0 to 6.0 ppm from 18 km to 25 km and then a constant value of 6.0 ppm up to 32 km in July. This profile was moved down by 1 km every month (to represent diabatic descent within the polar vortex).

The diameter of preexisting particles was assumed to be 100 nm. A surface area of 0.2 $\mu m^2$ $cm^{-3}$ was chosen for 215.15 K and 30 km altitude, which is consistent with Zhao et al. (1995) who reported that the subsiding mesospheric air is very clean and with Campbell et al. (2014) reporting a surface area of 0.3 $\mu m^2$ $cm^{-3}$ for 20-30 km altitude in early August 2010. This surface area was converted to each temperature/height combination according to the ideal gas equation.

Sensitivity tests concerning the influence of all these input values on the derived profiles were performed and are presented in Sect. 3.2.

**2.2.2 The CN layer trajectory**

With the described model setup, the sulfuric acid profiles were derived by using the SAWNUC model as a box model, simulating the nucleation process inside air parcels over the period of the four months. The most important air parcel trajectory for this study is the one that connects the monthly maximum of the measured CN (in the following termed "maximum trajectory"). It is assumed that this maximum of the particle concentration resides in a single air parcel that descends inside the polar vortex. This section describes the derivation process of the sulfuric acid values in the maximum trajectory.

[Figure]

Sulfuric acid concentrations were searched for on a monthly basis as the measured input values (CN and temperature) are also monthly averages. The ambient conditions and the sulfuric acid concentration were set at the beginning of a month and kept constant for the entire month. Simulation results were evaluated for each month. The simulated CN concentrations averaged over the month were compared to the measured values.  the input sulfuric acid

5  concentration was decreased and the model run was repeated. If the model  CN, the input sulfuric acid concentration was increased. This process was reiterated until the sulfuric acid concentration was determined that corresponds best to the measured monthly CN concentrations. This resulted in a derived sulfuric acid concentration of the simulated month.

This scheme was used for every month. The time evolution of temperature (representative for all ambient conditions), the

10  derived sulfuric acid volume mixing ratios (converted from the concentrations), and the simulated CN concentrations (> 5 nm) of the maximum trajectory's boxmodel simulation are shown in Fig. 1a. The derivation process is divided into five periods: the initialization phase and the June/July, August, September, and October periods (June/July are combined as the measurements in Campbell and Deshler, 2014, also combine these months).

Model initialization is necessary as some CN should already be present at the beginning of the first derived month. However,

15  as the CN concentrations before June are not known, the trajectory simulation was started with an initialization phase from May 1st to June 15th, in which it builds up 110% of the June/July CN amount at June/July ambient conditions. This extra 10% was chosen because CN concentrations are expected to be higher at the beginning of the June/July period than at its end (because of missing sulfuric acid production and air compression, see below).

The June/July sulfuric acid concentration was derived by searching for the sulfuric acid concentration that reproduces the

20  measured June/July CN amount. This sulfuric acid concentration is lower than during the initialization period as no new particles have to form and the existing ones have to decrease in number.

At the beginning of August the ambient conditions were changed to the August conditions (see temperature change in Fig. 1a). As the pressure increases during the descent of the air, this ambient change also includes a compression of the air volume and thereby an increase in the CN per cm$^3$. This is seen in the CN jump at the beginning of August. The compression

25  due to pressure increase is stronger than the expansion due to the temperature increase (both were calculated and combined). After the air compression, the sulfuric acid concentration was searched that reproduces the measured August mean CN amount. Here, a decrease from the compression-increased CN amount is necessary. Therefore, what seems like an increase in CN from July to August in the maximum trajectory values turns out to be a decrease due to the adiabatic compression.

At September 1st the ambient conditions were changed to the September conditions, the CN were compressed (only a small

30  change can be discerned in Fig. 1a), and the sulfuric acid concentration was searched for that reproduces the September CN amount.

Finally in October the same procedure was used. Unfortunately, the use of a month-long time interval for temperature and sulfuric acid concentration produces an undesired increase of CN at the beginning of October  Stable clusters below the counting threshold of 5 nm still exist from September and rapidly grow due to the

strongly increased sulfuric acid concentration in October. Only later in the month, the CN amount decreases as only few new particles are produced while the old CN coagulate and are lost to preexisting particles. This unrealistic behavior can only be addressed by modeling at higher time resolution where the ambient conditions are changed more gently in shorter time steps.

5    Therefore, to avoid this undesired increase at the beginning of October and to reduce the „compression jumps", the same derivation process was preformed with shorter time steps of 5 days. However, time developments of altitude, temperature, and CN at the shorter time steps had to be assumed as the measured values are presented as monthly averages. The time developments were described by assumed functions for which the monthly mean values match with the measured values. Therefore, the simulated maximum trajectory with 5-day time steps (Fig. 1b) follows these assumed functions. The

10   adjustments of the CN concentration to the air compressions are still visible at the beginning of every time step, but they are much smaller compared to the monthly simulations as the ambient changes are much smaller. Comparing the monthly average of the 5-day sulfuric acid mixing ratios with the derived monthly values shows that the monthly simulation overestimates the derived sulfuric acid for June/July to September and underestimates it for October. However, the difference always stays below a factor of 2. As the derived mixing ratios in this study span over several orders of magnitude,

15   this is considered a reasonable agreement. This comparison strengthens the confidence in the monthly derived sulfuric acid values which are used in the rest of this study.

The derived values comprise four sulfuric acid concentrations / mixing ratios for the months June/July to October. Sulfuric acid is at very low mixing ratios (below 0.1 ppt) in July due to lack of sunlight and therefore absence of sulfuric acid

20   production during Antarctic winter. When sunlight returns in August, in the beginning the mixing ratio decreases further, as the gas phase sulfuric acid production takes some time to become larger than the losses (the chemical lifetime of $SO_2$ by OH and thereby the sulfuric acid production time is about a month at 20-30 km altitude; SPARC Report No.4 chapter 2.4.1). Starting in September, the sulfuric acid amount increases first slowly and then strongly from September to October by one order of magnitude reaching a maximum of ~1 ppt.

25   **2.2.3 Complementing the profiles with more trajectories**

To complement the profiles, more sulfuric acid mixing ratios were derived by simulating more trajectories that start at different altitudes. The trajectories are derived from the maximum trajectory by considering that the subsidence velocity decreases with decreasing altitude and that the polar vortex declines towards spring. Thereby, 23 trajectories starting at altitudes between 18 km and 40 km were simulated for this study. If the altitude was above 32 km, the initialization phase

30   was extended until the trajectory arrived below 32 km.
For all of these trajectories the procedure of deriving the corresponding sulfuric acid concentration was used as described above with two exceptions: First, at some points the temperature was a bit below 190 K (max. 4 K below) which is below SAWNUC's temperature range. Therefore these temperatures were fixed at 190 K. This introduces little uncertainty to these

[Figure]

values as the amount can be estimated with the sensitivity test concerning inter-annual temperature variations, presented below in Fig. 5d. Second, due to the design of the derivation process, the model was not able to reproduce the September CN amount of three trajectories in September at 27-28.5 km altitude. Therefore, the procedure was adjusted for these 3 data points, so that it does not search for the mean CN during this month but the CN are only required to match at the end of the

5    month. This is also expected to introduce only a small additional uncertainty to these three September values as the specific CN amount only has a small influence on the derived sulfuric acid amount (see below).

Deriving the sulfuric acid concentrations and mixing ratios for all trajectories resulted in values for nearly every combination of month and altitude. All these values then were combined to the four derived Antarctic sulfuric acid profiles.

**3 Results and discussion**

10   The derived Antarctic sulfuric acid profiles are shown in Fig. 2. The general shape of the derived sulfuric acid profiles is plausible. In Antarctic winter the values are well below one part per trillion because there is no sulfuric acid production when the sunlight is missing. Then, with the return of sunlight in August the values increase at high altitudes and in September at all altitudes. In October, they reach maximum values of above 10 ppt. The reason for the higher sulfuric acid mixing ratios at high altitudes is probably that fairly large amounts of source $SO_2$ are transported downward from the

15   mesosphere and that the actinic flux is high at these altitudes.

The figure also reveals that the temperature is the most important controlling variable for the sulfuric acid as the shapes of both profiles are very similar. This is because temperature and sulfuric acid concentration mainly determine the nucleation rate. A change in the sulfuric acid concentration by one order of magnitude leads to a change in the nucleation rate by also order(s) of magnitude. At a given temperature, there is a small sulfuric acid window where particle concentrations of 10 to

20   100 cm$^{-3}$ magnitude can exist. The nucleation rate has to be in this small window, otherwise there would be by far too many, or, too few CN. This small window is determined by the temperature. Therefore, whether 20 or 50 particles are present has only little influence on the derived sulfuric acid as this only changes the sulfuric acid inside this small window. So from the two main input parameters of the derivation process, the temperature controls the derived sulfuric acid's order of magnitude and the exact CN amount decides about the decimal places. This is why the derived profiles look very similar to the

25   temperature profiles and the CN layer's influence cannot be seen that clearly, however it is present indirectly as the magnitude of the CN defines the sulfuric acid window.

The derived sulfuric acid mixing ratios for October at altitudes above 25 km are very high. In fact the model calculations yield a particle distribution with a total sulfur mass that is higher than the total stratospheric sulfur at these altitudes. A detailed analysis reveals that at these high sulfuric acid mixing ratios the particles grow above 300 nm. However, as the

30   measurements only counted particles below 300 nm, here the derivation mechanism fails. The reason for this is that at these high temperatures a water vapor mixing ratio of about 5 ppm leads to very low relative humidities and therefore very high particle evaporation rates. To compensate these high evaporation rates the derivation mechanism predicts these very high

sulfuric acid concentrations to reproduce the observed particle amount. Based on this analysis we conclude that it seems unlikely that the particles above 25 km in October (and maybe also at the highest altitudes in September) are volatile, pure sulfuric acid water particles as evaporation rates are too high. The measurements of Campbell and Deshler (2014) show that there are some non-volatile particles present especially at high altitudes towards spring and also Curtius et al. (2005) observed non-volatile particles in the Arctic lower stratosphere. Therefore, we assume that most of the October particles above 25 km are non-volatile and therefore cannot be simulated with the SAWNUC model. Nevertheless, we show the too high sulfuric acid values in Fig. 2 as dotted line for completeness but omit these values for the rest of the study.

The derived sulfuric acid profiles cannot be compared directly to data from in situ or remote sensing measurements of Antarctic stratospheric sulfuric acid as such data does not exist to our knowledge. However, northern mid-latitude balloon-borne measurements have been published (Arnold et al., 1981; Reiner and Arnold, 1997; Schlager and Arnold, 1987; Viggiano and Arnold, 1981). Mills et al. (2005) presented a summary of these measurements. A comparison of these measurements with our derived sulfuric acid concentrations shows good agreement (Fig. 3). The concentrations range from $10^4$ cm$^{-3}$ to above $10^7$ cm$^{-3}$ and they have a similar shape: low values at low altitudes with an increase to high values at high altitudes. Due to the different locations (43°N vs. 78°S) the altitudes cannot be compared directly as the tropopause is expected to be at lower altitudes above Antarctica. Therefore, our derived profiles should be shifted upwards for comparison which improves the agreement. Note that also the adiabatic expansion has to be considered when shifting the profiles upwards. The derived October profile has an uncertainty towards lower values (see below) which also increases the agreement with the measurements. The derived lower concentrations in June/July at high altitudes compared to mid-latitudes are an expected result of the spare sunlight during polar night. However, the low June/July concentrations are of the same order of magnitude as inferred Arctic sulfuric acid concentrations presented by Krieger and Arnold (1994) (Fig. 3). In summary, our derived profiles are generally in agreement with stratospheric sulfuric acid measurements from other latitudes.

**3.1 Ion-induced nucleation**

To study the role of ion-induced nucleation during the formation of the CN layer, the sulfuric acid profiles were derived again, but without simulating ions. The comparison is shown in Fig. 4. In some areas the removal of ions has nearly no effect on the derived profiles, however, in other areas the sulfuric acid mixing ratios increase by almost an order of magnitude. The regions that are most affected are the ones with higher temperatures. At low temperatures the neutral clusters are stable enough so that including an ion to the cluster does not increase its stability against evaporation in an amount that would change the nucleation rate. Therefore, even if ions are present and ion-induced nucleation occurs, it is not more efficient than neutral nucleation. At higher temperatures  the neutral clusters are not as stable  and including an ion to the clusters  Thus to create the same amount of CN when no ions are present, more sulfuric acid is required than if ions were present simultaneously. In conclusion, in the areas of lower temperatures (which includes the formation area of the CN layer) the ions do not significantly influence the nucleation rate. However, in the areas of higher temperatures the CN are almost exclusively produced by ion-induced nucleation.

[Figure]

[Figure]

**3.2 Sensitivity studies**

To estimate the uncertainties of the derived sulfuric acid profiles, sensitivity tests were performed. Besides the 1-month time step for the derivation periods (already discussed in Sect. 2.2.2), significant uncertainties are mainly introduced by three factors: a) the uncertainty of the CN measurement cutoff size, b) the uncertainty of the climatological preexisting particle

5    surface area, and c) uncertainties of the thermochemical data, the condensation and coagulation rates used in the SAWNUC model. The Antarctic stratospheric sulfuric acid is also expected to vary from year to year (e.g. depending on the strength of the diabatic descent of mesospheric air masses within the polar vortex).

**3.2.1 CN measurement cutoff size**

The first sensitivity test investigates the influence of the CN measurement cutoff size on the derived profiles. Campbell and

10    Deshler (2014) reported that their CN counters' efficiencies were at around 75% for particles with 3 nm radius (6 nm diameter). So the 50% cutoff size should be below 6 nm diameter at the ground. However, they also reported that the cutoff could increase to 20 nm diameter at conditions of 20 km altitude. This, unfortunately, is a source of uncertainty for the measured CN concentrations.

For the derived profiles it was assumed that the measured CN are > 5 nm in diameter. For this sensitivity test the profiles

15    were derived assuming a cutoff of 10 nm diameter. A higher cutoff means that the particles have to grow larger before they are counted, thus the required sulfuric acid concentrations have to increase in the nucleation areas. In October during the decay of the CN layer, the mixing ratios have to decrease though, because even if no nucleation occurs there still are stable, growing particles left that are smaller than the cutoff size. Therefore, the derived sulfuric acid mixing ratios would be smaller to produce less new CN and to keep the growth rates small enough so that some of these CN stay below the cutoff

20    size. The result of this sensitivity test is shown in Fig. 5a and the predictions are confirmed. The July to September mixing ratios are higher and in October inside the CN layer the mixing ratios are lower. Therefore, the 50% cutoff uncertainty of the CN measurements has an impact on the derived profiles, especially on the October profile.

**3.2.2 Preexisting particles surface area**

The second sensitivity test investigates the influence of the total surface area of the preexisting particles. For the derived

25    profiles a surface area of 0.2 $\mu m^2$ $cm^{-3}$ at 30 km and 215.15 K was chosen, according to Campbell et al. (2014) reporting 0.3 $\mu m^2$ $cm^{-3}$ for 20-30 km altitude in early August 2010, and converted to the temperature/pressure conditions according to the ideal gas law. For this sensitivity test the surface area was increased to 0.5 $\mu m^2$ $cm^{-3}$ at 30 km and 215.15 K. The results show a significant influence on derived sulfuric acid mixing ratios below 1 ppt (Fig. 5b). In this area, mixing ratios increase by about a factor of 2 as at these conditions the losses to preexisting particles are in competition with the nucleation (Ehrhart

30    and Curtius, 2013). Therefore, if the chosen surface area of 0.2 $\mu m^2$ $cm^{-3}$ is not representative for all years, the derived profiles will have to be shifted either to higher or lower values according to this sensitivity test.

[Figure]

[Figure]

**3.2.3 Model uncertainties**

The third sensitivity test investigates the influence of the model uncertainties on the derived profiles. First, the uncertainty of the measured stabilities of the negatively charged clusters was tested according to Lovejoy et al. (2004) by adding 0.5 kcal to all changes in Gibbs free energy of negatively charged clusters. Second, all coagulation and condensation rates were reduced

5 by 20%. And third, the updated neutral sulfuric acid dimer thermodynamic stabilities presented by Kürten et al. (2015) were used for the calculations. They suggest a higher relative humidity dependence of the equilibrium constant of uncharged clusters which leads to higher dimer evaporation rates at low relative humidities.

The result of this combined sensitivity test is shown in Fig. 5c. An examination of the individual influences (not shown here) leads to the following conclusions: The changes of the thermodynamic values for the charged clusters influence the derived

10 mixing ratios in the area where ion-induced nucleation occurs. The changes due to the varied coagulation coefficients introduce a little shift to all values. The updated dimer stabilities with relative humidity dependence have a significant influence on the derived sulfuric acid mixing ratios in the regions of low relative humidity (higher temperature), but only if neutral binary nucleation dominates there.

**3.2.4 Inter-annual variations and other tests**

15 The last test estimates how the derived profiles may vary from year to year as this study used measurements averaged over 30 years. To estimate the inter-annual variations, all measured CN concentrations were increased by 60% and all temperatures were increased by 5 K. Both changes should lead to higher sulfuric acid values which is confirmed by the results (Fig. 5d). As the sulfuric acid profiles are mainly controlled by the temperature, the increase is mainly due to the temperature increase. The inter-annual variations are significant but they do not change the order of magnitude of the

20 profiles.

Additional sensitivity studies (not shown here) showed that the exact amount of ions or water molecules has only a very small influence on the derived profiles most probably because there are enough ions present and bigger changes than the stratospheric water vapor uncertainty of a few parts per million would be necessary to significantly change the derived profiles. Similarly, the CN and temperature measurement uncertainties (besides the cutoff uncertainty) have only a very little

25 influence on the derived profiles as the temperature uncertainty is very low. Adding hourly temperature variations generated according to a Gaussian distribution with mean value of 0 K and a standard deviation of 2 K to the 5 day maximum trajectory simulation had only a small influence on the derived sulfuric acid amounts.

**3.2.5 Summary and discussion**

Uncertainties are introduced by the model design of monthly derivation periods as the monthly values are an overestimation

30 in June/July to September and an underestimation in October. However, the cutoff measurement uncertainty leads to higher sulfuric acid values in June/July to September and lower values in October, so these two uncertainties are partly

compensating each other. For the October profile however, the uncertainty towards lower mixing ratios introduced by the uncertainty of the CN counter cutoff is bigger. At low sulfuric acid mixing ratios (<1ppt), the derived profiles have an uncertainty due to preexisting particles. The model uncertainties introduce uncertainties to the profiles towards higher or lower values at the areas of ion-induced nucleation and towards higher values at low relative humidities when neutral

5    nucleation dominates. The sulfuric acid profiles are expected to vary from year to year.

Therefore, the October profile should have the highest uncertainty, followed by the June/July profile. The August and September profiles should be the most accurate, which is during the formation of the CN layer.

**3.3 Sulfuric acid production (estimate)**

10    To estimate altitude profiles of the sulfuric acid production rates the modeling process had to be extended. As described, the sulfuric acid profiles were derived by searching for a constant sulfuric acid concentration that produces the measured CN amount. The extended approach was to assume a constant sulfuric acid production rate instead of a constant sulfuric acid concentration, and to simulate the sulfuric acid molecule concentration. The production was simulated by continually adding molecules throughout the modelled time periods. This added amount describes the net production ($H_2SO_4$ production from

15    $SO_2$ minus $H_2SO_4$ photolysis to $SO_2$, and potential other production processes). It does not contain the sulfuric acid that evaporates from freshly forming CN as this is now explicitly simulated by the model. However, the preexisting particles are not assumed to evaporate in the model, therefore this production term could also represent any sulfuric acid that evaporates from preexisting particles.

With this approach, sulfuric acid production profiles were derived with monthly production rates (Fig. 6). Their shapes look

20    much like the derived sulfuric acid profiles, which is expected. They also range over 3-4 orders of magnitude from 0.5 to about 500 $cm^{-3}\ s^{-1}$.

The June/July production rates need further investigation. They indicate that some (small) sulfuric acid production should occur even though there is nearly no sunlight during winter. This sulfuric acid could be produced by processes that do not require sunlight. Krieger and Arnold (1994) presented Arctic negative ion composition measurements and inferred gaseous

25    sulfuric acid concentrations. They found strong evidence for an OH production process that does not require sunlight as they also observed sulfuric acid production during Arctic winter. They proposed OH production via ambient ions as additional sulfuric acid source and calculated a sulfuric acid production rate by ions of approximately 0.2-0.9 $cm^{-3}\ s^{-1}$. Compared to our derived June/July production rates, these are lower (max. by one order of magnitude). This difference could be due to different ambient conditions in the Arctic compared to the Antarctic stratosphere or the OH production by ions explains our

30    derived production rates only partly. Our derived sulfuric acid concentrations for June/July could be too high as the production and  nature of the CN measured in June/July  (there are no measurements in Antarctic fall). They could be more stable if they were not produced only by binary nucleation (but e.g. with meteoritic dust). Or they could be the result of an Antarctic fall nucleation event that was predicted by the global modeling of Campbell et al. (2014).

[Figure]

[Figure]

Estimates of Antarctic sulfuric acid production profiles are presented in Fig. 6. They do indicate some sulfuric acid production during Antarctic winter. However, they should only be considered as first estimates as they are one additional step away from the measurements. Processes like preexisting particle evaporation, OH production from cosmic rays, and CN production by other processes could influence the derived production profiles.

**4 Comparison with global modeling of Campbell et al. (2014)**

Our derived sulfuric acid profiles can now be compared to the global modeling of Campbell et al. (2014) to discuss the origin of the CN layer's altitude shift in their model. For this, we compare our derived sulfuric acid profiles to the global model simulation presented by Campbell et al. (2014) in Fig. 2. We compare the values between early July and late October. Our sulfuric acid volume mixing ratios were derived for the altitude range of 18 km to 32 km. In the global model, we find corresponding sulfuric acid values roughly between 25 km and 38 km. In this range the high altitude July value and the low altitude October value match. Also high sulfuric acid mixing ratios at high October altitudes are found in the global model. However, the global model simulates much lower sulfuric acid concentrations in July at low altitudes. They are the result of missing sunlight and therefore no sulfuric acid production. As discussed above, SAWNUC seems to predict sulfuric acid production in this region. However, as discussed, this production could need an additional process like OH production by ions that can produce sulfuric acid without sunlight. As this additional process is not simulated by the global model, it predicts lower sulfuric acid mixing ratios in July at low altitudes. Therefore, it could be possible that the global model's minimum sulfuric acid mixing ratio of about $1 \cdot 10^{-20}$ is a strong underestimation.

Nevertheless, the general orders of magnitude of our derived sulfuric acid profiles are mostly found in the global model output, though with a 7 km altitude shift to higher altitudes. Thus, the global model has an altitude shift in CN, temperature, and sulfuric acid. This is further strengthened by comparing the modelled $SO_2$ with satellite observations by Höpfner et al. (2013) which suggests an even stronger altitude shift. Thus our results support the suggestion by Campbell et al. (2014) that the altitude shift in their modelled CN layer seems to be a result of altitude shifts in the controlling variables of the nucleation.

Note however, that this result could increase the confidence in the simulated global extent of the CN layer. If only the temperature had an altitude shift and not the sulfuric acid, at each altitude there would be very different temperature / sulfuric acid combinations, resulting in different nucleation rates compared to the real stratosphere. However, as the sulfuric acid also has an altitude shift, the nucleation rates should be closer to reality and therefore also the simulation of the global extent of the CN layer.

[Figure]

**5 Summary**

Balloon-born measurements of stratospheric CN above McMurdo Station, Antarctica, reveal the presence of a CN layer of freshly formed particles at 21-27 km altitude in August to October. Campbell et al. (2014) showed the global extent of this CN layer with a global model that reproduced the production of the CN layer by binary sulfuric acid water nucleation.

5    However, in their model the CN layer was located at too high altitudes. Unfortunately, no Antarctic stratospheric sulfuric acid measurements exist for comparison. Therefore, Campbell et al. (2014) derived sulfuric acid concentrations from the measured CN and temperatures. However, they did not use a microscopic nucleation model that includes processes such as coagulation, ion-induced nucleation, and losses to preexisting particles. Therefore, the goal of the present study was to use the nucleation model SAWNUC as a box model to derive Antarctic stratospheric sulfuric acid profiles based on their

10    measurements and to investigate the nucleation process.

The sulfuric acid profiles were derived for the altitudes of 18 - 32 km by simulating air parcel trajectories that descend inside the polar vortex. For each trajectory, monthly sulfuric acid values were derived by searching for the sulfuric acid amount that reproduces the observed CN at the given ambient conditions. The derived sulfuric acid concentrations (volume mixing ratios) are of the order of $10^4$ cm$^{-3}$ ($10^{-14}$) in July. In the following months the concentrations increase to about $10^7$ cm$^{-3}$ ($10^{-11}$)

15    in October. They depend strongly on the temperature because at a given temperature the nucleation rate varies with the sulfuric acid amount, leaving only a small sulfuric acid window to reproduce the observed magnitude of CN. The derived sulfuric acid profiles compare well with measured mid-latitude profiles and also with inferred Arctic sulfuric acid concentrations.

Ion-induced nucleation occurs, however, at low temperatures it has no significant influence on the nucleation rates as the

20    neutral clusters are already stable enough, so that an additional charge does not significantly increase their stabilities. However, at higher temperatures the neutral clusters are not as stable anymore and ion-induced nucleation becomes the dominant nucleation mechanism.

Uncertainties of the derived profiles are caused by uncertainties of the instrumental cutoff diameter of the CN counter used for the measurements, the uncertainties of the preexisting particles surface area, and model uncertainties. The October profile

25    has an uncertainty towards lower values mainly due to the uncertainty of the CN measurement cutoff. Sulfuric acid mixing ratios below 1 ppt  preexisting particles surface area. The profiles are expected to vary from year to year.

Estimates of sulfuric acid production rates range from 0.5 to about 500 molecules cm$^{-3}$ s$^{-1}$. Sulfuric acid production during Antarctic winter seems to be necessary to explain the measurements. This would require a second production process that

30    does not require sunlight (e.g. OH production by ambient ions).  production rates should only be considered  as  variety of processes that were not simulated with this model.

[Figure]

Finally, a comparison of the derived sulfuric acid profiles with the global modeling of Campbell et al. (2014) strengthens the assumption that the global model represents the processes in general, but at too high altitudes as also the sulfuric acid seems to be simulated at too high altitudes.

**Acknowledgements**

5  We thank Edward R. Lovejoy, Karl D. Froyd, Jan Kazil, and Sebastian Ehrhart for providing the SAWNUC code and Andreas Engel for useful discussion.

**References**

Arnold, F., Fabian, R., and Joos, W. (1981). Measurements of the height variation of sulfuric acid vapor concentrations in the stratosphere. Geophysical Research Letters, 8(3), 293-296.

10  Ball, S. M., Hanson, D. R., Eisele, F. L., and McMurry, P. H. (1999). Laboratory studies of particle nucleation: Initial results for H2SO4, H2O, and NH3 vapors. Journal of Geophysical Research: Atmospheres (1984–2012), 104(D19), 23709-23718.

Campbell, P., and Deshler, T. (2014). Condensation nuclei measurements in the midlatitude (1982–2012) and Antarctic (1986–2010) stratosphere between 20 and 35 km. Journal of Geophysical Research: Atmospheres, 119(1), 137-152.

Campbell, P., Mills, M., and Deshler, T. (2014). The global extent of the mid stratospheric CN layer: A three-dimensional

15  modeling study. Journal of Geophysical Research: Atmospheres, 119(2), 1015-1030.

Carslaw, K. S., Clegg, S. L., and Brimblecombe, P. (1995). A thermodynamic model of the system HCl-HNO3-H2SO4-H2O, including solubilities of HBr, from< 200 to 328 K. The Journal of Physical Chemistry, 99(29), 11557-11574.

Chan, T. W., and Mozurkewich, M. (2001). Measurement of the coagulation rate constant for sulfuric acid particles as a function of particle size using tandem differential mobility analysis. Journal of aerosol science, 32(3), 321-339.

20  Curtius, J., Weigel, R., Vössing, H. J., Wernli, H., Werner, A., Volk, C. M., ... and Schlager, H. (2005). Observations of meteoric material and implications for aerosol nucleation in the winter Arctic lower stratosphere derived from in situ particle measurements. Atmospheric chemistry and physics, 5(11), 3053-3069.

English, J. M., Toon, O. B., Mills, M. J., and Yu, F. (2011). Microphysical simulations of new particle formation in the upper troposphere and lower stratosphere. Atmos. Chem. Phys, 11(17), 9303-9322.

25  Ehrhart, S., and Curtius, J. (2013). Influence of aerosol lifetime on the interpretation of nucleation experiments with respect to the first nucleation theorem. Atmospheric Chemistry and Physics, 13(22), 11465-11471.

Froyd, K. D., and Lovejoy, E. R. (2003a). Experimental thermodynamics of cluster ions composed of H2SO4 and H2O. 2. Measurements and ab initio structures of negative ions. The Journal of Physical Chemistry A, 107(46), 9812-9824.

Froyd, K. D., and Lovejoy, E. R. (2003b). Experimental thermodynamics of cluster ions composed of H2SO4 and H2O. 1.

30  Positive ions. The Journal of Physical Chemistry A, 107(46), 9800-9811.

[Figure]

[Figure]

Fuchs, N. A. (1964). The mechanisms of aerosols. Pergamon. New York.

Hamill, P., Toon, O. B., and Turco, R. P. (1990). Aerosol nucleation in the winter Arctic and Antarctic stratospheres. Geophysical Research Letters, 17(4), 417-420.

Hanson, D. R., and Lovejoy, E. R. (2006). Measurement of the thermodynamics of the hydrated dimer and trimer of sulfuric acid. The Journal of Physical Chemistry A, 110(31), 9525-9528.

Hofmann, D. J. (1990). Measurement of the condensation nuclei profile to 31 km in the Arctic in January 1989 and comparisions with Antarctic measurements. Geophysical Research Letters, 17(4), 357-360.

Hofmann, D. J., and Rosen, J. M. (1985). Antarctic observations of stratospheric aerosol and high altitude condensation nuclei following the El Chichon eruption. Geophysical Research Letters, 12(1), 13-16.

Höpfner, M., Glatthor, N., Grabowski, U., Kellmann, S., Kiefer, M., Linden, A., ... and Boone, C. D. (2013). Sulfur dioxide ($SO_2$) as observed by MIPAS/Envisat: temporal development and spatial distribution at 15–45 km altitude. Atmospheric Chemistry and Physics, 13(20), 10405-10423.

Hurrell, J. W., Holland, M. M., Gent, P. R., Ghan, S., Kay, J. E., Kushner, P. J., ... and Marshall, S. (2013). The community earth system model: A framework for collaborative research. Bulletin of the American Meteorological Society, 94(9), 1339-1360.

Kazil, J., and Lovejoy, E. R. (2007). A semi-analytical method for calculating rates of new sulfate aerosol formation from the gas phase. Atmospheric Chemistry and Physics, 7(13), 3447-3459.

Kirkby, J., Curtius, J., Almeida, J., Dunne, E., Duplissy, J., Ehrhart, S., ... and Stratmann, F. (2011). Role of sulphuric acid, ammonia and galactic cosmic rays in atmospheric aerosol nucleation. Nature, 476(7361), 429-433.

Krieger, A., and Arnold, F. (1994). First composition measurements of stratospheric negative ions and inferred gaseous sulfuric acid in the winter Arctic vortex: implications for aerosols and hydroxyl radical formation. Geophysical research letters, 21(13), 1259-1262.

Kürten, A., Münch, S., Rondo, L., Bianchi, F., Duplissy, J., Jokinen, T., ... and Curtius, J. (2015). Thermodynamics of the formation of sulfuric acid dimers in the binary ($H_2SO_4$-$H_2O$) and ternary ($H_2SO_4$-$H_2O$-$NH_3$) system. Atmospheric Chemistry and Physics Discussions, 15(10), 13957-14006.

Lovejoy, E. R., Curtius, J., and Froyd, K. D. (2004). Atmospheric ion-induced nucleation of sulfuric acid and water. Journal of Geophysical Research: Atmospheres (1984–2012), 109(D8).

Lovejoy, E. R., and Curtius, J. (2001). Cluster ion thermal decomposition (II): Master equation modeling in the low-pressure limit and fall-off regions. Bond energies for $HSO_4^-(H_2SO_4)_x(HNO_3)_y$. The Journal of Physical Chemistry A, 105(48), 10874-10883.

Mills, M. J., Toon, O. B., and Solomon, S. (1999). A 2D microphysical model of the polar stratospheric CN layer. Geophysical research letters, 26(8), 1133-1136.

[Figure]

[Figure]

Mills, M. J., Toon, O. B., Vaida, V., Hintze, P. E., Kjaergaard, H. G., Schofield, D. P., and Robinson, T. W. (2005). Photolysis of sulfuric acid vapor by visible light as a source of the polar stratospheric CN layer. Journal of Geophysical Research: Atmospheres (1984–2012), 110(D8).

Reiner, T., and Arnold, F. (1997). Stratospheric SO3: Upper limits inferred from ion composition measurements-Implications for H2SO4 and aerosol formation. Geophysical research letters, 24(14), 1751-1754.

Rosen, J. M., and Hofmann, D. J. (1983). Unusual behavior in the condensation nuclei concentration at 30 km. Journal of Geophysical Research: Oceans (1978–2012), 88(C6), 3725-3731.

Schlager, H., and Arnold, F. (1987). Balloon-borne composition measurements of stratospheric negative ions and inferred sulfuric acid vapor abundances during the MAP/GLOBUS 1983 campaign. Planetary and space science, 35(5), 693-701.

Usoskin, I. G., Bazilevskaya, G. A., and Kovaltsov, G. A. (2011). Solar modulation parameter for cosmic rays since 1936 reconstructed from ground-based neutron monitors and ionization chambers. Journal of Geophysical Research: Space Physics (1978–2012), 116(A2).

Vaida, V., Kjaergaard, H. G., Hintze, P. E., and Donaldson, D. J. (2003). Photolysis of sulfuric acid vapor by visible solar radiation. Science, 299(5612), 1566-1568.

Viggiano, A. A., and Arnold, F. (1981). Extended sulfuric acid vapor concentration measurements in the stratosphere. Geophysical Research Letters, 8(6), 583-586.

Yu, F., and Turco, R. P. (1998). The formation and evolution of aerosols in stratospheric aircraft plumes: Numerical simulations and comparisons with observations. Journal of Geophysical Research: Atmospheres (1984–2012), 103(D20), 25915-25934.

Zhao, J., Toon, O. B., and Turco, R. P. (1995). Origin of condensation nuclei in the springtime polar stratosphere. Journal of Geophysical Research: Atmospheres (1984–2012), 100(D3), 5215-5227.

[Figure]

[Figure]

[Figure]

**Figure 1: Derivation of the "maximum trajectory" with model time steps of 1 month (a), and 5 days (b). Temperature (red), sulfuric acid (blue), and the other ambient conditions are changed at the beginning of each model time step (1 month or 5 days) and kept constant during the model time step while the CN (> 5 nm) concentrations (black) are simulated. The gaseous sulfuric**
5 **acid was varied until the number concentration of CN particles matched the observations. At the beginning of each time step the CN concentration shifts abruptly as the changed ambient conditions lead to an adiabatic compression. The higher time resolution in b) breaks these big shifts into smaller more frequent shifts.**

[Figure]

**Figure 2: The derived Antarctic sulfuric acid profiles corresponding to CN and temperature measurements above McMurdo,**
10 **Antarctica (78°S) presented as concentration (a), and volume mixing ratio (b). The dotted line shows the unrealistic high October values (see discussion). The profiles are strongly correlated with temperature. Mixing ratios below 1e-12 also depend on preexisting particles surface area.**

[Figure]

[Figure]

[Figure]

**Figure 3:** Comparison of the derived Antarctic sulfuric acid profiles (solid lines) with mid latitude measurements and modeling of Arnold et al. (1981), Reiner and Arnold (1997), Schlager and Arnold (1987), Viggiano and Arnold (1981), and Mills et al. (2005) (shaded area) and inferred Arctic sulfuric acid from measurements presented by Krieger and Arnold (1994). The derived profiles have to be compared to mid-latitude concentrations at higher altitudes as the tropopause is lower over Antarctica.

[Figure]

**Figure 4:** Comparison of the sulfuric acid profiles derived including ion-induced nucleation (solid lines, reference profiles) and without simulating ions (dashed lines). In the area of lower temperatures (earlier and lower) the neutral clusters are stable enough so that the ions do not significantly change the nucleation rates. In the areas of higher temperatures (later and higher) the neutral clusters are not as stable anymore and more sulfuric acid is needed to reproduce the observed CN when ion-induced nucleation is not simulated.

[Figure]

**Figure 5: Sensitivity studies varying (a) CN counter cutoff size, (b) preexisting particles surface area, (c) model parameters, and (d) inter-annual temperature and CN amounts, to estimate the uncertainties of the derived reference sulfuric acid profiles. The solid lines always show the derived reference profiles as presented in Fig. 2. The dashed lines show the profiles for model runs with the input values changed according to the sensitivity test. The October profile should have the highest uncertainty, followed by the June/July profile. The August and September profiles should be the most accurate. Detailed description of the varied parameters and discussion can be found in the text.**

[Figure]

[Figure]

[Figure]

[Figure]

**Figure 6: Estimates of sulfuric acid production rates presented per volume and second (a) and as mixing ratio per second (b). Some sulfuric acid production during the Antarctic winter is predicted. However, the modeling results should only be considered as first estimates as they are one additional step away from the measurements. Processes like preexisting particle evaporation, dark OH production (e.g. from cosmic rays), and CN production by other processes could change or could be included in these production profiles.**

---

## Referee Comment (RC2) · Anonymous Referee #2 · 29 Aug 2016

This paper follows some work by Campbell and Deshler [2014] to derive a sulfuric acid mixing ratio profile from observations of temperature and rate of nucleation of new particles, assuming binary homogeneous nucleation. The present work employs a microphysical box model (SAWNUC) to derive the sulfuric acid mixing ratio necessary to reproduce the CN observations of Campbell and Deshler [2014]. This is a useful exercise and should lead to a publishable paper. Unfortunately the present paper needs a lot of work to bring it to scientific maturity and there are some major issues that need to be clarified.

The primary issue is the authors' seeming assumption that underlying the presence of any stratospheric sulfuric acid particle concentration is a necessary sulfuric acid

concentration. Yes there was a sulfuric acid profile at one time, but once the sulfuric acid condenses its concentration decreases to vanishing amounts. Because there are CN particles existent during polar winter does not mean that they were nucleated then and there. Stratospheric CN are ubiquitous in concentrations near 10 cm-3. Thus as the polar vortex forms the CN particles present at the beginning will persist in the vortex, their sedimentation quite negligible. The observations of Campbell and Deshler [2014] bear this out for much of the CN profiles they present. Here, however, the authors derive a sulfuric acid mixing ratio profile in times of the year and altitude regions where no new particle formation is observed. They then speculate on what could be causing the oxidation of so2 to obtain the sulfuric acid. Isn't this speculation on possibly an imaginary profile?

I understand when sulfuric acid particles are present there will be a residual sulfuric acid concentration; however, given that the vapor pressure of 75% by weight sulfuric acid is exceedingly low, this concentration will be quite small. Is this what is calculated by SAWNUC in the absence of new particle formation? The authors need to make clear at the outset how the observations of new particle formation are used in deriving the sulfuric acid profiles and how such profiles are obtained in the absence of new particle formation. In the latter case can there be anything more than an upper limit?

The other major issue is the writing is not up to the standards expected for a scientific paper. The writing is imprecise, many details are missing, while irrelevant comments creep in. The authors should consider having some English help prior to the next submission and not keep the reader working out what is intended to be said.

Here are more specific details.

1.13. Reproduces what observations? Be specific here. Reproduces the new particles that are observed or . . .?

1.14 As will be mentioned further, below, it is not clear how a sulfuric acid profile is obtained in July. At most this should be an upper limit.

1.16 "the observed magnitude of CN..." What does this mean? Was nucleation of new particles observed at higher temperatures?

1.18-19. The authors have not convinced me of the need to include a sulfuric acid production rate, which is then one further step removed from the observations. Thus I see no need to go further and try and explain a winter-time production mechanism, when it has not been established that one is required.

1.20. Why is there no comparison with the sulfuric acid profile derived by Campbell and Deshler [2014]?

1.23-26. This is a terrible first sentence. In fact it is not even a sentence. The clause set off with commas is a complete sentence and should stand on its own. But it is still a poor first sentence. In fact the definition of condensation nuclei (CN) is not correct. Condensation nuclei are observed by forcing small particles to grow to optically detectable sizes using a condensing fluid and a supersaturation chamber. There are a number of references available on the technique and perhaps one should be included. The size ranges vary somewhat, but typically the CN concentration is used to define the total aerosol concentration larger than some nominal size, say 10 nm. CN measurements also include particles > 300 nm, but their concentration is inconsequential compared to the total aerosol population.

1.27. Not all CN are formed by nucleation. It appears a significant fraction of CN are non-volatile suggesting that this fraction of CN particles appears due to the condensation of sulfuric acid on pre-existing solid cores [Borrmann et al., 2010; Campbell and Deshler, 2014].

2.12. Where does the h2so4 in the mesosphere come from to be available for photolysis?

2.15 "performed continuously for 24 years" What does this mean every month, every day? I understood the observations are only available for a few months each year.

2.17-20. Very awkward sentence. Try. They present monthly averaged CN concentration and temperature profiles which capture the unperturbed CN, with concentrations around 10-20 cm-3 in June/July as well as the development of a layer of particles at 20-25 km, with concentrations increasing to 100 cm-3 from August until October during sunrise and warming.

2.26-28. Campbell et al. [2014] did not invert the nucleation equation, they ran a three dimensional chemistry model which produces its own h2so4.

2.30. Who are they?

3.30. Delete the sentence, "This section describes the derivation of the Antarctic stratospheric sulfuric acid profiles" and add the information to section title, e.g. Deriving the Antarctic stratospheric sulfuric acid profiles.

4.14. "3e5 ion pairs per gram of air and second" Awkward. Use standard SI units and symbols to describe a rate.

4.16-18. There should be some references for the water vapor profile and the diabatic descent rate?

4.19-24. 100 nm is at the upper limit of possible pre-existing particles. What concentrations are used for these particles? Since the surface area is chosen based on reasonable sources why is the particle size important? What does it mean that the surface area was converted . . . according to the ideal gas equation? Do you mean the surface area density was adjusted to different temperatures and pressures assuming a constant mixing ratio?

4.28-31. Is there really a trajectory of an air parcel here, or do the authors just mean variations in the various parameters required by the SAWNUC model? The language is confusing. If the latter then I think the more appropriate words would be box model simulation rather than air parcel trajectory. If the former then where does the air parcel start, what altitude, location, dimensions? What does it mean to "connect" to the

monthly maximum of the measured CN? Do the authors mean the trajectory which reproduces the monthly maximum CN concentration? The next sentence is equally awkward, "It is assumed that this maximum of the particle concentration resides in a single air parcel that descends inside the polar vortex." The maximum of the particle concentration doesn't reside in the air parcel until the nucleation has occurred. What the authors are trying to suggest, as far as I can understand, is the optimum trajectory to reproduce the observed maximum in CN concentration. The phrase "maximum trajectory" does not convey this idea and I would recommend changing this language to something more representative of what is being simulated. Something like optimum trajectory to reproduce the observed maximum in CN concentration. This maximum in CN concentration is height dependent. Is this height dependence consistent with the diabatic descent in the model?

5.1. I believe the authors mean the sulfuric acid concentrations were varied on a monthly basis? Search implies that these concentrations exist on some data base or?

5.4-6. Awkward and unnecessary detail. The authors are merely iterating the model until they achieve a match with the measured CN by varying the sulfuric acid concentrations. If so say this. It is not the sulfuric acid concentration that corresponds to the monthly CN concentrations, it is the simulated CN concentrations based on the sulfuric acid concentrations. The writing here and elsewhere is awkward and requires the reader to do a lot of guessing as to what is meant.

5.14-18, Fig. 1b. The model is initialized with a CN concentration of zero. Is this realistic? There will be some residual particles, CN, persisting in the air as the polar vortex forms. It seems more plausible to assume these residual particles appear as the background of about 10 cm-3 shown in Campbell and Deshler [2014] outside of the CN layers than to assume a concentration of zero. If the model is allowed to initialize to 110% of the June/July CN maximum, then there would not be any further new particle formation in the June/July period. So no additional sulfuric acid is required. This point

needs to be clarified.

5.18-19. Since the model is already at 110% of the CN maximum, why is further searching required? Is sulfuric acid required to maintain the pre-existing particles? Are these particles losing sulfuric acid? I understood these particles change size due to fluctuations in water based on temperature but not on the loss of sulfuric acid.

It is not clear how the subsequently derived sulfuric acid profile for June/July is obtained. How does it differ from the sulfuric acid profile that is required to build up the CN from zero to 110% of June/July values? How does the work here then take advantage of the measurements which show a maximum of CN near 30 km above a possible background of around 10 cm-3? If there is no new particle formation in June July then the speculation on the source of a sulfuric acid profile is misplaced.

5.26. Please show the mean CN values on Fig. 1 a. It would also be instructive to include on both panels the observations which are being reproduced.

5.27-28. If there is a decrease from June/July – August as could be argued from Campbell and Deshler [2014] since the CN maximum concentration does not change in their Fig. 1, then it appears again that no new particles are formed so only an upper limit can be placed on the sulfuric acid concentration.

5.32. How is this month long time interval different than the previous ones? Why is the sulfuric acid concentration now strongly increased? In Campbell and Deshler [2014] there is almost no change in CN concentration between September and October in the CN layer, arguing again for little information on a sulfuric acid profile.

5.10-6.4 and Fig. 1a). Is all of this necessary to understand what is done, and do the authors really want to explain the subtleties of Fig. 1a? In the end more realistic time steps are used and profiles created. I don't really think the reader is benefitting any from the previous discussion and it raises many questions such as listed above.

6.17-24. This paragraph is confusing by suggesting that the sulfuric acid is dependent

on other processes in the model rather than just the amount needed to create the CN maximums. Here the authors claim that the sulfuric acid is dependent on sunlight, which we know it is, but I didn't think this was considered by the SAWNUC model.

6.21. What is meant by, "the chemical lifetime of SO2 by OH"?

6.23. Since the authors are now dealing with monthly averages, how can they make this claim, "Starting in September, the sulfuric acid amount increases first slowly and then strongly from September to October"? Monthly averages cannot supply this information.

Fig. 1b. What is the explanation for the fast rise in CN near the beginning of September, while the sulfuric acid only increases at the end of September? What is causing the CN formation if not the sulfuric acid? There is a disconnect here.

6.26. "To complement the profiles" What profiles? So far the authors have been calculating the conditions required to reproduce the maximum in observed CN concentrations. This would then only be a sulfuric acid concentration at one altitude in one month. Perhaps the authors intend to say they will now create profiles. If so the title of this section is misleading.

7.3 "in September" is redundant and should be deleted.

Section 2.2.3 and Fig. 2. This section and figure raise again my main question about this work. How is a sulfuric acid profile derived if there is no new particle formation? The CN layers in the CN profiles shown by Campbell and Deshler [2014] are limited in altitude to a fraction of the 18-40 km to be investigated here. Thus, for example in September above 27 km there is no evidence of new particle formation. So can anything more than an upper limit be placed on the sulfuric acid profile? Yyet Fig. 2 shows the September sulfuric acid profile to continue to increase above 27 km. How are profiles created to 40 km when there are no observations above 35 km?

7.10-15. Here again the authors are discussing results suggesting that the sulfuric

acid profile is derived from sources outside the matching of the SAWNUC model with observations of CN.

7.17 "the shapes of both profiles are very similar" The authors discuss the temperature profile as if it were shown in Fig. 2, but it is not. Thus the readers cannot follow this point. The temperature profile should be included in Fig. 2.

7.19-21. I don't understand. Do the authors mean there is a narrow sulfuric acid concentration range leading to the nucleation of particles in the 10-100 cm-3 range? There is no reason particles cannot exist at these concentration levels if they are already there. The nucleation rate is not in a small window. The nucleation rate is determined by the concentration and the temperature. The small window is determined by both temperature and sulfuric acid concentration. Far too many or too few CN for what? I thought the particle concentrations were used to derive the sulfuric acid profile given the temperature, so if there are more or less CN this just affects the sulfuric acid concentration.

7.22. "20 or 50 particles present" Are these particles already existing or nucleated particles? I assume the authors mean an observation of 20 – 50 new particles has a small influence on the derived sulfuric acid.

7.25-26. This is confusing. If the CN layer were not observed then there would be no attempt to derive the sulfuric acid profile. Above and below the CN layer there is not enough information to derive the sulfuric acid profile, only upper bounds, determined by the temperature, can be placed on it.

7.30. This statement is not correct CN instruments measure all particles above some nominal size, typically about 10 nm. In addition I believe Campbell and Deshler report observations explicitly of particles > 150 nn using a second instrument, and no enhancements of these particles were observed in the CN layers.

7.27-29. These statements should give the authors pause to consider if there is an

error somewhere rather than trying to explain these high sulfur amounts.

7.30-32. This argument is incorrect. The water vapor concentration is low throughout the stratosphere. The particles are not composed of water but a sulfuric acid water mixture and the temperature determines how much water is retained by the highly acidic particle. The water vapor concentration has nothing to do with it, only temperature. There are plenty of water molecules.

7.32. "To compensate there high . . ." For the reasons above I don't' understand this argument. Isn't the nucleation of particles dependent on the gathering of critical clusters of the acid not the water?

8.5-7. Why is this assumption made? Campbell and Deshler, Fig. 5 show directly that the non-volatile fraction above 25 km is similar to that below 20 km, about 60%. It is not at all clear how the authors derive a sulfuric acid profile when there is no obvious nucleation. Again there can only be an upper bound.

Fig. 3. Why isn't the profile derived by Campbell and Deshler [2014] included in this figure?

8.33 "However, in the areas of higher temperatures the CN are almost exclusively produced by ion-induced nucleation." This assumes that CN are nucleated, but there is really no evidence for this, thus it is questionable whether any CN are nucleated in these areas. Sentence should be rewritten to reflect that, if particles nucleated they would require ion-induced nucleation, but in fact there is no evidence that such particles are nucleated.

10.29-30. What is over/under estimated? Why is it clearly related to the months? I believe the authors mean the size cutoff measurement uncertainty.

11.7-8. This seems somewhat obvious as all the nucleation occurs in August, September. When there is no new particle formation this approach can only provide an upper bound on the sulfuric acid.

11.15-16. Sulfuric acid evaporating from freshly forming particles? How does this occur? My understanding is that the vapor pressure of sulfuric acid is very low, thus it condenses readily and then stays condensed. Same question about pre-existing particles. At the temperatures used in the modeling I would not believe that the pre-existing particles would do much more than adjust their water content to the temperature changes.

11.14-18. If I understand correctly the production rate is simulated by continually adding the necessary h2so4 molecules to simulate the observations without specifying where the molecules come from e.g. so2, or particles. Thus all this discussion about the possible sources is speculation without basis and should be eliminated.

11.22-25. I am again confused as to why there is production of h2so4 in June July when the CN particle concentration is not changing from the model initialization. Much of the rest of this paragraph is again speculation and should be strictly limited.

12.1-4. This is all redundant.

Overall the paper would not suffer if section 3.3 and figure 6 were removed.

12.13-15. It is not clear why SAWNUC is predicting a sulfuric acid mixing ratio at this time in this region.

Borrmann, S., Kunkel, D., Weigel, R., Minikin, A., Deshler, T., Wilson, J. C., Curtius, J., Volk, C. M., Homan, C. D., Ulanovsky, A., Ravegnani, F., Viciani, S., Shur, G. N., Belyaev, G. V., Law, K. S., and Cairo, F.: Aerosols in the tropical and subtropical UT/LS: in-situ measurements of submicron particle abundance and volatility, Atmos. Chem. Phys., 10, 5573-5592, doi:10.5194/acp-10-5573-2010, 2010.

---

## Author Response (AR1)

**Author's response to the discussion paper:**

**Nucleation modeling of the Antarctic stratospheric CN layer and derivation of sulfuric acid profiles**

Steffen Münch[1,*] and Joachim Curtius[1]

(Please notice the changed title)

**Answers to Referee No. 1**

We want to thank the reviewer for carefully reading our manuscript and for providing numerous helpful comments. The reviewers' comments are repeated in full below in black font, with our replies indicated after each comment in blue font. Text which has been changed in the manuscript is shown in red font.

**Major changes**

As both reviewers noted we used an incomplete definition of condensation nuclei (CN) in the text and also for our simulation. CN were defined in our model as just the smallest particles that are smaller than those counted by the optical particle counter (diameter $d_p < 0.3$ µm) instead of all particles that can be counted by a condensation nuclei counter. Usually this does not make a big difference as the smaller particles dominate in number. However, for our study, the two formulations unfortunately make a substantial difference.

We assumed that the CN presented in Campbell and Deshler (2014) were all smaller than 300 nm diameter (as this was the cutoff of their OPC). Therefore we simulated these CN as small particles and *additionally* simulated the preexisting particles surface area loss by the larger particles. This extra loss term reduced our simulated CN which made nucleation of new CN necessary in all months at all altitudes and not just during the formation of the CN layer. This is why we could derive profiles for all months; which was the major point of criticism by Reviewer 2.

We now corrected our method. As a result, we do not simulate the long warm up phase anymore, but start at the beginning of July and use the CN of Campbell and Deshler (2014) as initial particles for the first month. These CN are now the only particles in our simulation, we have no additional loss to preexisting particles anymore.

This changes our results a lot. We now do not need sulfuric acid anymore for reproducing the observed CN, except for the formation of the CN layer. This solves the major point of Reviewer 2.

We can not derive profiles of sulfuric acid for all months anymore, and therefore some of the main messages of the paper change. We focus more on the result that we can reproduce the observed CN just with coagulation and air volume compression except the CN layer formation. This is consistent with Campbell and Deshler (2014) and the general idea of the formation of the CN layer: low sulfuric acid during winter but then an increase after the onset of sunlight. However, we still derive profiles at which nucleation would occur and interpret these as upper limits (as suggested by Reviewer 2).

As the method and results changed, large parts of the paper were rewritten. We did this with the suggestions of the reviewers in mind and we also used this opportunity and changed the paper to active-writing-style. Here are the major changes:

Title: We changed the title of the paper to" Nucleation modeling of the Antarctic stratospheric CN layer and derivation of sulfuric acid profiles". This flip of the two parts reflects that the derived profiles are not our main/only result any more and this is also the order in which we present and discuss the results.

Abstract: The abstract is almost completely rewritten and describes the new results of the paper.

Introduction: We added more information on Antarctic stratospheric CN to the introduction as suggested by the reviewers. We removed the discussion of the too high CN layer altitude in the global model of Campbell et al. (2014) as we do not discuss this in the paper anymore (see below).

Methods: We rewrote and shortened this section. We shortened the SAWNUC description (Sect. 2.1) but added a short description of the most relevant processes. We removed the preexisting particle discussion from Sect. 2.2 as this is not necessary anymore. As reviewer 2 suggested the detailed discussion of the single parcel simulation is not necessary.

Results: We rewrote the Results section according to the new modelling results. We start with a simple reference case (Figure 1). We give additional information on the simulated descending air parcel paths (as both reviewers asked for this) (Figure 2a). We now show the simulated CN for all air parcels once without nucleation and then with nucleation (Figure 2b+3b). And we show the different sulfuric acid profiles and describe step-by-step how we derived them (Figure 1b+3a+4). We moved the comparison to the mid-latitude and Arctic profiles in a separate section (now 3.4) and added a comparison to the derived profile of Campbell and Deshler (2014) as reviewer 2 suggested (also 3.4).
We shortened the descriptive text of the sensitivity studies a lot and now show the effect of a 5 K temperature increase in Fig 6d.
We removed the discussion and the figure about the sulfuric acid production rates as we can not derive them anymore for all months.
We removed the comparison to the global modeling of Campbell et al. (2014) as we do not have 4 profiles anymore and as we cannot test the possible causes for the altitude shift with SAWNUC.

Discussion: We rewrote the Discussion to reflect the new results. We now state clearer what our study adds to the discussion.

We thank the reviewer for the helpful comments which significantly improved the paper.

**Individual comments**

**From ACPD questions**

While the basic description of experiments and calculations are sufficiently laid out, it would be difficult to reproduce them unless more detailed descriptions/equations of the SAWNUC model are provided, possibly in a supplementary section. Although, I understand the complexity of this box model, and that providing all details is not necessarily possible.

We added a description of the most important processes to the manuscript. The major equations of the model for the individual size bins has been described in detail by Lovejoy et al. (2004).

For this study, the basic processes simulated by SAWNUC are condensation and evaporation of sulfuric acid and coagulation for every size bin. Preexisting particles are fully simulated as particles and not just as surface area to which particles can be lost. Condensation and evaporation of sulfuric acid are the dominating processes for the formation of new particles while coagulation and condensation of sulfuric acid, if present, determine growth and reduction of existing particles.

While the overall presentation is good, there are some areas that can be better structured and written. These include grammatical errors, a cumbersome writing style, confusing section titles and organization, and a general writing fatigue in the later sections of the paper. These issues need to be fixed to improve the overall quality of the paper. I make suggestions in the annotated PDF of the paper.

Thank you for the suggestions. We followed them closely. We also changed the structure to make it clearer now. We reformulated various parts of the later sections.

There are no mathematical formulae present, but it may be nice to include some of the fundamental equations used in SAWNUC (e.g., the major nucleation equation that is inverted) possibly in a supplemental section. Showing and explaining the driving terms in the SAWNUC nucleation equation would be helpful.

See above. We added the major processes in the description of SAWNUC and in the description of our method in Sect. 3.1.

**Introduction**

Page 1, Line 25: This is technically not true, because Campbell and Deshler (2014) note that for balloon-borne measurements, this two-cylinder growth chamber (to grow CN measurements) is mounted vertically on top of an optical particle counter sensitive to particles> 150 nm radius [Rosen, 1964]. This CN instrument has been used to measure CN above Laramie since the early 1980s and above McMurdo since 1986. Thus their CN counters measure particles that could be larger than what you state as a 300 nm upper size threshold for a "typical CN counter". Please revise. Rosen, J. M. (1964), Vertical distribution of dust to 30 kilometers, J. Geophys. Res., 69, 4673–4676, doi:10.1029/JZ069i021p04673.

Thank you again for clarifying that point. We revised it as part of the change of the first section.

Therefore, condensation nuclei (CN) are defined as all aerosol particles that are large enough to be measured by a CN-counter, which typically can measure particles with diameters larger than ~10 nm.

Page 1, Lines 25-26: It is important to note that Campbell and Deshler (2014) results also qualify Murphy et al. [1998, 2013], where it is suggested that in polar winter, sulfuric acid primarily condenses on nonvolatile meteoritic material. Campbell and Deshler results are in general agreement with this, except in the CN layers with significant, and rapid, formation of new particles without a nonvolatile core. Its possible that the upper boundaries of the CN layer may be impacted by meteoritic material. This may be even more important to introduce here, if some type of sensitivity run is completed by adding an enhancement in theoretical meteoritic pre-existing particles are added to your study, and impacts on sulfuric acid profiles and CN layers are discussed (see comments in Sections 2 and 3). Campbell, P., and T. Deshler (2014), Condensation nuclei measurements in the midlatitude (1982–2012) and Antarctic (1986–2010) stratosphere between 20 and 35 km, J. Geophys. Res. Atmos., 119, doi:10.1002/2013JD019710. Murphy, D. M., D. S. Thomson, and M. J. Mahoney (1998), In situ measurements of organ- ics, meteoritic material, mercury, and other elements in aerosols at 5 to 19 kilometers, Science, 282, 1664–1669, doi:10.1126/science.282.5394.1664. Murphy, D. M., K. D. Froyd, J. P. Schwarz, and J. C. Wilson (2013), Observations of the chemical composition of stratospheric aerosol particles, Q. J. R. Meteorol. Soc., doi:10.1002/qj.2213.

Thank you. We included this as part of the rephrasing of the first passage and later in the introduction. In our changed method all particles could contain a nonvolatile core, except the ones that nucleate in the CN layer. Therefore, if they were measured by the CN counter, the meteoritic preexisting particles are now included in our simulation.

Page 2, Lines 11-12: Please be consistent with your "sulfuric acid" or "H2SO4" convention through the paper. It is used rather interchangeably.

Done

**Method**

Page 3, Lines 9 – 11: This is rather different then in Campbell et al. (2014), where their CARMA model divides sulfate aerosols amongst 30 size bins in CARMA, with sulfate mass increasing by a factor of 2.4 between adjacent bins. The dry radii of these 30 aerosol size bins range from 0.343nm to 2.17 μm.

Yes, it is rather different. The design with the linear bins first is one of the key elements of SAWNUC to accurately simulate the nucleation and is deeply implemented in the code. This can not be changed. However, we did change the geometric scale factor for the larger bins to 2.4 and thereby increased the size of the largest bin. It had only a very small effect on the derived profiles.

Page 4, Line 9: Is it technically true to call the SAWNUC output CN concentrations, as you have previously defined CN from a measurement perspective? The output from SAWNUC contains concentrations of particles that are not observable in CN counters at the supersaturations employed in Campbell and Deshler, 2014 (~ < 3 - 10 nm). Maybe just state "Particle concentrations and sizes...."

We agree. Done.

Page 4, Lines 12-13: What is the chosen 5 nm detection limit based on here? Is this an arbitrary cutoff selection for the model, or do you have appropriate reference for this? Campbell and Deshler (2014) report that their CN measurements contain all particles r > 3 - 10 nm in size, dependent on pressure altitude.

We did choose it as a lower limit of the 50 % cutoff (as 6 nm diameter was reported as 75% cutoff). However, during the redesign of the paper we now changed it to the reported 20 nm diameter for all altitudes and use 6 nm for all altitudes in the sensitivity test.

Page 4 Lines 16 – 17: Provide justification for this H2O profile. Show references, such as "...based on July MLS and hygrometer measurements in Figure 7a in Campbell and Deshler (2014)", or if you used a combination of different measurements to get your assumed profile.

Done.

Page 5, Lines 9 – 11: What altitude(s) is (are) represented in the model "maximum trajectories" in Figure 1? This information is very important to describe and show. If these "maximum trajectory" altitudes are representative of the matched maximum CN concentrations in Campbell and Deshler (2014) Figure 1, then better explanation is needed in this section.

Thank you, we added a Figure that shows the trajectories and rewrote the text.

Page 5, Lines 17 -21: Why a decrease in10% of CN from the May 1 - June 15 period to the beginning of June/July period? Can this possibly be based on limited CN measurements in June/July from Campbell and Deshler (2014)? Is this an arbitrary value?

It was a somewhat arbitrary choice. Some decrease is expected because of coagulation and we chose it to be 10%. We redesigned the preexisting particles simulation to avoid this choice.

Page 6, Lines 7 – 8: What are these "assumed functions"? Are they derived from Campbell et al. (2014), for example in Figure 3 they show time series of CN concentration and maximum CN altitude with time for McMurdo, derived from many years of measurements at different julian days.

No, they were really assumed; laid around the monthly values and chosen so that the monthly mean matches to the measurements.

Page 6, Lines 32-33: Does this statement really justify that neglecting temperatures < 190 K introduces little uncertainty. Suggest rewording, if not additional discussion as to the uncertainty it really introduces to the nucleation rate. See Figure 5d.

(The referred phrase was moved to Sect. 2.2.) We agree. We now use the sensitivity test to show the influence in Fig 6d.

This introduces some uncertainty which is estimated in our sensitivity test of a 5 K temperature increase (Sect. 3.3).

However, this temperature sensitivity has to be considered when interpreting the July and August profiles at low altitudes as there the temperature was below SAWNUC's lower temperature range of 190 K (maximum 5 K below, see Fig. 1a).

Page 7, Lines 2-3: What part of the design does not allow for these three trajectories in Sept? Rather a mystery here for the reader.

Thank you. We discuss it now in more detail in the rewritten results section.

In September above the CN layer at ~28 km, too many CN are simulated even without sulfuric acid being present (Fig. 2b). This is the result of an air volume compression in the subsiding air parcels from 31 km in August to 28 km in September which increases the CN concentration by ~60 %. Here, coagulation is not efficient enough to reduce the monthly mean CN to the observed value.

**Results**

Page 7, Line 28: What is the total stratospheric sulfur value based upon?

We agree that a reference would be needed. However, according to the new modelling results this discussion was changed anyhow.

Page 7, Lines 29 – 30: Issue regarding your definition of a "typical CN Counter" and the statements made here. Typical CN counters in Campbell and Deshler (2014) and Campbell et al. (2014) measure all particles with r > 3 - 10 nm (dependent on pres- sure/altitude), and thus could technically count particles above 300 nm. Thus, I do not understand why the derivation mechanism fails here.

We agree. See discussion at the beginning.

Page 8, Lines 3 – 5: Also Murphy et al. (2013) suggested that in polar winter, sulfuric acid primarily condenses on nonvolatile meteoritic material. The results of Campbell and Deshler (2014) are in general agreement with this at higher altitudes, except in the CN layers with significant, and rapid, formation of new particles without a nonvolatile core. Murphy, D. M., K. D. Froyd, J. P. Schwarz, and J. C. Wilson (2013), Observations of the chemical composition of stratospheric aerosol particles, Q. J. R. Meteorol. Soc., doi:10.1002/qj.2213.

Thank you, we added this to the introduction and discussion sections.

Page 8, Lines 8 – 21: This is an intriguing comparison to measured mid-latitude profiles indeed. However, couldn't the SAWNUC inversion method also be applied to derive a mid-latitude sulfuric acid profile using the mid-latitude seasonal CN/temperature profiles shown in Figure 1a and 1b of Campbell and Deshler (2014)? I would like to see an additional derived mid-latitude sulfuric acid profile since it is more a direct comparison to the mid-latitude measurements, and also does not incur the same altitude/adiabatic expansion shifts when comparing polar to mid-latitude regions. The summary of Figure 1 in Mills et al. (2005) show sulfuric acid measurements mainly during September and October, so you could use the representative fall season CN/temp profiles in Campbell and Deshler Figure 1a and 1b to derive a "fall mid-latitude sulfuric acid profile". This can provide supplemental information on the performance of the SAWNUC inversion method used.

(This comparison was moved to Sect. 3.4) Thank you for the suggestions. Unfortunately, as the CN layer above Wyoming is expected to form in the polar region and then is transported to the mid-latitudes we can not model its nucleation properly. But we added a discussion about how the Antarctic profiles can be compared to mid-latitude profiles in Sect. 3.4.

Page 8, Lines 31 – 33: You should add that these results also further support Campbell and Deshler (2014) findings that the majority of the observable polar CN layer is due to neutral nucleation.

Added in the discussion

In agreement with Campbell and Deshler (2014) we find that the development of the CN layer can be explained by neutral sulfuric acid-water nucleation.

Page 9, Line 8: What (if any) sensitivity runs were performed due to holding H2O profile constant? Were different H2O profiles tested? It is likely a small uncertainty in the nucleation rate, but may be worth investigating and mentioning.

Yes, we also tested 5 ppm everywhere and it results in only very little change.

Additional sensitivity studies (not shown here) showed that the exact amount of ions or water molecules (e.g. 5 ppm everywhere) has only a small influence on the derived profiles because the ion concentrations are high enough so that they are not a limiting factor, and the few parts per million stratospheric water vapor uncertainty is too small.

Page 9, Lines 11 – 14: Since the profiles derived are ~> 20 km, then why is the cutoff > 5 nm chosen, which is more representative of lower atmosphere conditions. Shouldn't the profiles be derived using a cutoff closer to > 20 nm (as reported in Campbell and Deshler, 2014)?

Yes, we changed this in the new method and in the text.

The measured CN were compared with the simulated CN by summing over all simulated particles with diameter above 20 nm, as Campbell and Deshler (2014) reported a detection limit of their CN counters of 6-20 nm diameter.

Page 9, Lines 15 – 16: If the main growth mechanism to a higher observable cutoff size was due to coagulation, then does the sulfuric acid concentrations have to increase in the nucleation areas? In other words, what effect does coagulation rates have on this sensitivity? Also, the profiles are nearly the same above about 28 km between June/July and September, regardless of cutoff size.

Here the coagulation is not the main effect but condensation of sulfuric acid on the small clusters. Therefore more sulfuric acid is needed. This effect reduces with higher sulfuric acid amounts as at these high concentrations, the clusters grow quickly after nucleation.

The lower cutoff leads to lower sulfuric acid concentrations as the nucleated CN do not have to grow as large by sulfuric acid condensation to be counted (Fig. 6a). This effect decreases with increasing sulfuric acid as at higher concentrations, the clusters grow quickly once they are nucleated.

Page 9, Lines 21-22: Ultimately, it would be best to avoid using a constant CN measurement cutoff size, since it is clearly dependent on altitude. Can this be varied for the multiple trajectories used in SAWNUC to derive the sulfuric acid profiles, possibly arriving at a "best guess" sulfuric acid profiles over McMurdo?

Yes, the cutoff could be varied but more information is needed for that e.g. a cutoff vs. altitude dependence. We are not sure how we could arrive at a best guess as there are too many unknowns. Note, however, that in the revised version the cutoff uncertainty is not as high anymore as we assume the CN outside the layer to be large particles.

Page 10, Line 10: Does both coagulation and condensation rates both have little influence?

Yes, it is now clearer.

A reduction of all coagulation and condensation rates by 20% increases all profiles only a little […]

Page 10, Line 16: What is this inter-annual CN increase based on? Campbell and Deshler (2014) indicate a range in CN concentrations dependent on altitude in Figure 3, and for the CN layer maximum level in Figure 4 for McMurdo.

We agree that this was not a proper choice as the actual CN variation is more complex. We decided to remove the CN increase and just vary temperature in the sensitivity test.

Page 10, Line 28: This whole section seems rather out of place in the paper, and would be better placed as the last summary part for Section 3, and named "3.4 Summary and discussion of model uncertainties"

Yes, we agree. We now only use the sensitivity tests in the comparison and discussion sections to interpret the results.

Page 11, Line 1: I don't see a compensation between the monthly values and cutoff size, as they are in the same direction for both June/July to September and in October. Additional clarification is needed.

Yes, they are in the same directions. But the monthly values already were an overestimation and the cutoff showed that the values should be higher. So this matches. However, as we now use a different cutoff size, this text is not needed anymore.

Page 11, Lines 5 - 6: If accepting my previous comment about moving this whole section to the last part in Section 3, this statement could then be made more robust and stated such as the following: "The June/July profiles have additional uncertainty due to rather unknown sulfuric acid production rates from processes that do not require sunlight over Antarctica in the winter. This uncertainty is compounded by sparse CN measurements made during June/July (and prior)

over Antarctica."

Good suggestion, thank you. The production rates were removed from the paper.

Page 11, Lines 7-8: This should not be a separate paragraph if that was what was intended. Also, I do not know if this can be quantitatively determined from the results presented. Please make sure to state that this is a qualitative approximation. Finally, from the discussion of the sulfuric acid production rates for June/July, it seems that this period may have even greater uncertainty compared to the October period, especially considering the much less CN measurements during June/July compared to October (see Figure 3a in Campbell and Deshler, 2014)

We agree that this was not quantitative. We removed the statement.

Page 11, Lines 17 – 18: Shouldn't this be "therefore this production term cannot represent any sulfuric acid that evaporates from preexisting particles"

No, the idea was that everything that is not explicitly simulated could be included in this production term. (This section was removed.)

Page 12, Lines 2 – 3: Please clarify what you mean by "one additional step away from the measurements"

The idea was that going from CN measurements to sulfuric acid concentration is one step which involves the processes simulated by SAWNUC. Determining the production rates represents a second step as this could also include processes that are not simulated by SAWNUC, which makes these results more uncertain. (This section was removed.)

Page 12, Lines 3 – 4: This is a confusing argument. You note that the derived sulfuric acid profiles in Figure 6 could be an overestimate (by a max of one order of magnitude) in this model, but then wouldn't the inclusion of these processes further increase the sulfuric acid production rates, and thus lead to a possibly even greater overestimate (although understandably unknown). In essence, it may indeed be difficult to compare to the limited measurements of Krieger and Arnold over the Arctic, and may in fact be underestimated in this model for the Antarctic June/July period. Please revise this statement.

We agree that this sentence was poorly phrased but the whole section was removed anyhow.

**4 Comparison with global modeling**

Page 12, Lines 8 -12: Why not generate a plot to show these temperature, sulfuric acid, and CN comparisons. It would be much clearer for the reader to associate the altitude discrepancies between this work and Campbell et al. (2014) during early July and late October, and beneficial

to the discussions that follow in this section.

Page 12, Lines 16 – 17: While this is indeed possible, is there any evidence of OH production of sulfuric acid over the Antarctic region? I would be very careful in drawing such suggestions that cannot be backed by measurements. Also, you make suggestions regarding July underestimate by global model compared to SAWNUC, but what about the October overestimate?

We agree, but as we decided to drop the comparison it is no longer relevant.

Page 12, Lines 18 – 23: This argument seems to suffer from some writing fatigue and is rather weak. No major assessments are made as to why this altitude shift of nucleation controlling variables occurs in the global model. Please improve this paragraph and make better supporting statements/arguments. Are you (and the satellite obs. by Hopfner et al., 2013) suggesting too high $SO_2$ at hight altitudes in the global model compared to observations and SAWNUC? And what can be inferred as to the cause of this using your SAWNUC model (e.g., enhanced $SO_2$ subsidence from mesosphere)?

Thank you. While we think that a comparison of the global model $SO_2$ with the satellite could improve our understanding, we removed it as there is no direct relation to the SAWNUC model.

The idea was that in the satellite data the high $SO_2$ values penetrate deeper in the Antarctic stratosphere than they do in the global model. Therefore there should be more $SO_2$ at lower altitudes according to the satellite, which then could lead to the formation of the CN layer at lower altitudes. Therefore it looked like this downward transportation of $SO_2$ and maybe the downward transport of everything else was not strong enough in the global model. However, this is speculation and we would like to be cautious and therefore removed the comparison.

Page 12, Lines 24 – 28: Section 3 and Figures 1 and 3 in Campbell et al. (2014) already support the ability of the global model to reproduce the observed temperature, water vapor, and CN profiles Antarctica and mid-latitudes, with the exception of an altitude shift in CN over Antarctica, and possible underprediction in magnitude of sulfuric acid. Again, can you provide potential reasoning into why this altitude shift occurs for all nucleation controlling variables against SAWNUC? Have you checked to make sure that there are no discrepancies between altitude definitions in the global model and how they are defined in SAWNUC to ensure an apples-to-apples comparison? Remember that the global model is an Eulerian framework, while the trajectories predicted in SAWNUC are Lagrangian.

Yes we made sure that we compare the right altitudes. In Campbell et al. (2014) Figure 1b it looked to us as if the temperature was underpredicted by the global model at higher altitudes. Which could result in the nucleation there. Then both controlling variables would have the altitude shift. But yes, we cannot support or model this properly with SAWNUC and therefore we removed the comparison with the global model from the paper.

**Summary**

Page 13, Lines 6 – 7: This is misleading as stated, because it implies that the global model in Campbell et al. (2014) was run with their derived sulfuric acid profiles, while in fact, it was run with the WACCM sulfuric acid profiles, and the derived profiles were used for further insight and to assess possible uncertainties in the WACCM profiles. Please revise.

We agree that this was misleading. We removed the reference to Campbell et al. (2014) from the summary as we do not discuss the global modeling any more.

Page 13, Lines 16 – 18: This argument could be improved from my previous comment regarding derived SAWNUC sulfuric acid profiles over mid-latitudes compared to measurements over mid-latitudes.

Yes, we added a short discussion about the mid-latitudes in the comparison section.

Page 14, Lines 2- 3: While this section adequately summarizes the findings presented here, more discussion on what SAWNUC can answer as to why the global model predicts sulfuric acid at too high an altitude, will make the discussion and results more robust in this paper. What are some of the potential processes leading to this altitude shift?

Unfortunately, SAWNUC can not answer more than Campbell et al. (2014) already did. We can just confirm the expectation of Campbell and Deshler (2014) that their profiles are an underestimation. But they already discussed the consequences of this for the global modeling in detail in Campbell et al. (2014). This is why we removed the comparison to the global model. If we should speculate for reasons for the altitude shift, we would suggest either the downward transport as mentioned above. But maybe a too fast oxidation of SO2 could lead to the same results? This question can not be answered with our box model, but only with a global model maybe by analyzing tracers… We changed the Conclusions section so that we now not only summarize but also say what knowledge we add to existing one.

We thank the reviewer again for the helpful comments which significantly improved the paper.

**Answers to Referee No. 2**

We want to thank the reviewer for carefully reading our manuscript and for providing numerous helpful comments. The reviewers' comments are repeated in full below in black font, with our replies indicated after each comment in blue font. Text which has been changed in the manuscript is shown in red font.

**Major changes**

As both reviewers noted we used an incomplete definition of condensation nuclei (CN) in the text and also for our simulation. CN were defined in our model as just the smallest particles that are smaller than those counted by the optical particle counter (diameter $d_p < 0.3$ μm) instead of all particles that can be counted by a condensation nuclei counter. Usually this does not make a big difference as the smaller particles dominate in number. However, for our study, the two formulations unfortunately make a substantial difference.

We assumed that the CN presented in Campbell and Deshler (2014) were all smaller than 300 nm diameter (as this was the cutoff of their OPC). Therefore we simulated these CN as small particles and *additionally* simulated the preexisting particles surface area loss by the larger particles. This extra loss term reduced our simulated CN which made nucleation of new CN necessary in all months at all altitudes and not just during the formation of the CN layer. This is why we could derive profiles for all months; which was the major point of criticism by Reviewer 2.

We now corrected our method. As a result, we do not simulate the long warm up phase anymore, but start at the beginning of July and use the CN of Campbell and Deshler (2014) as initial particles for the first month. These CN are now the only particles in our simulation, we have no additional loss to preexisting particles anymore.

This changes our results a lot. We now do not need sulfuric acid anymore for reproducing the observed CN, except for the formation of the CN layer. This solves the major point of Reviewer 2.

We can not derive profiles of sulfuric acid for all months anymore, and therefore some of the main messages of the paper change. We focus more on the result that we can reproduce the observed CN just with coagulation and air volume compression except the CN layer formation. This is consistent with Campbell and Deshler (2014) and the general idea of the formation of the CN layer: low sulfuric acid during winter but then an increase after the onset of sunlight. However, we still derive profiles at which nucleation would occur and interpret these as upper limits (as suggested by Reviewer 2).

As the method and results changed, large parts of the paper were rewritten. We did this with the suggestions of the reviewers in mind and we also used this opportunity and changed the paper to active-writing-style. Here are the major changes:

Title: We changed the title of the paper to" Nucleation modeling of the Antarctic stratospheric CN layer and derivation of sulfuric acid profiles". This flip of the two parts reflects that the derived profiles are not our main/only result any more and this is also the order in which we present and discuss the results.

Abstract: The abstract is almost completely rewritten and describes the new results of the paper.

Introduction: We added more information on Antarctic stratospheric CN to the introduction as suggested by the reviewers. We removed the discussion of the too high CN layer altitude in the global model of Campbell et al. (2014) as we do not discuss this in the paper anymore (see below).

Methods: We rewrote and shortened this section. We shortened the SAWNUC description (Sect. 2.1) but added a short description of the most relevant processes. We removed the preexisting particle discussion from Sect. 2.2 as this is not necessary anymore. As reviewer 2 suggested the detailed discussion of the single parcel simulation is not necessary.

Results: We rewrote the Results section according to the new modelling results. We start with a simple reference case (Figure 1). We give additional information on the simulated descending air parcel paths (as both reviewers asked for this) (Figure 2a). We now show the simulated CN for all air parcels once without nucleation and then with nucleation (Figure 2b+3b). And we show the different sulfuric acid profiles and describe step-by-step how we derived them (Figure 1b+3a+4). We moved the comparison to the mid-latitude and Arctic profiles in a separate section (now 3.4) and added a comparison to the derived profile of Campbell and Deshler (2014) as reviewer 2 suggested (also 3.4).
We shortened the descriptive text of the sensitivity studies a lot and now show the effect of a 5 K temperature increase in Fig 6d.
We removed the discussion and the figure about the sulfuric acid production rates as we can not derive them anymore for all months.
We removed the comparison to the global modeling of Campbell et al. (2014) as we do not have 4 profiles anymore and as we cannot test the possible causes for the altitude shift with SAWNUC.

Discussion: We rewrote the Discussion to reflect the new results. We now state clearer what our study adds to the discussion.

We thank the reviewer for the helpful comments which significantly improved the paper.

**Individual comments**

**Major points**

The primary issue is the authors' seeming assumption that underlying the presence of any stratospheric sulfuric acid particle concentration is a necessary sulfuric acid concentration. Yes there was a sulfuric acid profile at one time, but once the sulfuric acid condenses its concentration decreases to vanishing amounts. Because there are CN particles existent during polar winter does not mean that they were nucleated then and there. Stratospheric CN are ubiquitous in concentrations near 10 cm-3. Thus as the polar vortex forms the CN particles present at the beginning will persist in the vortex, their sedimentation quite negligible. The observations of Campbell and Deshler [2014] bear this out for much of the CN profiles they present. Here, however, the authors derive a sulfuric acid mixing ratio profile in times of the year and altitude regions where no new particle formation is observed. They then speculate on what could be causing the oxidation of so2 to obtain the sulfuric acid. Isn't this speculation on possibly an imaginary profile?

I understand when sulfuric acid particles are present there will be a residual sulfuric acid concentration; however, given that the vapor pressure of 75% by weight sulfuric acid is exceedingly low, this concentration will be quite small. Is this what is calculated by SAWNUC in the absence of new particle formation? The authors need to make clear at the outset how the observations of new particle formation are used in deriving the sulfuric acid profiles and how such profiles are obtained in the absence of new particle formation. In the latter case can there be anything more than an upper limit?

Thank you for pointing this out and we agree with all these points. As we describe above we had an additional loss term in our simulation. This loss removed particles that the model replaced with new nucleating particles. This point is corrected in the revised version and now the results match the expectations that no sulfuric acid profile is required during winter time.

The other major issue is the writing is not up to the standards expected for a scientific paper. The writing is imprecise, many details are missing, while irrelevant comments creep in. The authors should consider having some English help prior to the next submission and not keep the reader working out what is intended to be said.

We put significant effort into revising the manuscript, clarifying the structure, making the descriptions more precise, removing irrelevant and speculative comments and improving the English style.

**Abstract**

1.13. Reproduces what observations? Be specific here. Reproduces the new particles that are observed or . . .?

The formulations should be clearer in the new version.

1.14 As will be mentioned further, below, it is not clear how a sulfuric acid profile is obtained in July. At most this should be an upper limit.

We agree and changed the text accordingly.

1.16 "the observed magnitude of CN. . ." What does this mean? Was nucleation of new particles observed at higher temperatures?

This was about the discussion of temperatures (see below). We removed the statement from the abstract.

1.18-19. The authors have not convinced me of the need to include a sulfuric acid production rate, which is then one further step removed from the observations. Thus I see no need to go further and try and explain a winter-time production mechanism, when it has not been established that one is required.

We agree with this comment. With the correction to our calculations as stated above, a winter-time production mechanism is not needed anymore.

1.20. Why is there no comparison with the sulfuric acid profile derived by Campbell and Deshler [2014]?

Added. Thank you!

Our concentrations are about one order of magnitude higher than previously presented concentrations as our simulations consider that nucleated clusters have to grow to CN size and can coagulate with preexisting particles.

**Introduction**

1.23-26. This is a terrible first sentence. In fact it is not even a sentence. The clause set off with commas is a complete sentence and should stand on its own. But it is still a poor first sentence. In fact the definition of condensation nuclei (CN) is not correct. Condensation nuclei are observed by forcing small particles to grow to optically detectable sizes using a condensing fluid and a supersaturation chamber. There are a number of references available on the technique and perhaps one should be included. The size ranges vary somewhat, but typically the CN concentration is used to define the total aerosol concentration larger than some nominal size, say 10 nm. CN measurements also include particles > 300 nm, but their concentration is inconsequential compared to the total aerosol population.

We rewrote considerable parts of the introduction including the first sentence. We tried to improve the readability and to include a correct definition of CN as well as giving a proper reference.

Atmospheric aerosol particles are of interest due to their various influences on radiation, clouds, chemistry, and air quality. Condensation nuclei counters measure the number concentration of aerosol particles by growing them to optically detectable sizes by condensation (e.g. McMurry, 2000). Therefore, condensation nuclei (CN) are defined as all aerosol particles that are large enough to be measured by a CN-counter, which typically can measure particles with diameters larger than ~10 nm.

1.27. Not all CN are formed by nucleation. It appears a significant fraction of CN are non-volatile suggesting that this fraction of CN particles appears due to the condensation of sulfuric acid on pre-existing solid cores [Borrmann et al., 2010; Campbell and Deshler, 2014].

Yes, we included this now:

In the Antarctic polar vortex a background CN concentration of ~10 cm$^{-3}$ is found (Campbell and Deshler, 2014). Volatility measurements indicate that more than half of them have a nonvolatile core which could be meteoric material (Campbell and Deshler, 2014; Curtius et al., 2005). Sulfuric acid is expected to condense on these CN (Borrmann et al., 2010; Murphy et al., 1998, 2013) […]

2.12. Where does the h2so4 in the mesosphere come from to be available for photolysis?

Potential sources are evaporation of particles and maybe oxidation some remaining OCS and SO2 in the higher stratosphere/lower mesosphere.

2.15 "performed continuously for 24 years" What does this mean every month, every day? I understood the observations are only available for a few months each year.

Added

More recently, Campbell and Deshler (2014) presented an overview of all balloon-borne CN measurements between 15 and 35 km above McMurdo Station, Antarctica (78°S), that were performed between 1986 and 2010, 2-3 times a year during winter.

2.17-20. Very awkward sentence. Try. They present monthly averaged CN concentra- tion and temperature profiles which capture the unperturbed CN, with concentrations around 10-20 cm-3 in June/July as well as the development of a layer of particles at 20-25 km, with concentrations increasing to 100 cm-3 from August until October during sunrise and warming.

Done. Thank you.

They present monthly averaged CN concentration and temperature profiles which capture the unperturbed CN, with concentrations around 10-20 cm$^{-3}$ in June/July as well as the development of the CN layer at 21-27 km, with concentrations increasing to 100 cm$^{-3}$ from August until October during sunrise and warming.

2.26-28. Campbell et al. [2014] did not invert the nucleation equation, they ran a three dimensional chemistry model which produces its own h2so4.

Yes, they run the 3D chemistry model. But, they also compared this to a derived sulfuric acid profile by inverting the nucleation equation. They describe this in more detail in Campbell and Deshler (2014).

Campbell and Deshler (2014) describe a method where they derive an Antarctic sulfuric acid profile from the measured CN by inverting the neutral binary nucleation equation. They used the difference between the CN before sunrise and two weeks after sunrise averaged over all years to derive a nucleation rate for all altitudes from which they derived the corresponding sulfuric acid.

2.30. Who are they?

Campbell and Deshler (2014) (This sentence was removed.)

**Method**

3.30. Delete the sentence, "This section describes the derivation of the Antarctic stratospheric sulfuric acid profiles" and add the information to section title, e.g. Deriving the Antarctic stratospheric sulfuric acid profiles.

Done.

4.14. "3e5 ion pairs per gram of air and second" Awkward. Use standard SI units and symbols

to describe a rate.

It was communicated to us in this unit and they also use this units in their papers, so we would like to keep it like this. But we added an example conversion to ion pairs cm-3 s-1.

The ionization rate of the Antarctic stratosphere in August-September 2010 was 3e5 ion pairs per gram of air and second (Ilya Usoskin, personal communication, 2013; according to Usoskin et al., 2011) which converts to e.g. ~10 ion pairs cm$^{-3}$ s$^{-1}$ at 200 K and 20 hPa.

4.16-18. There should be some references for the water vapor profile and the diabatic descent rate?

Done.

The water vapor profile for July was chosen to be a linear increase from 3.0 to 6.0 ppm from 18 km to 25 km and above a constant value of 6.0 ppm up to 32 km based on MLS and hygrometer measurements in Fig. 7a in Campbell and Deshler (2014).

4.19-24. 100 nm is at the upper limit of possible pre-existing particles. What concentrations are used for these particles? Since the surface area is chosen based on reasonable sources why is the particle size important? What does it mean that the surface area was converted . . . according to the ideal gas equation? Do you mean the surface area density was adjusted to different temperatures and pressures assuming a constant mixing ratio?

The concentration was calculated from the surface area and the size. Size and concentration were necessary to calculate the coagulation rates on these preexisting particles. Yes, it was adjusted to different temperatures and pressured assuming a constant mixing ratio.

However, as we now fully simulate the preexisting particles in the model, this preexisting surface area loss is not needed any more.

4.28-31. Is there really a trajectory of an air parcel here, or do the authors just mean variations in the various parameters required by the SAWNUC model? The language is confusing. If the latter then I think the more appropriate words would be box model simulation rather than air parcel trajectory. If the former then where does the air parcel start, what altitude, location, dimensions? What does it mean to "connect" to the monthly maximum of the measured CN? Do the authors mean the trajectory which reproduces the monthly maximum CN concentration? The next sentence is equally awkward, "It is assumed that this maximum of the particle concentration resides in a single air parcel that descends inside the polar vortex." The maximum of the particle concentration doesn't reside in the air parcel until the nucleation has occurred. What the authors are trying to suggest, as far as I can understand, is the optimum trajectory to reproduce the observed maximum in CN concentration. The phrase "maximum trajectory" does not convey this idea and I would recommend changing this language to

something more representative of what is being simulated. Something like optimum trajectory to reproduce the observed maximum in CN concentration. This maximum in CN concentration is height dependent. Is this height dependence consistent with the diabatic descent in the model?

Thank you for pointing out these issues and the imprecise wording in our explanation. We rewrote the whole description and tried to include these points. We also added a Figure which shows the assumed descent pathways of the simulated air parcels.

We assume that the CN maximum of each month resides in a single subsiding air parcel, and place the other air parcel trajectories around this CN maximum pathway (Fig. 2a).

5.1. I believe the authors mean the sulfuric acid concentrations were varied on a monthly basis? Search implies that these concentrations exist on some data base or?

Yes, we don't use the word search anymore.

5.4-6. Awkward and unnecessary detail. The authors are merely iterating the model until they achieve a match with the measured CN by varying the sulfuric acid concentrations. If so say this. It is not the sulfuric acid concentration that corresponds to the monthly CN concentrations, it is the simulated CN concentrations based on the sulfuric acid concentrations. The writing here and elsewhere is awkward and requires the reader to do a lot of guessing as to what is meant.

Thank you. We rewrote the methods and the results section and rearranged the structure to make this much clearer in the new description.

5.14-18, Fig. 1b. The model is initialized with a CN concentration of zero. Is this realistic? There will be some residual particles, CN, persisting in the air as the polar vortex forms. It seems more plausible to assume these residual particles appear as the background of about 10 cm-3 shown in Campbell and Deshler [2014] outside of the CN layers than to assume a concentration of zero. If the model is allowed to initialize to 110% of the June/July CN maximum, then there would not be any further new particle formation in the June/July period. So no additional sulfuric acid is required. This point needs to be clarified.

Thank you for pointing that out. We changed our method of the initial particles so that this should not be a problem anymore. Now we just use the June/July measurements as initial preexisting particles for the following months.

5.18-19. Since the model is already at 110% of the CN maximum, why is further searching required? Is sulfuric acid required to maintain the pre-existing particles? Are these particles losing sulfuric acid? I understood these particles change size due to fluctuations in water based on temperature but not on the loss of sulfuric acid.

This was again because of the wrong additional losses. So some CN had to be produced again.

It is not clear how the subsequently derived sulfuric acid profile for June/July is obtained. How does it differ from the sulfuric acid profile that is required to build up the CN from zero to 110% of June/July values? How does the work here then take advantage of the measurements which show a maximum of CN near 30 km above a possible background of around 10 cm-3? If there is no new particle formation in June July then the speculation on the source of a sulfuric acid profile is misplaced.

Yes, we changed our method and now have no July sulfuric acid profile any more.

5.26. Please show the mean CN values on Fig. 1 a. It would also be instructive to include on both panels the observations which are being reproduced.

We agree. But as we removed this Figure as suggested by the reviewer this is not necessary anymore.

5.27-28. If there is a decrease from June/July – August as could be argued from Campbell and Deshler [2014] since the CN maximum concentration does not change in their Fig. 1, then it appears again that no new particles are formed so only an upper limit can be placed on the sulfuric acid concentration.

We agree. We clarify in the revised version that we can only give an upper limit for the sulfuric acid concentration.

5.32. How is this month long time interval different than the previous ones? Why is the sulfuric acid concentration now strongly increased? In Campbell and Deshler [2014] there is almost no change in CN concentration between September and October in the CN layer, arguing again for little information on a sulfuric acid profile.

It is not different. It had to be strongly increased because some nucleation was needed (due to the additional CN loss) and temperature is much higher in October.

5.10-6.4 and Fig. 1a). Is all of this necessary to understand what is done, and do the authors really want to explain the subtleties of Fig. 1a? In the end more realistic time steps are used and profiles created. I don't really think the reader is benefitting any from the previous discussion and it raises many questions such as listed above.

We agree. Especially as now as no sulfuric acid is necessary in July and August (and October) this is not needed anymore. We shortened this section but extended the description of the modelled air parcel paths to clarify our approach and the results.

6.17-24. This paragraph is confusing by suggesting that the sulfuric acid is dependent on other processes in the model rather than just the amount needed to create the CN maximums. Here

the authors claim that the sulfuric acid is dependent on sunlight, which we know it is, but I didn't think this was considered by the SAWNUC model.

We wanted to show that the profiles agree with the big picture. But we see that this could be rather confusing as we did not produce these results. We agree that it needs to be clarified which parts are simulated by SAWNUC and which are the assumed process that could lead to the simulated values. We considered this during the rewriting of the text and concentrate on the results that can be simulated by SAWNUC.

6.21. What is meant by, "the chemical lifetime of SO2 by OH"?

We meant how fast the SO2 is oxidized by OH. The lifetime is the SO2 concentration divided by the loss rate. However, this paragraph was removed.

6.23. Since the authors are now dealing with monthly averages, how can they make this claim, "Starting in September, the sulfuric acid amount increases first slowly and then strongly from September to October"? Monthly averages cannot supply this information.

When there is only a small change from August to September but a big change from September to October, then a bigger change is expected later in September. But we agree that this argumentation could be problematic and removed the statement.

Fig. 1b. What is the explanation for the fast rise in CN near the beginning of September, while the sulfuric acid only increases at the end of September? What is causing the CN formation if not the sulfuric acid? There is a disconnect here.

Well observed. The sulfuric acid was longer on higher levels in the beginning of September compared to the months before. So there was a sulfuric acid increase even though it was only a small one. This allowed new small particles to form and to grow to observable size.

However, now we don't show this anymore.

6.26. "To complement the profiles" What profiles? So far the authors have been calculating the conditions required to reproduce the maximum in observed CN concentrations. This would then only be a sulfuric acid concentration at one altitude in one month. Perhaps the authors intend to say they will now create profiles. If so the title of this section is misleading.

We agree, that to this point we only had one value in each month. Hope this is clearer in the new method description where we just describe the method once for all simulations.

7.3 "in September" is redundant and should be deleted.

Done.

**3 Results**

Section 2.2.3 and Fig. 2. This section and figure raise again my main question about this work. How is a sulfuric acid profile derived if there is no new particle formation? The CN layers in the CN profiles shown by Campbell and Deshler [2014] are limited in altitude to a fraction of the 18-40 km to be investigated here. Thus, for example in September above 27 km there is no evidence of new particle formation. So can anything more than an upper limit be placed on the sulfuric acid profile? Yyet Fig. 2 shows the September sulfuric acid profile to continue to increase above 27 km. How are profiles created to 40 km when there are no observations above 35 km?

The profiles only go up until 32 km. The production outside the CN layer was needed because of the wrong additional CN loss. This is corrected now.

7.10-15. Here again the authors are discussing results suggesting that the sulfuric acid profile is derived from sources outside the matching of the SAWNUC model with observations of CN.

We wanted to show that the profiles are generally in agreement with the expected bigger picture. But we agree that this could be rather confusing as we did not produce these results. In the revised version we only discuss the direct results from SAWNUC.

7.17 "the shapes of both profiles are very similar" The authors discuss the temperature profile as if it were shown in Fig. 2, but it is not. Thus the readers cannot follow this point. The temperature profile should be included in Fig. 2.

We agree and now show the temperatures profiles in the revised version (Figure 1).

7.19-21. I don't understand. Do the authors mean there is a narrow sulfuric acid concentration range leading to the nucleation of particles in the 10-100 cm-3 range? There is no reason particles cannot exist at these concentration levels if they are already there. The nucleation rate is not in a small window. The nucleation rate is determined by the concentration and the temperature. The small window is determined by both temperature and sulfuric acid concentration. Far too many or too few CN for what? I thought the particle concentrations were used to derive the sulfuric acid profile given the temperature, so if there are more or less CN this just affects the sulfuric acid concentration.

Here we discussed how much the sulfuric acid would change if we want to nucleate only some more CN. We reformulated the description of our approach significantly to make this point clearer.

7.22. "20 or 50 particles present" Are these particles already existing or nucleated particles? I assume the authors mean an observation of 20 – 50 new particles has a small influence on the derived sulfuric acid.

Yes. An observation of 20 or 50 new particles.

7.25-26. This is confusing. If the CN layer were not observed then there would be no attempt to derive the sulfuric acid profile. Above and below the CN layer there is not enough information to derive the sulfuric acid profile, only upper bounds, determined by the temperature, can be placed on it.

We agree.

7.30. This statement is not correct CN instruments measure all particles above some nominal size, typically about 10 nm. In addition I believe Campbell and Deshler report observations explicitly of particles > 150 nn using a second instrument, and no enhancements of these particles were observed in the CN layers.

Yes. This was a misunderstanding on our side.

7.27-29. These statements should give the authors pause to consider if there is an error somewhere rather than trying to explain these high sulfur amounts.

As stated above, there was an error in our assumptions. Now this is corrected and there is no need to explain the high sulfur amounts anymore.

7.30-32. This argument is incorrect. The water vapor concentration is low throughout the stratosphere. The particles are not composed of water but a sulfuric acid water mixture and the temperature determines how much water is retained by the highly acidic particle. The water vapor concentration has nothing to do with it, only temperature. There are plenty of water molecules.

We agree with this statement in general. The argumentation of 7.30-7.32 was removed in the revised version.

7.32. "To compensate there high . . ." For the reasons above I don't' understand this argument. Isn't the nucleation of particles dependent on the gathering of critical clusters of the acid not the water?

See above

8.5-7. Why is this assumption made? Campbell and Deshler, Fig. 5 show directly that the non-volatile fraction above 25 km is similar to that below 20 km, about 60%. It is not at all clear how the authors derive a sulfuric acid profile when there is no obvious nucleation. Again there can only be an upper bound.

Again, agreed and corrected.

Fig. 3. Why isn't the profile derived by Campbell and Deshler [2014] included in this figure?

The profile of C&D 2014 is included now.

8.33 "However, in the areas of higher temperatures the CN are almost exclusively produced by ion-induced nucleation." This assumes that CN are nucleated, but there is really no evidence for this, thus it is questionable whether any CN are nucleated in these areas. Sentence should be rewritten to reflect that, if particles nucleated they would require ion-induced nucleation, but in fact there is no evidence that such particles are nucleated.

Yes, we agree.

10.29-30. What is over/under estimated? Why is it clearly related to the months? I believe the authors mean the size cutoff measurement uncertainty.

This referred to the discussion of the 5 day periods. It was removed in the revised version.

11.7-8. This seems somewhat obvious as all the nucleation occurs in August, September. When there is no new particle formation this approach can only provide an upper bound on the sulfuric acid.

We agree.

11.15-16. Sulfuric acid evaporating from freshly forming particles? How does this occur? My understanding is that the vapor pressure of sulfuric acid is very low, thus it condenses readily and then stays condensed. Same question about pre-existing particles. At the temperatures used in the modeling I would not believe that the pre-existing particles would do much more than adjust their water content to the temperature changes.

Yes, in most altitude regions and for most times they do not evaporate. Only at the highest altitudes and highest temperatures in October there is some evaporation of sulfuric acid predicted.

11.14-18. If I understand correctly the production rate is simulated by continually adding the necessary h2so4 molecules to simulate the observations without specifying where the molecules come from e.g. so2, or particles. Thus all this discussion about the possible sources is speculation without basis and should be eliminated.

Yes, this was speculation. It is removed from the revised manuscript as this source is not needed anymore.

11.22-25. I am again confused as to why there is production of h2so4 in June July when the CN particle concentration is not changing from the model initialization. Much of the rest of this paragraph is again speculation and should be strictly limited.

Removed.

12.1-4. This is all redundant. Overall the paper would not suffer if section 3.3 and figure 6 were removed.

Yes we agree. Removed.

**Summary**

12.13-15. It is not clear why SAWNUC is predicting a sulfuric acid mixing ratio at this time in this region.

Removed. Not an issue anymore, due to the revised loss calculation.

We thank the reviewer again for the helpful comments which significantly improved the paper.

**Tracked changes**

[revised manuscript text omitted]

evaporation of sulfuric acid are the dominating processes for the formation of new particles while coagulation and condensation of sulfuric acid, if present, determine growth and reduction of existing particles.

**2.2 Ambient parameters**

To perform a simulation of the Antarctic stratosphere with SAWNUC, we need to know temperature, pressure, ion pair production rate, relative humidity, and sulfuric acid concentration. Particle concentrations and sizes are the model output at every time step. As we invert the model, we also need the particle concentrations to derive the sulfuric acid concentrations. Temperatures above Antarctica were taken from Campbell and Deshler (2014). Temperatures that were below 190 K (maximum 5 K below), which is SAWNUC's lower temperature range, are held fixed at 190 K. This introduces some uncertainty which is estimated in our sensitivity test of a 5 K temperature increase (Sect. 3.3). Altitudes were converted to pressures according to the global modeling of Campbell et al. (2014).

The ionization rate of the Antarctic stratosphere in August-September 2010 was 3e5 ion pairs per gram of air and second (Ilya Usoskin, personal communication, 2013; according to Usoskin et al., 2011) which converts to e.g. ~10 ion pairs $cm^{-3}$ $s^{-1}$ at 200 K and 20 hPa.

The water vapor profile for July was chosen to be a linear increase from 3.0 to 6.0 ppm from 18 km to 25 km and above a constant value of 6.0 ppm up to 32 km based on MLS and hygrometer measurements in Fig. 7a in Campbell and Deshler (2014). The mixing ratio is kept constant during the subsidence of the simulated air parcel (see below).

CN concentrations were taken from Campbell and Deshler (2014). The measured CN were compared with the simulated CN by summing over all simulated particles with diameters above 20 nm, as Campbell and Deshler (2014) reported a detection limit of their CN counters of 6-20 nm diameter. As we do not know the exact size of the measured CN, we assume the initial preexisting CN to be large CN with a diameter of 100 nm (see below), but we also perform a sensitivity study assuming a diameter of 50 nm in Sect. 3.3. We simulate them as pure sulfuric acid-water particles but as temperatures are too low for significant evaporation, they could also include a nonvolatile core.

**3 Results**

**3.1 CN simulations and sulfuric acid profiles**

We start our simulations with a simplified *reference case* where we assume for all altitudes (18 - 32 km) and for every month (July - October) a constant monodisperse background CN concentration of 10 $cm^{-3}$ with a size of 100 nm diameter. For this reference case, we do not simulate the highest altitudes in September and October, as there, high temperatures lead to CN evaporation and complicate the interpretation. For all other altitudes and months, we simulate one month with constant ambient conditions chosen according to Sect. 2.2. We use the temperatures reported by Campbell and Deshler (2014) which are reproduced in Fig. 1a. We set the 10 CN $cm^{-3}$ as initial particles at the beginning of the month and simulate the month without gaseous sulfuric acid being present. The CN concentration then reduces somewhat over time as the particles

Moved down [1]: 1a.
Moved down [2]: 1a).
Moved down [3]: Mills et al.
Moved down [4]: The comparison is shown in Fig.
Moved (insertion) [1]

[revised manuscript text omitted]

Moved (insertion) [5]

Deleted: The result of this combined sensitivity test is shown in Fig. 5c. An examination of the individual influences (not shown here) leads to the following conclusions: The changes of the thermodynamic values for the charged clusters influence the derived mixing ratios in the area where ion-induced nucleation occurs. The changes due to the varied coagulation coefficients introduce a little shift to all values. The updated dimer stabilities with relative humidity dependence have a significant influence on the derived sulfuric acid mixing ratios in the regions of low relative humidity (higher temperature), 
[revised manuscript text omitted]

Deleted: Yu, F., and Turco, R. P. (1998). The formation and evolution of aerosols in stratospheric aircraft plumes: Numerical simulations and comparisons with observations. Journal of Geophysical Research: Atmospheres (1984–2012), 103(D20), 25915-25934.

[Figure]

Figure 1: Temperatures (a) during Antarctic winter above McMurdo, Antarctica (78°S), as presented in Campbell and Deshler (2014). The dashed line shows the lower temperature limit for which the SAWNUC model is valid and at which lower temperatures were kept fixed. In (b), corresponding sulfuric acid profiles are shown that lead to a 10 % CN increase by nucleation and growth to observable size during one month. For these *nucleation threshold* profiles, we assume a monodisperse CN background of 10 cm$^{-3}$ with 100 nm diameter at all altitudes (18-32 km) for every month (July-October).

Deleted: ), and 5 days (b). Temperature (red), sulfuric acid (blue), and the other ambient conditions are changed at the beginning of each model time step (1 month or 5 days) and kept constant during the model time step while the CN (> 5 nm) concentrations (black) are simulated. The gaseous sulfuric acid was varied until the number concentration of CN particles matched the observations. At the beginning of each time step the CN concentration shifts abruptly as the changed ambient conditions lead to an adiabatic compression. The higher time resolution in b) breaks these big shifts into smaller more frequent shifts. ⸱  ... [25]

[Figure]

Figure 2: Air parcel subsidence trajectories (a) and simulated CN (monthly mean) without gaseous sulfuric acid being present (b). The uncertainty ranges of the measured CN presented in Campbell and Deshler (2014) are shown as shaded areas in (b) for comparison. The trajectories of the simulated air parcels were placed around the subsidence of the measured CN maximum (red). In the simulation, the ambient conditions are kept constant during each month. For the first month of each trajectory, the CN concentrations are chosen based on Campbell and Deshler (2014). In the following months, the simulated CN concentrations are the result of only coagulation and air volume compression, as there is no gaseous sulfuric acid present.

[Figure]

[Figure]

Figure 3: CN layer gaseous sulfuric acid profiles (a) and the simulated CN using these profiles (b). We derive the sulfuric acid if the simulated CN concentration in Fig. 2b is too low without gaseous sulfuric acid and therefore nucleation and condensational

growth being present. The uncertainty ranges of the measured CN from Campbell and Deshler (2014) are shown as shaded areas for comparison.

[Figure]

Figure 4: Combination of the nucleation threshold sulfuric acid profiles from Fig. 1b (solid) and the CN layer sulfuric acid profiles from Fig. 3a (dashed). Additionally, we show sulfuric acid profiles that cause a CN increase in our CN simulation of Fig. 3b (dotted) which should represent upper limits of the Antarctic winter stratospheric sulfuric acid outside the CN layer.

[Figure]

**Figure 5: Comparison of the nucleation threshold sulfuric acid profiles derived including ion-induced nucleation (solid lines) and without simulating ions (dotted lines). At low sulfuric acid concentrations the derived profiles do not change. The CN layer profiles also hardly change (thick dashed lines; grey and light green are without ions and black and green are with ions, but they are almost identical).**

[Figure]

**Figure 6**: Sensitivity studies varying (a) CN counter cutoff size, (b) preexisting particle size, (c) model thermochemical and dynamic parameters, and (d) temperature, to estimate the uncertainties of the derived sulfuric acid profiles. As in Fig. 5, the solid and dark dashed lines show the nucleation threshold and CN layer formation profiles as presented in Fig. 4. The dotted and light dashed lines show the changed profiles according to the sensitivity tests.

[Figure]

**Figure 7: Comparison of our derived Antarctic sulfuric acid profiles (nucleation threshold: solid, CN layer: long dashed) with the derived profile from Campbell and Deshler (2014) (dark red, short dashed), inferred Arctic sulfuric acid from measurements in January presented by Krieger and Arnold (1994) (brown, short dashed), and mid latitude measurements and modeling of Arnold et al. (1981), Reiner and Arnold (1997), Schlager and Arnold (1987), Viggiano and Arnold (1981), and Mills et al. (2005) (shaded area). The September nucleation threshold profile for nucleation and growth to a lower cutoff of 6 nm from Fig. 6a is also included (black dotted).**

[Figure]

**Page 3: [1] Deleted**                      **The Authors**                      **14/12/16 12:16**

They simulated the year 2010 and compared the results to measurements above McMurdo Station (Campbell et al., 2014). The model reproduces the CN layer but at higher altitudes (around 30 km in the model vs. around 23 km in the observations). As an explanation they suggested (among others) biases in the critical nucleation variables: temperature, sulfuric acid and water concentration. However, there are no Antarctic stratospheric sulfuric acid measurements. Therefore, they inverted the nucleation equation to calculate the sulfuric acid concentration that corresponds to the CN increase over three weeks between two measurements. Their derived profile indicates that in the global model the sulfuric acid also has a shift towards higher altitudes which could explain the altitude shift of the simulated CN layer.

**Page 3: [2] Deleted**                      **The Authors**                      **14/12/16 12:16**

The

**Page 3: [3] Deleted**                      **The Authors**                      **14/12/16 12:16**

However, they did invert a nucleation formulation that describes

**Page 3: [4] Deleted**                      **The Authors**                      **14/12/16 12:16**

nucleation model SAWNUC, which simulates all these processes, to

**Page 3: [5] Deleted**                      **The Authors**                      **14/12/16 12:16**

and to investigate the processes that influence the nucleation of the CN layer. We also derived estimates of the corresponding sulfuric acid production rates

**Page 3: [6] Deleted**                      **The Authors**                      **14/12/16 12:16**

The model and the derivation process are described in Sect. 2. The derived profiles, the role of ion-induced nucleation, the uncertainties and estimated sulfuric acid production rates are presented and discussed in Sect. 3. Then, the derived profiles are compared to the global modeling of Campbell et al. (2014) in Sect. 4.

**Page 3: [7] Deleted**                      **The Authors**                      **14/12/16 12:16**

or chamber walls. Condensation, coagulation, and preexisting loss rates are calculated based on the hard sphere collision theory of Fuchs (1964). For the charged clusters, the Coulomb forces are calculated based on the intercluster potentials (Yu and Turco, 1998). For the neutral clusters, thermodynamic stabilities were calculated with the On-line Aerosol Inorganics Model (Carslaw et al., 1995). The neutral thermodynamics are adjusted to reproduce experimental nucleation rates of Ball et al. (1999). The thermodynamic stabilities of the negative clusters are calculated with the Thomson equation. However for small clusters, the values reported by Froyd and Lovejoy (2003a) and Lovejoy and Curtius (2001) are directly implemented into the model, which are based on experimental values and quantum chemical calculations (for more details see Lovejoy et al., 2004). For the neutral dimer and trimer, the thermodynamic stabilities presented by Hanson and Lovejoy (2006) are also implemented into the model. The SAWNUC model (Lovejoy et al., 2004) has been previously used (among others) in Ehrhart and Curtius (2013) and its parameterized version PARNUC (Kazil and Lovejoy, 2007) in Kirkby et al.

| Page 3: [8] Deleted | The Authors | 14/12/16 12:16 |
| --- | --- | --- |

.

2.2 Deriving

| Page 3: [9] Deleted | The Authors | 14/12/16 12:16 |
| --- | --- | --- |

**profiles**

This section describes the derivation of the Antarctic stratospheric

| Page 3: [10] Deleted | The Authors | 14/12/16 12:16 |
| --- | --- | --- |

profiles. The profiles are based on the measured CN concentrations and temperatures above McMurdo Station, Antarctica (78°S) as presented by Campbell and Deshler (2014, Fig. 1). Corresponding sulfuric acid concentrations and mixing ratios were derived for the time interval of June/July to October at altitudes from 18 km to 32 km. The SAWNUC model was "inverted" by performing multiple simulations with the same ambient conditions but varying sulfuric acid concentrations, and searching for the sulfuric acid concentration that reproduces the observed CN. Thereby, all effects like

| Page 4: [11] Deleted | The Authors | 14/12/16 12:16 |
| --- | --- | --- |

in July. This profile was moved down by 1 km every month (to represent diabatic descent within the polar vortex). The diameter of preexisting particles was assumed to be 100 nm. A surface area of 0.2 $\mu m^2$ $cm^{-3}$ was chosen for 215.15 K

| Page 4: [12] Deleted | The Authors | 14/12/16 12:16 |
| --- | --- | --- |

30 km altitude, which is consistent with Zhao et al. (1995) who reported that the subsiding mesospheric air is very clean and with Campbell et al. (2014) reporting a surface area of 0.3 $\mu m^2$ $cm^{-3}$ for 20-30 km altitude in early August 2010. This surface area was converted to each temperature/height combination according to the ideal gas equation. Sensitivity tests concerning the influence of all these input values on the derived profiles were performed and are presented in Sect. 3.2.

**2.2.2 The CN layer trajectory**

With the described model setup, the sulfuric acid profiles were derived by using the SAWNUC model as a box model, simulating the nucleation process inside air parcels over the period of the four months. The most important air parcel trajectory for this study is the one that connects the monthly maximum of the measured CN (in the following termed "maximum trajectory"). It is assumed that this maximum of the particle concentration resides in a single air parcel that descends inside the polar vortex. This section describes the derivation process of the sulfuric acid values in the maximum trajectory.

Sulfuric acid concentrations were searched for on a monthly basis as the measured input values (CN and temperature) are also monthly averages. The ambient conditions and the sulfuric acid concentration were set at the beginning of a month and kept constant for the entire month. Simulation results were evaluated for each month. The

simulated CN concentrations averaged over the month were compared to the measured values. If there were too many, the input sulfuric acid concentration was decreased and the model run was repeated. If the model did not produce enough CN, the input sulfuric acid concentration was increased. This process was reiterated until the sulfuric acid concentration was determined that corresponds best to the measured monthly CN concentrations. This resulted in a derived sulfuric acid concentration of the simulated month.

This scheme was used for every month. The time evolution of temperature (representative for all ambient conditions), the derived sulfuric acid volume mixing ratios (converted from the concentrations), and the simulated CN concentrations (> 5 nm) of the maximum trajectory's boxmodel simulation are shown in Fig.

| Page 4: [13] Deleted | The Authors | 14/12/16 12:16 |
|---|---|---|

The derivation process is divided into five periods: the initialization phase and the June/July, August, September, and October periods (June/July are combined as the

| Page 4: [14] Deleted | The Authors | 14/12/16 12:16 |
|---|---|---|

, also combine these months).

Model initialization is necessary as some CN should already be present at the beginning of the first derived month. However, as the CN concentrations before June are not known, the trajectory simulation was started with an initialization phase from May 1st to June 15th, in which it builds up 110% of the June/July CN amount at June/July ambient conditions. This extra 10% was chosen because CN concentrations are expected to be higher at the beginning of the June/July period than at its end (because of missing sulfuric acid production and air compression, see below).

| Page 4: [15] Deleted | The Authors | 14/12/16 12:16 |
|---|---|---|

June/July sulfuric acid concentration was derived by searching for the sulfuric acid concentration that reproduces the measured June/July CN amount. This sulfuric acid concentration is lower than during the initialization period as no new particles have to form and the existing ones have to decrease in number.

At the beginning of August the ambient conditions were changed to the August conditions (see temperature change in Fig.

| Page 4: [16] Deleted | The Authors | 14/12/16 12:16 |
|---|---|---|

As the pressure increases during the descent of the air, this ambient change also includes a compression of the air volume and thereby an increase in the CN per $cm^3$. This is seen in the CN jump at the beginning of August. The compression due to pressure increase is stronger than the expansion due to the temperature increase (both were calculated and combined). After the air compression, the sulfuric acid concentration was searched that reproduces the measured August mean CN amount. Here, a decrease from the compression-increased CN amount is necessary. Therefore, what seems like an increase in CN from July to August in the maximum trajectory values turns out to be a decrease due to the adiabatic compression.

At September 1st the ambient conditions were changed to the September conditions, the CN were compressed (only a small change can be discerned in Fig. 1a), and the sulfuric acid concentration was searched for that reproduces the September CN amount.

Finally in October the same procedure was used. Unfortunately, the use of a month-long time interval for temperature and sulfuric acid concentration produces an undesired increase of CN at the beginning of October (additional to the volume compression). Stable clusters below the counting threshold of 5 nm still exist from September and rapidly grow due to the strongly increased sulfuric acid concentration in October. Only later in the month, the CN amount decreases as only few new particles are produced while the old CN coagulate and are lost to preexisting particles. This unrealistic behavior can only be addressed by modeling at higher time resolution where the ambient conditions are changed more gently in shorter time steps.

Therefore, to avoid this undesired increase at the beginning of October and to reduce the „compression jumps", the same derivation process was preformed with shorter time steps of 5 days. However, time developments of altitude, temperature, and CN at the shorter time steps had to be assumed as the measured values are presented as monthly averages. The time developments were described by assumed functions for which the monthly mean values match with the measured values. Therefore, the simulated maximum trajectory with 5-day time steps (Fig. 1b) follows these assumed functions. The adjustments of the CN concentration to the air compressions are still visible at the beginning of every time step, but they are much smaller compared to the monthly simulations as the ambient changes are much smaller. Comparing the monthly average of the 5-day sulfuric acid mixing ratios with the derived monthly values shows that the monthly simulation overestimates the derived sulfuric acid for June/July to September and underestimates it for October. However, the difference always stays below a factor of 2. As the derived mixing ratios in this study span over several orders of magnitude, this is considered a reasonable agreement. This comparison strengthens the confidence in the monthly derived sulfuric acid values which are used in the rest of this study.

The derived values comprise four sulfuric acid concentrations / mixing ratios for the months June/July to October. Sulfuric acid is at very low mixing ratios (below 0.1 ppt) in July due to lack of sunlight and therefore absence of sulfuric acid production during Antarctic winter. When sunlight returns in August, in the beginning the

decreases further, as the gas phase sulfuric acid production takes some time to become larger than the losses (the chemical lifetime of $SO_2$ by OH and thereby the sulfuric acid production time is about a month at 20-30 km altitude; SPARC Report No.4 chapter 2.4.1). Starting in September, the sulfuric acid amount increases first slowly and then strongly from September to October by one order of magnitude reaching a maximum of ~1 ppt.

**2.2.3 Complementing the profiles with more trajectories**

To complement the profiles, more sulfuric acid mixing ratios were derived by simulating more trajectories that start at different altitudes. The trajectories are derived from the maximum trajectory by considering that the

velocity decreases with decreasing altitude and that the polar vortex declines towards spring. Thereby, 23 trajectories starting at altitudes between 18 km and 40 km were simulated for this study. If the altitude was above 32 km, the initialization phase was extended until the trajectory arrived below 32 km.

| Page 4: [19] Deleted | The Authors | 14/12/16 12:16 |

For all of these trajectories the procedure of deriving the corresponding sulfuric acid concentration was used as described above with two exceptions: First, at some points the temperature was a bit below 190 K (max. 4 K below) which is below SAWNUC's temperature range. Therefore these temperatures were fixed at 190 K. This introduces little uncertainty to these values as the amount can be estimated with the sensitivity test concerning inter-annual temperature variations, presented below in Fig. 5d. Second, due to the design of the derivation process, the model was not able to reproduce the September CN amount of three trajectories in September at 27-28.5 km altitude. Therefore, the procedure was adjusted for these 3 data points, so that it does not search for the mean CN during this month but the CN are only required to match at the end of the month. This is also expected to introduce only a small additional uncertainty to these three September values as the specific CN amount only has a small influence on the derived sulfuric acid amount (see below).

Deriving the sulfuric acid concentrations and mixing ratios for all trajectories resulted in values for nearly every combination of month and altitude. All these values then were combined to the four derived Antarctic sulfuric acid profiles.

**3 Results and discussion**

The derived Antarctic sulfuric acid profiles are shown in Fig. 2. The general shape of the derived sulfuric acid profiles is plausible. In Antarctic winter the values are well below one part per trillion because there is no sulfuric acid production when the sunlight is missing. Then, with the return of sunlight in August the values increase at high altitudes and in September at all altitudes. In October, they reach maximum values of above 10 ppt. The reason for the higher sulfuric acid mixing ratios at high altitudes is probably that fairly large amounts of source $SO_2$ are transported downward from the mesosphere and that the actinic flux is high at these altitudes.

The figure also reveals that the temperature is the most important controlling variable for the sulfuric acid as the shapes of both profiles are very similar. This is because temperature and sulfuric acid concentration mainly determine the nucleation rate. A change in the sulfuric acid concentration by one order of magnitude leads to a change in the nucleation rate by also order(s) of magnitude. At a given temperature, there is a small sulfuric acid window where particle concentrations of 10 to 100 $cm^{-3}$ magnitude can exist. The nucleation rate has to be in this small window, otherwise there would be by far too many, or, too few CN. This small window is determined by the temperature. Therefore, whether 20 or 50 particles are present has only little influence on the derived sulfuric acid as this only changes the sulfuric acid inside this small window. So from the two main input parameters of the derivation process, the temperature controls the derived sulfuric acid's order of magnitude and the exact CN amount decides about the decimal places. This is why the derived profiles look very similar to the temperature profiles and

the CN layer's influence cannot be seen that clearly, however it is present indirectly as the magnitude of the CN defines the sulfuric acid window.

The derived sulfuric acid mixing ratios for October at altitudes above 25 km are very high. In fact the model calculations yield a particle distribution with a total sulfur mass that is higher than the total stratospheric sulfur at these altitudes. A detailed analysis reveals that at these high sulfuric acid mixing ratios the particles grow above 300 nm. However, as the measurements only counted particles below 300 nm, here the derivation mechanism fails. The reason for this is that at these high temperatures a water vapor mixing ratio of about 5 ppm leads to very low relative humidities and therefore very high particle evaporation rates. To compensate these high evaporation rates the derivation mechanism predicts these very high sulfuric acid concentrations to reproduce the observed particle amount. Based on this analysis we conclude that it seems unlikely that the particles above 25 km in October (and maybe also at the highest altitudes in September) are volatile, pure sulfuric acid water particles as evaporation rates are too high. The measurements of Campbell and Deshler (2014) show that there are some non-volatile particles present especially at high altitudes towards spring and also Curtius et al. (2005) observed non-volatile particles in the Arctic lower stratosphere. Therefore, we assume that most of the October particles above 25 km are non-volatile and therefore cannot be simulated with the SAWNUC model. Nevertheless, we show the too high sulfuric acid values in Fig. 2 as dotted line for completeness but omit these values for the rest of the study.

The derived sulfuric acid profiles cannot be compared directly to data from in situ or remote sensing measurements of Antarctic stratospheric sulfuric acid as such data does not exist to our knowledge. However, northern mid-latitude balloon-borne measurements have been published (Arnold et al., 1981; Reiner and Arnold, 1997; Schlager and Arnold, 1987; Viggiano and Arnold, 1981).

| Page 4: [20] Deleted | The Authors | 14/12/16 12:16 |

(2005) presented a summary of these measurements. A comparison of these measurements with our derived sulfuric acid concentrations shows good agreement (Fig. 3). The concentrations range from $10^4$ cm$^{-3}$ to above $10^7$ cm$^{-3}$ and they have a similar shape: low values at low altitudes with an increase to high values at high altitudes. Due to the different locations (43°N vs. 78°S) the altitudes cannot be compared directly as the tropopause is expected to be at lower altitudes above Antarctica. Therefore, our derived profiles should be shifted upwards for comparison which improves the agreement. Note that also the adiabatic expansion has to be considered when shifting the profiles upwards. The derived October profile has an uncertainty towards lower values (see below) which also increases the agreement with the measurements. The derived lower concentrations in June/July at high altitudes compared to mid-latitudes are an expected result of the spare sunlight during polar night. However, the low June/July concentrations are of the same order of magnitude as inferred Arctic sulfuric acid concentrations presented by Krieger and Arnold (1994) (Fig. 3). In summary, our derived profiles are generally in agreement with stratospheric sulfuric acid measurements from other latitudes.

**3.1 Ion-induced nucleation**

To study the role of ion-induced nucleation during the formation of the CN layer, the sulfuric acid profiles were derived again, but without simulating ions.

4. In some areas the removal of ions has nearly no effect on the derived profiles, however, in other areas the sulfuric acid mixing ratios increase by almost an order of magnitude. The regions that are most affected are the ones with higher temperatures. At low temperatures the neutral clusters are stable enough so that including an ion to the cluster does not increase its stability against evaporation in an amount that would change the nucleation rate. Therefore, even if ions are present and ion-induced nucleation occurs, it is not more efficient than neutral nucleation. At higher temperatures on the other hand, the neutral clusters are not as stable any more and including an ion to the clusters stabilizes them and increases the nucleation rate significantly. Thus to create the same amount of CN when no ions are present, more sulfuric acid is required than if ions were present simultaneously. In conclusion, in the areas of lower temperatures (which includes the formation area of the CN layer) the ions do not significantly influence the nucleation rate. However, in the areas of higher temperatures the CN are almost exclusively produced by ion-induced nucleation.

**3.2 Sensitivity studies**

To estimate the uncertainties of the derived sulfuric acid profiles, sensitivity tests were performed. Besides the 1-month time step for the derivation periods (already discussed in Sect. 2.2.2), significant uncertainties are mainly introduced by three factors: a) the uncertainty of the CN measurement cutoff size, b) the uncertainty of the climatological preexisting particle surface area, and c) uncertainties of the thermochemical data, the condensation and coagulation rates used in the SAWNUC model. The Antarctic stratospheric sulfuric acid is also expected to vary from year to year (e.g. depending on the strength of the diabatic descent of mesospheric air masses within the polar vortex).

**3.2.1 CN measurement cutoff size**

The first sensitivity test investigates the influence of the CN measurement cutoff size on the derived profiles. Campbell and Deshler (2014) reported that their CN counters' efficiencies were at around 75% for particles with 3 nm radius (6 nm diameter). So the 50% cutoff size should be below 6 nm diameter at the ground. However, they also reported that the cutoff could increase to 20 nm diameter at conditions of 20 km altitude. This, unfortunately, is a source of uncertainty for the measured CN concentrations.

For the derived profiles it was assumed that the measured CN are > 5 nm in diameter. For this sensitivity test the profiles were derived assuming a cutoff of 10 nm diameter. A higher cutoff means that the particles have to grow larger before they are counted, thus the required sulfuric acid concentrations have to increase in the nucleation areas. In October during the decay of the CN layer, the mixing ratios have to decrease though, because even if no nucleation occurs there still are stable, growing particles left that are smaller than the cutoff size. Therefore, the derived sulfuric acid mixing ratios would be smaller to produce less new CN and to keep the growth rates small enough so that some of these CN stay below the cutoff size. The result of this sensitivity test is shown in Fig. 5a and the predictions are confirmed. The July to September mixing ratios are higher and in October inside the CN layer the

mixing ratios are lower. Therefore, the 50% cutoff uncertainty of the CN measurements has an impact on the derived profiles, especially on the October profile.

**3.2.2 Preexisting particles surface area**

The second sensitivity test investigates the influence of the total surface area of the preexisting particles. For the derived profiles a surface area of 0.2 $\mu m^2$ $cm^{-3}$ at 30 km and 215.15 K was chosen, according to Campbell et al. (2014) reporting 0.3 $\mu m^2$ $cm^{-3}$ for 20-30 km altitude in early August 2010, and converted to the temperature/pressure conditions according to the ideal gas law. For this sensitivity test the surface area was increased to 0.5 $\mu m^2$ $cm^{-3}$ at 30 km and 215.15 K. The results show a significant influence on derived sulfuric acid mixing ratios below 1 ppt (Fig. 5b). In this area, mixing ratios increase by about a factor of 2 as at these conditions the losses to preexisting particles are in competition with the nucleation (Ehrhart and Curtius, 2013). Therefore, if the chosen surface area of 0.2 $\mu m^2$ $cm^{-3}$ is not representative for all years, the derived profiles will have to be shifted either to higher or lower values according to this sensitivity test.

**3.2.3 Model uncertainties**

The third sensitivity test investigates the influence of the model uncertainties on the derived profiles. First, the uncertainty of the measured stabilities of the negatively

**Page 8: [22] Deleted**      **The Authors**      **14/12/16 12:16**

The result of this combined sensitivity test is shown in Fig. 5c. An examination of the individual influences (not shown here) leads to the following conclusions: The changes of the thermodynamic values for the charged clusters influence the derived mixing ratios in the area where ion-induced nucleation occurs. The changes due to the varied coagulation coefficients introduce a little shift to all values. The updated dimer stabilities with relative humidity dependence have a significant influence on the derived sulfuric acid mixing ratios in the regions of low relative humidity (higher temperature), but only if neutral binary nucleation dominates there.

**3.2.4 Inter-annual variations and other tests**

The last test estimates how the derived profiles may vary from year to year as this study used measurements averaged over 30 years. To estimate the inter-annual variations, all measured CN concentrations were increased by 60% and all temperatures were increased by 5 K. Both changes should lead to higher sulfuric acid values which is confirmed by the results (Fig. 5d). As the sulfuric acid profiles are mainly controlled by the temperature, the increase is mainly due to the temperature increase. The inter-annual variations are significant but they do not change the order of magnitude of the profiles.

**Page 10: [23] Deleted**      **The Authors**      **14/12/16 12:16**

Uncertainties are introduced by the model design of monthly derivation periods as the monthly values are an overestimation in June/July to September and an underestimation in October. However, the cutoff measurement uncertainty leads to higher sulfuric acid values in June/July to September and lower values in October, so these two uncertainties are partly compensating each other. For the October profile however, the uncertainty towards lower

mixing ratios introduced by the uncertainty of the CN counter cutoff is bigger. At low sulfuric acid mixing ratios (<1ppt), the derived profiles have an uncertainty due to preexisting particles. The model uncertainties introduce uncertainties to the profiles towards higher or lower values at the areas of ion-induced nucleation and towards higher values at low relative humidities when neutral nucleation dominates. The sulfuric acid profiles are expected to vary from year to year. The June/July profile has some additional uncertainties which are discussed in the next section. Therefore, the October profile should have the highest uncertainty, followed by the June/July profile. The August and September profiles should be the most accurate, which is during the formation of the CN layer.

**3.3 Sulfuric acid production (estimate)**

To estimate altitude profiles of the sulfuric acid production rates the modeling process had to be extended. As described, the sulfuric acid profiles were derived by searching for a constant sulfuric acid concentration that produces the measured CN amount. The extended approach was to assume a constant sulfuric acid production rate instead of a constant sulfuric acid concentration, and to simulate the sulfuric acid molecule concentration. The production was simulated by continually adding molecules throughout the modelled time periods. This added amount describes the net production ($H_2SO_4$ production from $SO_2$ minus $H_2SO_4$ photolysis to $SO_2$, and potential other production processes). It does not contain the sulfuric acid that evaporates from freshly forming CN as this is now explicitly simulated by the model. However, the preexisting particles are not assumed to evaporate in the model, therefore this production term could also represent any sulfuric acid that evaporates from preexisting particles.

With this approach, sulfuric acid production profiles were derived with monthly production rates (Fig. 6). Their shapes look much like the derived sulfuric acid profiles, which is expected. They also range over 3-4 orders of magnitude from 0.5 to about 500 cm$^{-3}$ s$^{-1}$.

The June/July production rates need further investigation. They indicate that some (small) sulfuric acid production should occur even though there is nearly no sunlight during winter. This sulfuric acid could be produced by processes that do not require sunlight. Krieger and Arnold (1994) presented Arctic negative ion composition measurements and inferred gaseous sulfuric acid concentrations. They found strong evidence for an OH production process that does not require sunlight as they also observed sulfuric acid production during Arctic winter. They proposed OH production via ambient ions as additional sulfuric acid source and calculated a sulfuric acid production rate by ions of approximately 0.2-0.9 cm$^{-3}$ s$^{-1}$. Compared to our derived June/July production rates, these are lower (max. by one order of magnitude). This difference could be due to different ambient conditions in the Arctic compared to the Antarctic stratosphere or the OH production by ions explains our derived production rates only partly. Our derived sulfuric acid concentrations for June/July could be too high as the production and therefore the nature of the CN measured in June/July are unknown (there are no measurements in Antarctic fall). They could be more stable if they were not produced only by binary nucleation (but e.g. with meteoritic dust). Or they could be the result of an Antarctic fall nucleation event that was predicted by the global modeling of Campbell et al. (2014).

Estimates of Antarctic sulfuric acid production profiles are presented in Fig. 6. They do indicate some sulfuric acid production during Antarctic winter. However, they should only be considered as first estimates as they are one

additional step away from the measurements. Processes like preexisting particle evaporation, OH production from cosmic rays, and CN production by other processes could influence the derived production profiles.

**4 Comparison with global modeling of Campbell et al. (2014)**

Our derived sulfuric acid profiles can now be compared to the global modeling of Campbell et al. (2014) to discuss the origin of the CN layer's altitude shift in their model. For this, we compare our derived sulfuric acid profiles to the global model simulation presented by Campbell et al. (2014) in Fig. 2. We compare the values between early July and late October. Our sulfuric acid volume mixing ratios were derived for the altitude range of 18 km to 32 km. In the global model, we find corresponding sulfuric acid values roughly between 25 km and 38 km. In this range the high altitude July value and the low altitude October value match. Also high sulfuric acid mixing ratios at high October altitudes are found in the global model.

However, the global model simulates much lower sulfuric acid concentrations in July at low altitudes. They are the result of missing sunlight and therefore no sulfuric acid production. As discussed above, SAWNUC seems to predict sulfuric acid production in this region. However, as discussed, this production could need an additional process like OH production by ions that can produce sulfuric acid without sunlight. As this additional process is not simulated by the global model, it predicts lower sulfuric acid mixing ratios in July at low altitudes. Therefore, it could be possible that the global model's minimum sulfuric acid mixing ratio of about $1 \cdot 10^{-20}$ is a strong underestimation.

Nevertheless, the general orders of magnitude of our derived sulfuric acid profiles are mostly found in the global model output, though with a 7 km altitude shift to higher altitudes. Thus, the global model has an altitude shift in CN, temperature, and sulfuric acid. This is further strengthened by comparing the modelled $SO_2$ with satellite observations by Höpfner et al. (2013) which suggests an even stronger altitude shift. Thus our results support the suggestion by Campbell et al. (2014) that the altitude shift in their modelled CN layer seems to be a result of altitude shifts in the controlling variables of the nucleation.

Note however, that this result could increase the confidence in the simulated global extent of the CN layer. If only the temperature had an altitude shift and not the sulfuric acid, at each altitude there would be very different temperature / sulfuric acid combinations, resulting in different nucleation rates compared to the real stratosphere. However, as the sulfuric acid also has an altitude shift, the nucleation rates should be closer to reality and therefore also the simulation of the global extent of the CN layer.

**5 Summary**

Balloon-born measurements of stratospheric CN above McMurdo Station, Antarctica, reveal the presence of a CN layer of freshly formed particles at 21-27 km altitude in August to October. Campbell et al. (2014) showed the global extent of this CN layer with a global model that reproduced the production of the CN layer by binary sulfuric acid water nucleation. However, in their model the CN layer was located at too high altitudes. Unfortunately, no Antarctic stratospheric sulfuric acid measurements exist for comparison. Therefore, Campbell et al. (2014) derived sulfuric acid concentrations from the measured CN and temperatures. However, they did not use a microscopic

nucleation model that includes processes such as coagulation, ion-induced nucleation, and losses to preexisting particles. Therefore, the goal of the present study was to use the nucleation model SAWNUC as a box model to derive Antarctic stratospheric sulfuric acid profiles based on their measurements and to investigate the nucleation process.

The sulfuric acid profiles were derived for the altitudes of 18 - 32 km by simulating air parcel trajectories that descend inside the polar vortex. For each trajectory, monthly sulfuric acid values were derived by searching for the sulfuric acid amount that reproduces the observed CN at the given ambient conditions. The derived sulfuric acid concentrations (volume mixing ratios) are of the order of $10^4$ cm$^{-3}$ ($10^{-14}$) in July. In the following months the concentrations increase to about $10^7$ cm$^{-3}$ ($10^{-11}$) in October. They depend strongly on the temperature because at a given temperature the nucleation rate varies with the sulfuric acid amount, leaving only a small sulfuric acid window to reproduce the observed magnitude of CN. The derived sulfuric acid profiles compare well with measured mid-latitude profiles and also with inferred Arctic sulfuric acid concentrations.

Ion-induced nucleation occurs, however, at low temperatures it has no significant influence on the nucleation rates as the neutral clusters are already stable enough, so that an additional charge does not significantly increase their stabilities. However, at higher temperatures the neutral clusters are not as stable anymore and ion-induced nucleation becomes the dominant nucleation mechanism.

Uncertainties of the derived profiles are caused by uncertainties of the instrumental cutoff diameter of the CN counter used for the measurements, the uncertainties of the preexisting particles surface area, and model uncertainties. The October profile has an uncertainty towards lower values mainly due to the uncertainty of the CN measurement cutoff. Sulfuric acid mixing ratios below 1 ppt depend significantly on preexisting particles surface area. The profiles are expected to vary from year to year.

Estimates of sulfuric acid production rates range from 0.5 to about 500 molecules cm$^{-3}$ s$^{-1}$. Sulfuric acid production during Antarctic winter seems to be necessary to explain the measurements. This would require a second production process that does not require sunlight (e.g. OH production by ambient ions). However, the production rates should only be considered as first estimates as they could be influenced by a variety of processes that were not simulated with this model.

Finally, a comparison of the derived sulfuric acid profiles with the global modeling of Campbell et al. (2014) strengthens the assumption that the global model represents the processes in general but at too high altitudes as also the sulfuric acid seems to be simulated at too high altitudes.

[Figure]

**Page 15: [25] Deleted**          **The Authors**          **14/12/16 12:16**

), and 5 days (b). Temperature (red), sulfuric acid (blue), and the other ambient conditions are changed at the beginning of each model time step (1 month or 5 days) and kept constant during the model time step while the CN (> 5 nm) concentrations (black) are simulated. The gaseous sulfuric acid was varied until the number concentration of CN particles matched the observations. At the beginning of each time step the CN concentration shifts abruptly as the changed ambient conditions lead to an adiabatic compression. The higher time resolution in b) breaks these big shifts into smaller more frequent shifts.

[Figure]

[Figure]

**Figure 2: The derived**

**Page 20: [26] Deleted**          **The Authors**          **14/12/16 12:16**

[Figure]

**Figure 4: Comparison of the sulfuric acid profiles derived including ion-induced nucleation (solid lines, reference profiles) and without simulating ions (dashed lines). In the area of lower temperatures (earlier and lower) the neutral clusters are stable enough so that the ions do not significantly change the nucleation rates. In the areas of higher temperatures (later and higher) the neutral clusters are not as stable anymore and more sulfuric acid is needed to reproduce the observed CN when ion-induced nucleation is not simulated.**

**Figure 5: Sensitivity studies varying (a) CN counter cutoff size, (b) preexisting particles surface area, (c) model parameters, and (d) inter-annual temperature and CN amounts, to estimate the uncertainties of the derived reference sulfuric acid profiles. The solid lines always show the derived reference profiles as presented in Fig. 2. The dashed lines show the profiles for model runs with the input values changed according to the sensitivity test. The October profile should have the highest uncertainty, followed by the June/July profile. The August and September profiles should be the most accurate. Detailed description of the varied parameters and discussion can be found in the text.**

[Figure]

**Figure 6: Estimates of sulfuric acid production rates presented per volume and second (a) and as mixing ratio per second (b). Some sulfuric acid production during the Antarctic winter is predicted. However, the modeling results should only be considered as first estimates as they are one additional step away from the measurements. Processes like preexisting particle evaporation, dark OH production (e.g. from cosmic rays), and CN production by other processes could change or could be included in these production profiles.**

---

## Referee Report (RR1)

[referee-annotated manuscript omitted]

---

## Author Response (AR2)

**Author's response to the discussion paper:**

**Nucleation modeling of the Antarctic stratospheric CN layer and derivation of sulfuric acid profiles**

**Steffen Münch and Joachim Curtius**

We thank the reviewers for carefully reading our manuscript and for providing numerous helpful comments. The reviewer comments are given below in black font with our replies in blue font. Text that was changed in the manuscript is shown in red font.

Note that the simulations were not changed for this revision. Most changes concerned the writing style and additional information and sensitivity tests were added at some places.

**Answers to Referee # 1**

Version 2 of this paper is significantly improved in regards to the scientific significance and quality of the SAWNUC model results presented; however, it still needs improvement in some of the analysis methods and discussion of the results, and in the presentation quality and writing.

The main issues raised by both Reviewer 1 and 2 on Version 1 of the paper are well addressed, where major changes have been made to provide a more focused paper on Antarctic CN layer formation, while providing valuable results that include derived sulfuric acid profiles in the Antarctic stratosphere (no measurements available). Some of my concerns that should be addressed going forward include the 1) impacts from preexisting particles, 2) temperature effects and sensitivity simulations, 3) trajectories of subsiding air parcels, methods, and sensitivity simulations, 4) zero sulfuric acid conditions for coagulation assessment, 5) initial preexisting particle size sensitivity simulations, 6) fixed ionization rate with altitude, 7) comparison against mid-latitude sulfuric acid concentrations, and 8) main conclusions and writing style. Please see the annotated PDF copy of my edits, suggestions, and final questions. Thank you.

Thank you for carefully reading our manuscript again, pointing out the unclear sections, and making suggestions for improvements. Our detailed answers are given below. We first answer the larger points in combined answers and then answer to the remaining individual points.

**Preexisting particle size**

4.19-21 This is a difficult argument to sell, as the pre-existing particles would of course be a mixture of sizes. We actually do have measurements of the size distribution of pre-existing particles from OPC measurements (down to a optically detectable size of 300 nm in diameter ) during the time just before new CN formation above McMurdo (see Deshler or Hofmann papers), but you are correct that we don't know the size distribution of the smaller "pre-existing CN" that are below the optical detection limits (

However, we can make a more sophisticated test. Here we redo the simulations but instead of assuming all preexisting particles to be 100 nm, we now assume that 10% are 300 nm, 50% are 100 nm, and 40% of the preexisting particles are 50 nm in size.

Here we see that this reduces the simulated CN and increases the derived sulfuric acid profiles. However, the amount of this effect depends on the exact size distribution which is still unknown. We added this sensitivity test to the supplementary information.

[in 2.2][...] but we also perform sensitivity studies assuming different sizes in Sect. 3.3. [in 3.3] If we assume the initial preexisting particles to be a distribution of different sizes (e.g. 40% of 50 nm, 50% of 100 nm, and 10% of 300 nm particles), the coagulation efficiency increases and leads to less simulated CN and higher derived sulfuric acid profiles (Fig S1).

**Zero sulfuric acid condition**

4.30-31 This seems like an unphysical situation of zero sulfuric acid. See additional detailed comment below regarding coagulation condition.

5.15-16 I have somewhat a problem with this condition of zero sulfuric acid being present. You must qualify the limitations of this condition, such that your coagulation results are only representative of zero sulfuric acid condition (unphysical).

5.19 at zero sulfuric acid condition (unphysical) with no competitive effects.

6.32-7.3 Yes, this relates to my previous comments on the zero sulfuric acid condition, which is rather unphysical. At the least, this should be discussed prior to these results for clarity. Then, these statements can be made in a more clear and concise writing style.

We understand that it seems a little strange to make this unphysical assumption of zero sulfuric acid. To test if this unphysical state influences the results, we did a sensitivity test, were we derived all sulfuric acid and CN profiles with a minimum sulfuric acid concentration of 1e4 cm-3 instead of 0 cm-3. The changes in the CN profiles of Figure 2+3 are so small that they can not be seen with the eye. And also the sulfuric acid profiles of Figure 4 hardly change; see here an overlay of both versions with half transparency: